# Image Polysemy in Contrastive Vision-Language Learning

## Abstract

An image can be described with multiple captions, where the semantic meaning of each caption can be highly different. Depending on the semantics of the caption it may be more challenging to align it to an image. While contrastive learning, the dominant paradigm for aligning image and captions, may prefer easy captions that are semantically overlapping with the image, it is unclear how well contrastively trained vision-language models (VLMs) scale to harder captions. In this work we introduce a dataset with diverse image captions to benchmark a wide-range of VLM across caption difficulty levels. Our findings show that existing VLM struggle with caption diversity, and scale poorly to challenging captions.

## 1 Introduction

Recently, vision-language models (VLMs) have drawn notable interest in the task of aligning images with text through contrastive learning. Models like CLIP (Radford et al., 2021) and ALIGN (Jia et al., 2021) have shown impressive performance on downstream tasks such as image classification (Abdelfattah et al., 2024), visual grounding (Xiao et al., 2024), and cross-modal retrieval (Yu et al., 2022a), largely due to training on massive, uncurated image-text datasets that mirror the diversity of human communication. However, benchmark datasets like Flickr30k (Plummer et al., 2015) and MS-COCO (Lin et al., 2015) predominantly feature descriptive captions closely tied to the images, limiting their ability to thoroughly evaluate the full potential and limitations of VLMs. This narrow focus makes it challenging to assess how well VLMs scale to diverse and semantically complex image-text pairs. To address this gap, it is essential to develop benchmarks that push the limits of their performance and provide deeper insights into their understanding of real-world image-text relationships.

In real-world data, an image often corresponds to multiple captions, each varying in semantic meaning—a phenomenon known as *image polysemy*. This results in multiple valid image-text pairs, each reflecting a different interpretation of the image. For instance, as shown in Figure 1 the image can be paired with captions ranging from straigtforward captions like "Woman walking in the middle of a forest" to less descriptive interpretations such as "The wandering soul embraced by nature's tranquility". This diversity challenges VLMs trained with contrastive objectives, which tend to prioritize captions that directly mirror the visual content. While humans can associate all these captions with the same image, it remains unclear how well VLMs can generalize across such diverse semantic interpretations.

Obtaining diverse yet related captions for images is challenging. Human annotated datasets such as Flickr30k (Plummer et al., 2015) and MS-COCO (Lin et al., 2015) have multiple captions per image, but their strict annotation guidelines result in semantically homogeneous captions. While recent works (Parekh et al., 2021; Chun et al., 2022) have expanded MS-COCO with additional captions, the diversity of these datasets remains limited. Conversely, web-scale datasets (Sharma et al., 2018; Schuhmann et al., 2022; Desai et al., 2021; Gadre et al., 2023; Thomee et al., 2016) naturally contain highly diverse captions and interpretations, i.e., polysemy - which we demonstrate with quantitative metrics using our proposed clustering-based method. Here, following prior works (Sharma et al., 2018; Gadre et al., 2023), we refer to datasets as "web-scale" to denote datasets harvested directly from large-scale internet sources, typically consisting of millions or billions of uncurated image-text pairs. Yet, the lack of human verification in these datasets makes it difficult to reliably evaluate caption diversity and quality. Hence, it is necessary to strike a balance between the

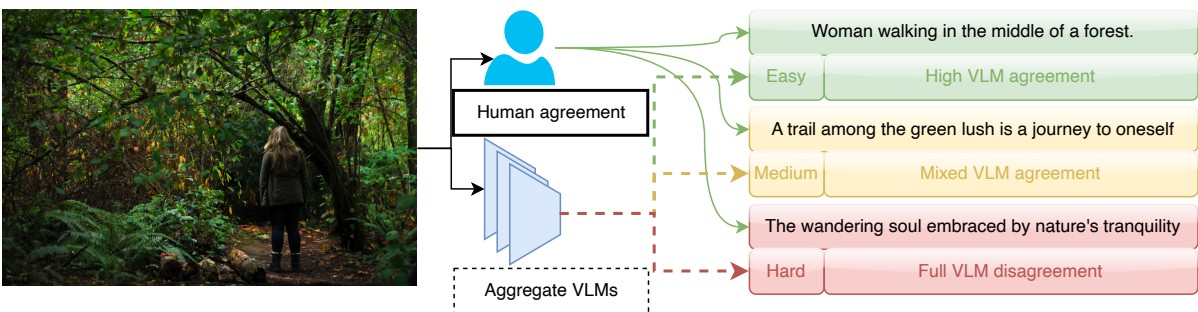

Figure 1: **Captions which pass human annotation have different levels of difficulty**. Humans can associate an image with semantically diverse captions, whereas vision-language models (VLMs) consistently struggle with certain captions. We determine caption difficulty through a VLM voting process: easy (high agreement), medium (partial agreement), and hard (low agreement)

reliable annotation of human-curated datasets and the semantic diversity of web-scale datasets for robustly evaluating VLMs.

We address the lack of datasets that strike this balance by proposing a new dataset, Image Polysemy (IMP), designed to test the alignment capabilities of VLMs for semantically diverse captions. IMP comprises 5K images, each paired with five captions curated from web-scale datasets (Sharma et al., 2018; Changpinyo et al., 2021) and verified with human annotations. During dataset construction, we observed that web-scale captions often include captions that are valid for humans but pose challenges for VLMs, particularly when addressing diverse semantic interpretations. To quantify this challenge, we define caption difficulty as the degree of misalignment between human and machine understanding, measured through an image-text matching task over aggregated results from multiple VLMs. While the easy subset includes captions similar to MS-COCO's highly descriptive style, the harder subset reflects the semantic complexity and diversity introduced by image polysemy, featuring captions that challenge the alignment capabilities of VLMs. To confirm the validity of the hard subset we conducted an additional human survey, which confirms the validity and quality of these image-text pairings.

To evaluate the challenges posed by diverse and semantically complex captions, we benchmark existing VLMs on IMP using image-text matching and cross-modal retrieval tasks, as shown in Figure 1. Our evaluation reveals that while VLMs perform well on simpler captions, their performance drops significantly as captions become richer and more complex. These results underline the need for more diverse evaluation datasets and emphasize the limitations of contrastive training objectives in handling real-world caption diversity. By providing a challenging yet systematic framework for evaluating VLMs, IMP aims to drive future research towards building models that generalize better across the spectrum of caption difficulty.

Our key contributions are as follows:

- **Systematically Quantify Polysemy in Web-scale Datasets:** We propose a clustering-based method that enable us to demonstrate the presence of polysemy in widely used web-scale datasets.

- **A Novel Dataset for Vision-Language Understanding:** We introduce IMP, a benchmark dataset designed to evaluate the ability of VLMs to align images with semantically rich captions, curated through a combination of web-scale data and human annotation.

- **Proxy-Based Difficulty Categorization:** We propose a proxy method to categorize image-text pairs into three difficulty levels of easy, medium, and hard, based on aggregated voting from state-of-the-art VLMs on an image-text matching task.

- **Comprehensive Evaluation of VLMs:** We benchmark a wide range of state-of-the-art VLMs on both image-text matching and cross-modal retrieval tasks, providing new insights into their

performance across varying levels of caption complexity. Our analysis reveals significant gaps in performance on semantically rich and complex captions.

## 2 Related work

### 2.1 Image Polysemy in Datasets

The challenge of modeling polysemy in VLMs is deeply tied to the structure and diversity of the datasets used during training and evaluation. Existing datasets can be broadly categorized into human-annotated datasets, web-scale datasets, and more recent specialized datasets that aim to address limitations in caption diversity or which are constructed through metric-based filtering.

Human-annotated datasets such as Flickr30K (Plummer et al., 2015) and MS-COCO (Lin et al., 2015) provide multiple captions per image, but these captions are often semantically similar, describing slightly different aspects of the same objects in a scene. While this consistency simplifies annotation and ensures reliability, it fails to capture the broader range of potential interpretations of real-world images. As a result, models trained or fine-tuned on these datasets tend to focus on descriptive captions and struggle with more abstract or subjective interpretations.

Web-scale datasets, such as Conceptual Captions (Sharma et al., 2018; Changpinyo et al., 2021) and Red-Caps (Desai et al., 2021), provide more diverse and abstract image-caption pairs by sourcing data directly from the web. These datasets introduce a wider variety of caption styles and semantics, including emotional and contextual nuances. However, most web-scale datasets (Schuhmann et al., 2022; Gadre et al., 2023) rely on a one-to-one mapping between each image and caption, limiting their ability to explicitly model polysemy. Moreover, the lack of multiple captions per image makes it difficult to evaluate whether models can handle different valid interpretations of the same image.

Several recent datasets have attempted to enhance caption diversity: CrissCrossed Captions (Parekh et al., 2021) augments MS-COCO with human-verified semantic similarity judgments, introducing additional intra- and inter-modality relationships. However, it focuses on improving diversity across different image-caption pairs, rather than explicitly modeling multiple interpretations of the same image. ECCV Caption (Chun et al., 2022) aims to address false negatives by adding high-quality captions missed in the original annotation process. While it helps models capture under-represented meanings, it focuses on correcting omissions rather than expanding the interpretative scope of captions.

Datasets based on CLIPscore like LAION (Schuhmann et al., 2022) and DataComp (Gadre et al., 2023), aim to improve dataset quality by filtering out noise. This approach enhances consistency in the dataset by removing low-confidence pairs, making it easier for models to learn clear image-caption relationships. However, the filtering process often eliminates "hard positives"—valid but semantically challenging captions. These hard positives are essential for training models that can handle polysemy, as they expose the model to different facets of an image. HYPE (Kim et al., 2024), introduces a ranking-based metric that filters for both multi-modal alignment and uni-modal specifity. HYPE emphasizes semantic accuracy while keeping complex, diverse captions in the dataset, making it a valuable resource for studying models' ability to navigate multiple interpretations.

### 2.2 Evaluating Image Polysemy

Evaluating how well VLMs handle image polysemy is challenging due to the difficulty of quantifying the semantic diversity of valid image-caption pairs. Current approaches can be categorized into human-based evaluation, model-based metrics, and ensemble methods.

Human evaluation is the most reliable method for assessing polysemy, as it directly involves annotators verifying whether multiple captions for an image are valid. This approach ensures high-quality judgments but is expensive, time-consuming, and difficult to scale for large datasets. Moreover, human evaluations are often subjective, with judgments varying across annotators. Despite these limitations, human evaluation remains essential for validating dataset quality and identifying hard positives (Emam et al., 2021).

Model-based metrics offer a more scalable and cost-effective alternative to human evaluation. Popular metrics include BERTScore (Zhang et al., 2020), which measures the semantic similarity between the embeddings of predicted and reference captions, and cosine similarity within the shared embedding space of models like CLIP. HYPE (Kim et al., 2024), is similarly used as a model-based metric to evaluate model performance on human-centered image-caption matching tasks. These metrics are easy to implement and inexpensive to compute, but they come with limitations. They are often unable to distinguish between **hard positives** (semantically diverse but valid captions) and **noise** (irrelevant or incorrect captions). As a result, while they provide a quick way to gauge performance, they are less effective in cases where captions are valid but semantically distant from the primary reference.

Ensemble-based methods aggregate outputs from multiple models to make a combined judgment on image-caption pairs, which is frequently used for model fine-tuning (Bai et al., 2024; Liu et al., 2024) and dataset filtering (Wu et al., 2022; Parekh et al., 2021; Chun et al., 2022). The underlying strategy is that multiple models run on the same samples, and their outputs (e.g., cosine similarity scores) are aggregated to assess the overall quality. This approach allows pooling the strengths of different models, potentially improving robustness in filtering out noise or irrelevant captions. The aggregation process typically favors captions that are consistently rated high by multiple models, which might overlook cases of legitimate semantic diversity. On the other hand, the disagreement across different models can help identify misjudged pairs. For example, CLIP-blind pairs (Tong et al., 2024) are found by comparing the cosine similarity between CLIP and vision-only self-supervised learning models like DINOv2 (Oquab et al., 2024). These pairs show cases where CLIP vision transformers mis-encoded images, verifying the validity of aggregating model outputs.

For our dataset we opt for a mixed approach, as we first rely on human evaluation to curate a polysemy-aware dataset, ensuring that all captions associated with an image are valid and semantically diverse. Once this dataset is established, we aggregate outputs from multiple models on this human-verified dataset to obtain splits that allow for a more fine-grained analysis of a model's ability to handle polysemy. By combining human-validated data with model aggregation, this method provides a clear distinction between truly diverse captions (hard positives) and noise, which is often missed by model-only methods. Moreover, instead of aggregating outputs from multiple models directly, we split the dataset into subsets with different levels of difficulty based on model agreement. The resulting IMP dataset provides a challenging new benchmark for VLMs, featuring semantically diverse, human-verified captions.

## 3 Image Polysemy in Web-Scale Datasets

To demonstrate the presence of image polysemy in web-scale datasets we present an analysis of Conceptual Captions (Changpinyo et al., 2021) (CC12M); a widely adopted dataset for vision-language learning due to its scale and diversity. Unlike human-annotated datasets like MS-COCO, which are constrained by annotation guidelines, captions in web-scale datasets exhibit great variability. However, the one-to-one mapping between images and captions in these datasets makes it challenging to systematically measure polysemy—the presence of multiple valid interpretations for a single image. In this section, we explore the limitations of traditional metrics for measuring polysemy and propose a clustering-based approach to better reveal and quantify it.

### 3.1 Clustering to Identify Polysemy

To quantify polysemy in web-scale datasets, we propose a clustering-based approach to group visually similar images and examine the diversity of captions dynamically within these clusters. We use DINOv2 (Oquab et al., 2024) features for image representations to avoid the clip-pair-blindness issues described in (Tong et al., 2024). We clustering using HDBSCAN (Malzer & Baum, 2020) and reduce the dimensionality of features with UMAP (McInnes et al., 2020) for efficient processing. We apply this approach to a randomly sampled 5M subset of CC12M and the entire MS-COCO training set.

The one-to-one mapping in web-scale datasets limits our ability to directly evaluate the presence of polysemy. Existing metrics for caption diversity, such as SentenceBERT (Reimers & Gurevych, 2019) similarity or BERTScore (Zhang et al., 2020), are effective in measuring semantic similarity between pairs of captions, but they cannot quantify polysemy when only one caption is available per image. To address this limitation,

we consider the joint embedding space of CLIP with aligned image and text modality and compute the geodesic distance Fletcher et al. (2004); Oldfield et al. (2023) between the query image and all retrieved captions as a valid metric. In this way, we treat the image embedding as the cluster centroid and the retrieved captions as the cluster members. This allows us to measure the diversity among captions in the cluster but also consider the relevance of the captions to the image. This clustering is similarly performed in the low-dimensional space, however, all the metrics are computed in the original embedding space to ensure accurate evaluation.

This clustering approach leverages the poor ability of VLM to handle polysemy to identify polysemy. In particular, for the geodesic distance metric, we consider a set of visually similar images (in the CLIP embedding space); if CLIP were able to well-handle polysemy then the captions for this set of images would all be close to their respective images, and hence close to each other. However, we observe that (especially for CC12M) the captions paired with these images are far away in CLIP embedding space, even when the images are highly similar. By leveraging this mismatch in CLIP embedding space (i.e., images being similar but their respective captions being dissimilar) we are thus able to identify polysemy.

### 3.2 Insights for IMP Construction

The clustering and diversity analysis directly informed the construction of IMP. The results of the clustering analysis are visualized in Figure 3, for both the image-image similarity (DINOv2) and geodesic distance (CLIP). The left y-axis shows the DINOv2 similarity, for wich image-image similarity is undefined at radius (and thus zero) since only a single image is considered. From radius 5 onward, the image-image similarity naturally decreases as the radius grows, as including more images gradually leads to incorporating visually less similar images into the cluster. This drop occurs similarly for both datasets, but notably, CC12M maintains higher similarity scores compared to MS-COCO across all radii. This is not merely due to dataset size differences, but reflects inherent differences in visual embedding density. The image-caption geodesic distance on the right y-axis measures the semantic diversity between a query image and captions of visually similar images within the embedding radius. As the radius increases, we observe a substantial increase in geodesic distance for CC12M, indicating rapidly increasing semantic diversity. In contrast, MS-COCO shows minimal change, suggesting lower semantic diversity.

Within CC12M there are many more similar images as compared to MS-COCO. In particular, we observe that for CC12M the image similarity at $R = 100$ is comparable to MS-COCO at $R = 5$. To further illustrate the difference, when the radius is between 25 to 50, CC12M has an average image-image similarity around 0.75, Based on prior empirical results these can be considered as visually and semantically similar. On the other hand, the image-image similarity of MS-COCO drops to 0.60 at $R = 100$, indicating that the images are no longer similar for larger clusters. Notably, different from the one-to-one mapping case, the geodesic distance of these large clusters within CC12M is much higher than in MS-COCO, which shows that whilst

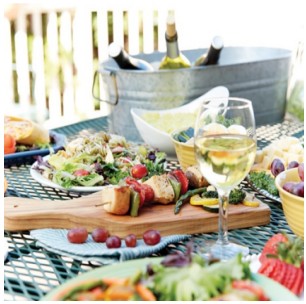 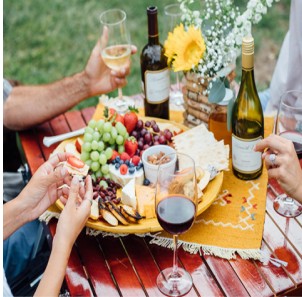 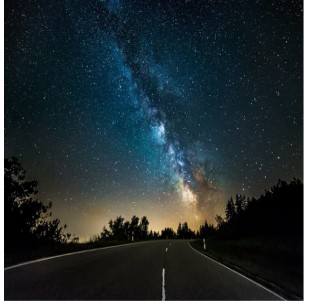 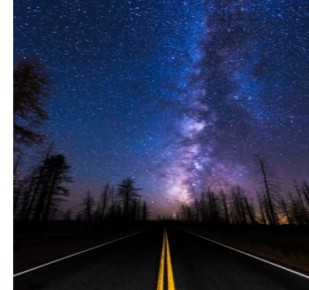

(a) Table filled with food from Creative Catering.  (b) The Secret Behind This Wine Club's Success  (c) The mystery milky way in the sky.  (d) Highway to Heaven ... like a shooting star.

Figure 2: **Polysemic image-caption pairs from the Conceptual Captions dataset**. While the images are highly similar, their captions are semantically diverse. For such cases, it is undesirable to map all images and captions close together in embedding space, yet contrastive learning aims to achieve this.

the clustered CC12M images remain similar their captions are semantically diverse. Moreover, as illustrated by the examples in Figure 2, these captions are valid pairings despite their semantic differences.

While increasing the cluster radius reveals polysemy, it also introduces noise in the form of irrelevant or low-quality captions. At very large radius, clusters may include images that are only marginally related, leading to captions that are no longer valid interpretations of the image. This highlights the need for careful selection of an optimal cluster radius that balances the visual relevance with textual diversity whilst avoiding noisy pairings. Through empirical analysis, we find that radius optimized for intermediate visual similarity (e.g., 0.70–0.80 cosine similarity in DINOv2 space) achieve this balance effectively. These clusters retain semantically diverse captions while minimizing the inclusion of irrelevant captions.

With our clustering analysis, we demonstrate that polysemy is implicitly but naturally present in web-scale datasets, through the existence of large cluster of visually highly similar images, which are described with semantically diverging captions. However, the consistent presence of noise and irrelevant texts alongside polysemy highlights the need for human annotation.

## 4 IMP

### 4.1 IMP: Image Polysemy Dataset

We build a new benchmark dataset, IMP, to study the impact of polysemy on vision-language models by leveraging the presence of polysemy in web-scaled datasets. An example is shown in Figure 4. To construct IMP, we follow the structure of the MS-COCO test split, with 5K images and 5 captions per image. Through our clustering-based approach and human labelling, we ensure that the captions for each image are relevant and diverse.

We randomly sample a subset of 5K images from the Unsplash Lite Dataset[1] as image source, which has an open usage license, and high-quality images that have been re-used across the web. For each image, we build a candidate pool of captions by using clustering-based approach in section 3 to retrieve captions from similar images found in CC3M and a 7M subset of CC12M. Additional captions are retrieved from websites where the images have been used. The candidate pool is then filtered through manual verification (by two

---

[1]`https://unsplash.com/data`

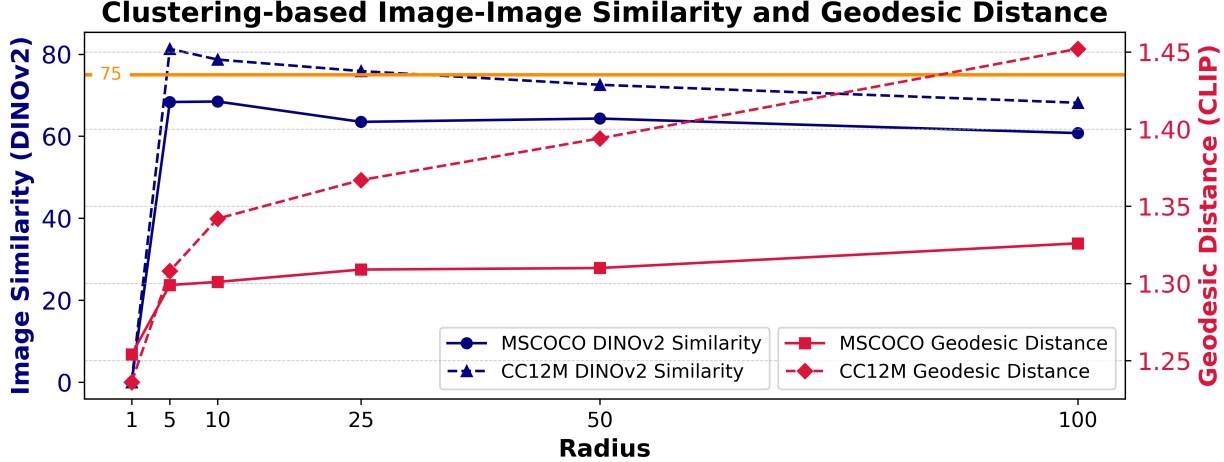

Figure 3: **Clustering results for CC12M (5M subset) and MS-COCO(Train)**.
The x-axis represents the radius of the cluster, the left y-axis represents the image-image similarity (DINOv2), and the right y-axis represents the geodesic distance (CLIP) between the query image and captions belonging to similar images. We can observe that for CC12M there is high textual diversity even for large clusters, which despite their size still have high image similarity.

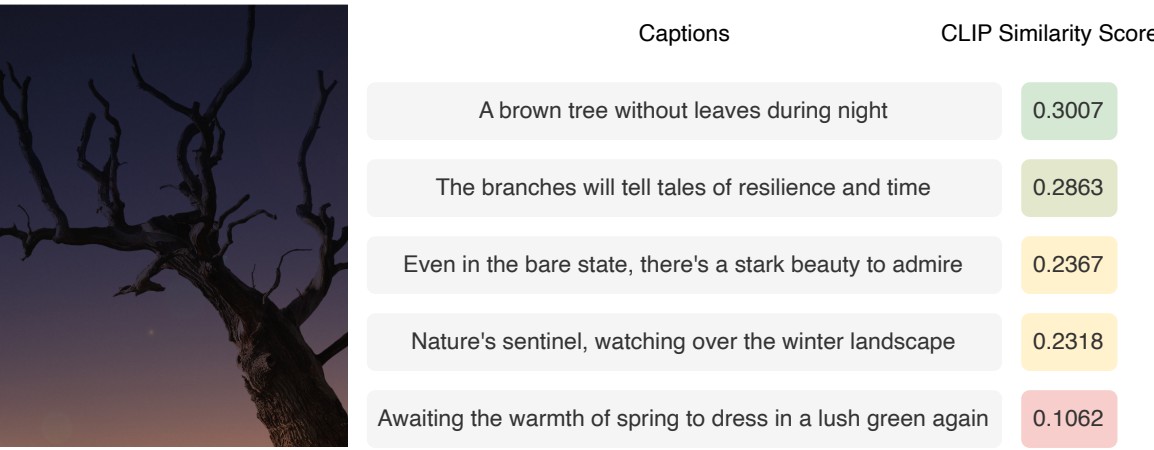

Figure 4: **Example of an image from IMP with its five captions**. The cosine similarity between the image and captions is computed with CLIP; only the most descriptive caption obtains a similarity above 0.3.

annotators) to ensure valid pairings. From the remaining captions we select five captions with the lowest pair-wise SentenceBert similarity, thereby maximizing the diversity of the captions.

We compare IMP with the MS-COCO test split by showing CLIP similarity density plots in Figure 5. We observe that the captions for each image (i.e., co-captions) within IMP have lower similarity than in MS-COCO, which implies higher degree of diversity. Similarly, the distribution of IMP image-text pair similarity is also shifted to the left, suggesting that IMP pairs are harder to match for vision-language models. In Figure 4 we show an example image from IMP, where the captions are semantically diverse. Moreover, only descriptive caption obtains a cosine similarity above 0.3 by CLIP, highlighting the difficulty of IMP. Further details on the dataset construction process, more examples, and additional analysis of IMP can be found in the Appendix C.

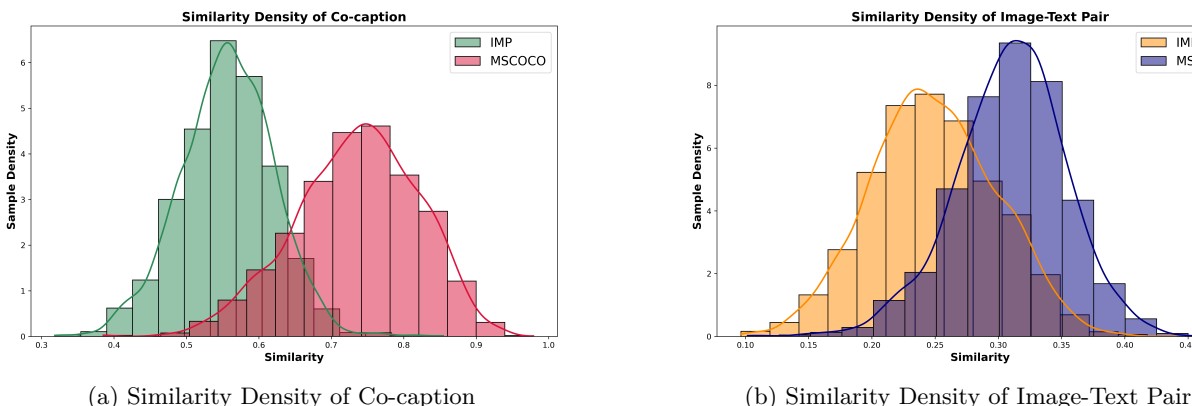

(a) Similarity Density of Co-caption            (b) Similarity Density of Image-Text Pair

Figure 5: **Similarity Density of IMP and MS-COCO-test**. On both metrics IMP shows lower similarity scores, which implies greater diversity than MS-COCO. The x-axis represents the cosine similarity score and the y-axis represents the density of the similarity score evaluated by CLIP.

## 4.2 IMP Division

To evaluate how effectively VLMs handle increasing semantic diversity, we divide IMP into splits of varying difficulty using six variants of CLIP, ranging from RN50 to CLIP ViT-L/14-336, and include an additional, more powerful model, CLIP ViT-g/14 trained on LAION-2B, as a tie-breaker in cases of disagreement. Before proceeding with evaluation on IMP, we first establish a zero-shot image-text matching similarity

threshold for each model. This is achieved by applying a precision-recall curve on a calibration dataset, DataComp (Gadre et al., 2023), because it is curated from web sources and filtered using CLIPScore, which helps ensure a high quality of true image-text pairs while maintaining a diverse set of content.

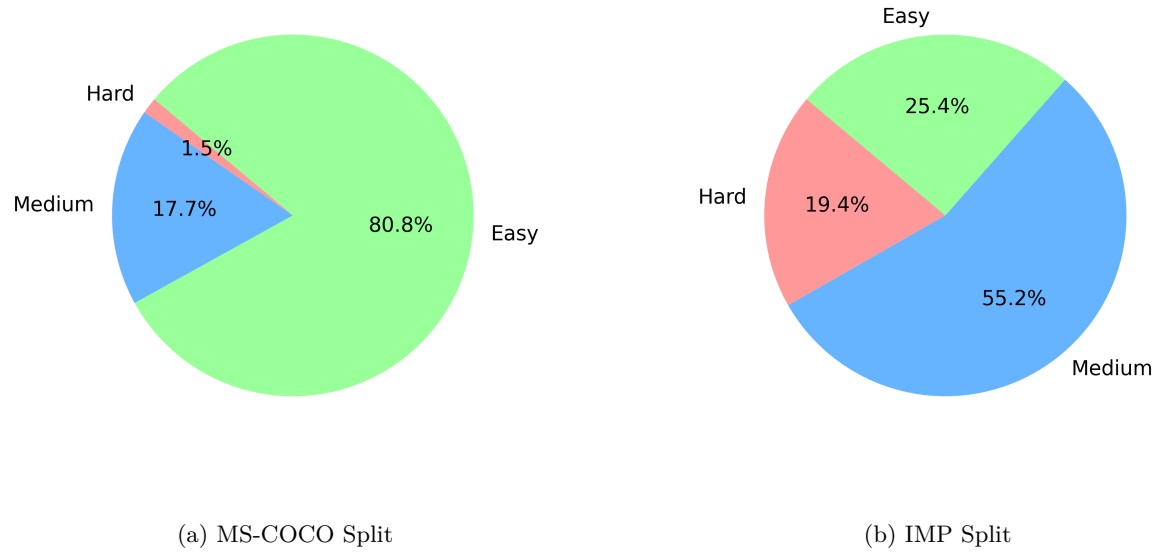

(a) MS-COCO Split          (b) IMP Split

Figure 6: **Pie chart of the Easy, Medium, and Hard subsets in MS-COCO and IMP**. The Easy subset is dominant in MS-COCO, while IMP has a more balanced distribution.

We sample a subset of 100k pairs from DataComp and, within each batch, treat the diagonal entries (matching image-text pairs) as positives and all off-diagonal entries as negatives. For each model, we compute the precision-recall curve and use the argmax of this curve to determine the optimal similarity threshold, which maximizes the precision-recall balance for image-text matching. This threshold ensures that each model is calibrated to distinguish between matching and non-matching pairs effectively in a zero-shot setting. The thresholds of CLIP ViT-B/32 and ViT-L/14 are 0.26 and 0.22 respectively, refer to approximately 40% fraction of the pool pairs (Gadre et al., 2023). Once thresholds are established, we use these models to evaluate the true positives in IMP by having each model "vote" on whether it agrees with the given image-caption pair. Based on the number of models that agree on the match, we classify samples from IMP into Easy, Medium, and Hard subsets.

To validate the robustness of our caption difficulty definition, we performed an additional experiment using a larger pool of 30 VLMs, with partial overlapping with CLIP models from Table **??**. Specifically, we randomly selected subsets of 7 models from these 30 VLMs and repeated our caption difficulty assignment 10 times. We then computed the internal consistency across these difficulty assignments using Cohen's Kappa, a standard measure of inter-rater agreement. Across all repetitions, we obtained Cohen's Kappa values between 0.65 and 0.87. In standard interpretation, Cohen's Kappa scores above 0.6 indicate substantial agreement, while scores above 0.8 represent almost perfect agreement. The consistently high values in our analysis clearly demonstrate that the caption difficulty assignment is robust and not overly sensitive to the specific choice of VLMs. Furthermore, we observed that caption difficulty categorizations are most sensitive to the accuracy extremes among the selected models—specifically, the strongest and weakest performers. For instance, if the weakest model improves its accuracy, certain captions initially categorized as "easy" shift to "medium". Conversely, if the strongest model performs worse, captions initially categorized as "hard" can shift to "medium". Such shifts reflect the intuitive notion that caption difficulty is inherently linked to model performance boundaries.

As a comparison, we also evaluate the MS-COCO test split using the same models and thresholds. As shown in Figure 6, the MS-COCO split is dominated by the Easy subset, with 80% of the samples falling into this

category. In contrast, IMP has the majority of its samples in the medium subset, and more than 20% of the samples are classified as Hard. We argue that due to the saturation of Easy samples in MS-COCO, there is limited room to continue benchmarking improved VLMs. IMP, on the other hand, provides a more challenging and diverse dataset.

### 4.3 Qualitative Analysis on the Hard Subset

To give further insight into the challenging nature of IMP and the benefits of the splits, we qualitatively analyze the hard subsets of both MS-COCO and IMP.

MS-COCO has 1.5% of its samples in the hard subset, which are cases where the model struggles to match the image and caption. We observe that the hard subset in MS-COCO primarily contains "objectively inaccurate" captions. These are cases where the caption does not describe the image accurately. Examples are shown in Figure 7.

In contrast, 20% of the samples in IMP fall into the hard subset, which are cases where models struggle to match the image and its polysemic caption. To verify the validity of these pairings we compare the results of human-evaluation with VLMs. For the evaluation, we pick the 100 hardest pairs (i.e., the pairs with lowest cosine similarity with CLIP embeddings) in the subset to investigate further. Based on these hard pairs, we performed a user-study with 40 participants, where each participant is shown a random subset of 25 pairs. To eliminate potential bias, the participants are told that each image-text pair can be either good or bad, and we added true negative distractor pairs to reinforce this. We report the results in Table 1 and showcase two hard examples in Figure 8. We observe that over 92% of the overall samples are rated as good pairs, with no sample dropping below 85% agreement. This suggests that there exists a gap between the way of VLMs and human to treat the polysemic captions and link them to images. For Figure 8a the human agreement is 100%, while the CLIP image-to-text similarity is only around 0.1. The second example 8b has 90.32 % human agreement compared to 0.09 similarity score. Additionally, through human-evaluation, we were able to verify that the hard samples are indeed polysemic, and not random noise or outliers. Notice the difference between MS-COCO example in Figure 7b and IMP example in Figure 8a. The former is an inaccurate caption, while the later is a metaforic caption.

In addition to the survey, we evaluate the seven CLIP models on each splits of IMP by computing the area under the precision-recall curve (AUPRC), the captions that failed to pass human annotation during construction are used as the negative samples. In total 5K negative samples are used, with one true negative for each image. We report the mean AUPRC and the error bars in Figure 2. This result shows that the performance of models diverges more when the samples are harder. The error bars are larger for the hard subset, which indicates the difficulty of the task.

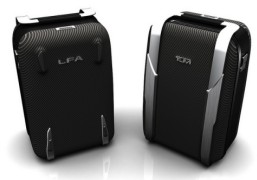 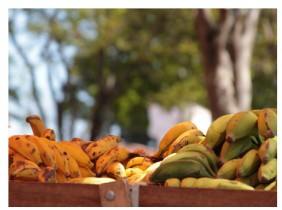 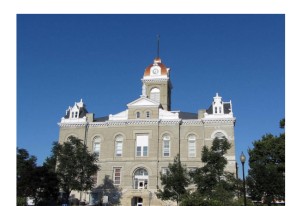 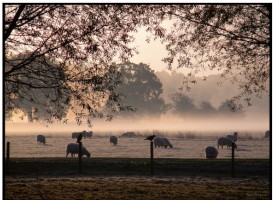

(a) Two black bags placed standing on the ground.  (b) Vegetables are displayed in a wooden barrel outdoors.  (c) A big tower that is surrounded by trees.  (d) Some animals are standing out in the water.

Figure 7: **Examples of imag-text pairs in MS-COCO hard subset.** The inaccurate captions are highlighted in red.

CLIP Similarity: 0.10, Human evaluation: 100%     CLIP Similarity: 0.09, Human evaluation: 90.32%

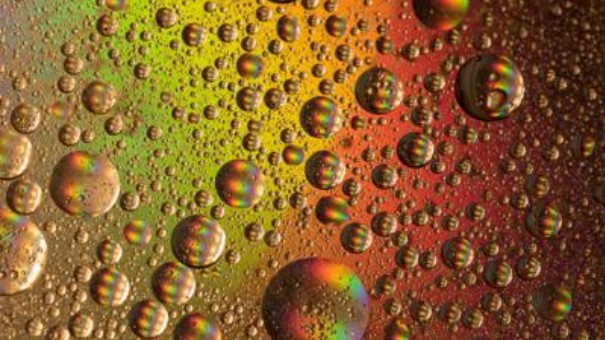

(a) A furry tide ebbs and flows across the green shore.     (b) A windowpane's temporary tattoos from the sky.

Figure 8: **Examples of image-text pairs in IMP hard subset.** Captions are polysemic and hard to match with the image by VLMs, but are rated as good pairs by human-evaluation.

Table 1: **Human-evaluation results**.
We report the mean and minimum agreement of the human-evaluation. The samples are equally split into four subsets of size 25. The result shows high agreement percentage.

Table 2: **AUPRC results**.
We report the mean AUPRC of the seven models on IMP with error bars.

| Samples | Mean Agreement | Min Agreement |
|---------|----------------|---------------|
| 25      | 92.3%          | 87.1%         |
| 100     | 92.2%          | 86.0%         |

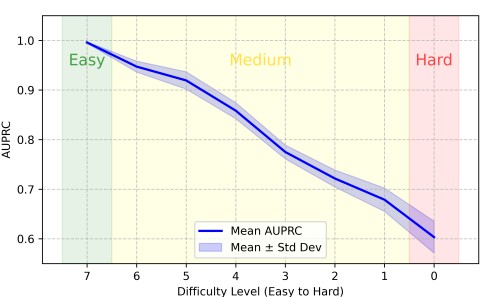

# 5  Benchmarking Polysemy in Vision-Language Models

To evaluate how VLMs handle image polysemy, we benchmarked them on the IMP dataset. Our study focuses on three key objectives: (1) Quantifying performance using cross-modal retrieval metrics (e.g., Recall@$K$, median rank) and classification metrics (e.g., AUPRC); (2) Assessing their performance across IMP's difficulty categories (easy, medium, hard); (3) Analyzing hard subset examples to identify failure points in aligning semantically diverse captions.

## 5.1  Evaluation Metrics

We employed a combination of standard cross-modal retrieval metrics and binary classification metrics to benchmark model performance, tailoring our approach to the challenges of image polysemy.

1. **Cross-modal retrieval:** We use Recall@$K$ (R@$K$) with $K = 1, 5, 10$, which is the percentage of queries that have at least one relevant item in the top-$K$ retrieved items (Song & Soleymani, 2019; Kim et al., 2023). However, while R@$K$ is a standard metric, it falls short in evaluating polysemy, as retrieving even a single easy caption can result in high scores without reflecting the ranking quality for other, more diverse captions. To address this, we include mean rank (meanR) and median rank (medR), which provide a more holistic view of the overall ranking of all relevant captions, capturing how well the model ranks diverse captions throughout the retrieval list (Parekh et al., 2021). In

addition, we follow ECCV captions (Chun et al., 2022) and report mean average precision at recall R (mAP@$R$), which focuses on the precision when the model retrieves a fixed number $R$ of items, emphasizing its ability to retrieve relevant items consistently.

2. **Binary image-text classification:** We use the area under the precision-recall curve (AUPRC) as the evaluation metric to evaluate the model's ability to distinguish between matching and non-matching image-text pairs. AUPRC is particularly useful for polysemic captions, as it emphasizes sensitivity to the positive class, making it suitable for evaluating performance on semantically diverse datasets like IMP.

## 5.2 Cross-modal Retrieval Evaluation

We evaluate three groups of state-of-the-art vision-language models (VLMs), categorized based on their architecture (Awais et al., 2023):

- **Dual-encoder VLMs:** These models consist of separate encoders for images and text, with a loss function computed between their outputs. Examples include CLIP (Radford et al., 2021), ALIGN (Jia et al., 2021), AltCLIP (Chen et al., 2022), ConvNeXt-CLIP (Liu et al., 2022), GroupViT (Xu et al., 2022), and Siglip (Zhai et al., 2023).

- **Fusion VLMs:** Fusion models incorporate a module that combines image and text features in addition to dual encoders, enabling richer pre-training tasks. Examples include ALBEF(Li et al., 2021), BLIP (Li et al., 2022), FLAVA (Singh et al., 2022), and COCA (Yu et al., 2022b).

- **Other VLMs:** This group encompasses diverse architectures such as encoder-decoder models (EVA-CLIP and EVA-02), models leveraging large language models as textual backbones (BLIP2), and models trained on multiple modalities (ImageBind (Girdhar et al., 2023)).

As most of the VLMs use the same or a larger visual backbone than ViT-L/14, we use CLIP with the ViT-L/14 pretrained on CLIP400M (Radford et al., 2021) as the baseline for comparison. Table 3 summarizes the zero-shot retrieval performance of all evaluated models on IMP, reporting key metrics such as Recall@$K$, mean rank, median rank, and mAP@$R$. These metrics collectively provide insights into how well the models rank semantically diverse captions and retrieve relevant image-text pairs without additional fine-tuning.

From the results, we observe that most VLMs exhibit significantly lower performance on the IMP dataset compared to MSCOCO. For example, CLIP ViT-L/14 achieves R@$K$ scores of [88.0, 98.7, 99.4] for image-to-text and [68.7, 90.6, 95.2] for text-to-image on MSCOCO, but these scores drop drastically on IMP. This underscores the increased difficulty of aligning images to semantically diverse captions in IMP. Among the evaluated models, EVA02 achieves the highest overall performance on IMP, except for mAP@$R$ on the text-to-image task, where ImageBind slightly outperforms it. This suggests that while larger models like EVA02 can better handle polysemy, challenges still remain in retrieval consistency for semantically rich captions.

We observe that ALIGN achieves better image-to-text R@$K$ scores than AltCLIP but performs worse in mean rank (meanR) and median rank (medR), as well as in text-to-image retrieval. This disparity highlights a potential limitation of ALIGN in ranking captions consistently across the entire spectrum of difficulty, even if its top results are relevant. Similarly, inconsistencies are observed across BLIP2 variants: the version with a ViT-L/14 backbone outperforms the ViT-g/14 variant overall, but fine-tuning on M-SCOCO negatively affects its ability to retrieve diverse captions, as seen in lower meanR and medR despite higher Recall@$K$. The gap between meanR and medR across most models further emphasizes that VLMs may treat polysemic captions as outliers. While medR reflects the ranking of the closest relevant caption, the much higher meanR indicates that other valid captions are often ranked far lower, leading to large discrepancies in retrieval quality. These findings highlight the need for models that better capture the semantic breadth of polysemy without over-prioritizing easier captions.

To further investigate the impact of model architecture, we benchmark three one-to-many approaches: PVSE (Song & Soleymani, 2019), PCME (Chun et al., 2021), and DivE (Kim et al., 2023). These models aim to address polysemy through specialized mechanisms. PVSE uses multiple local representations, while

Table 3: **Zero-shot cross-modal retrieval performance on IMP for SOTA models.** The best results within each column are highlighted with **bold**. The best results within each group of models are underlined.

| Method | Image-to-Text | | | | | | Text-to-Image | | | | | |
|---|---|---|---|---|---|---|---|---|---|---|---|---|
| | R@1 | R@5 | R@10 | meanR | medR | mAP@R | R@1 | R@5 | R@10 | meanR | medR | mAP@R |
| CLIP-ViT-L/14 | 15.8 | 30.9 | 38.8 | 2057 | 531 | 7.865 | 6.4 | 15.3 | 20.5 | 503 | 114 | 11.363 |
| AltCLIP | 17.3 | 33.1 | 42.4 | 1826 | 435 | 8.795 | 7.1 | 16.2 | 21.9 | 471 | 103 | 12.181 |
| ALIGN | 18.6 | 35.2 | 44.4 | 2292 | 580 | 8.331 | 6.2 | 15.0 | 20.1 | 536 | 125 | 11.101 |
| ConvNeXt | 15.3 | 31.2 | 40.2 | 2594 | 703 | 7.121 | 5.9 | 13.8 | 19.0 | 584 | 145 | 10.355 |
| GroupViT | 9.8 | 21.8 | 29.1 | 3773 | 1274 | 4.741 | 4.4 | 11.0 | 15.5 | 671 | 195 | 8.272 |
| Siglip | 20.3 | 37.7 | 46.8 | 2414 | 496 | 9.378 | 7.8 | 17.4 | 23.2 | 519 | 99 | 13.11 |
| ALBEF | 8.7 | 19.4 | 27.1 | 3811 | 1380 | 4.159 | 0.9 | 4.5 | 9.2 | 1933 | 1708 | 5.280 |
| BLIP | 13.3 | 27.8 | 36.0 | 3103 | 862 | 6.341 | 6.1 | 13.7 | 18.3 | 634 | 160 | 10.366 |
| FLAVA | 12.6 | 27.2 | 34.8 | 2692 | 824 | 6.122 | 6.3 | 14.8 | 20.4 | 503 | 115 | 11.134 |
| COCA | 19.6 | 36.9 | 46.2 | 1893 | 433 | 9.406 | 7.7 | 16.9 | 22.7 | 460 | 97 | 12.876 |
| BLIP2-g | 12.6 | 28.2 | 37.1 | 2627 | 616 | 6.751 | 5.6 | 13.9 | 18.9 | 563 | 138 | 10.283 |
| BLIP2-ViT-L | 16.6 | 33.3 | 42.1 | 2593 | 574 | 8.026 | 6.0 | 14.1 | 19.6 | 553 | 131 | 10.572 |
| BLIP2-g-COCO | 14.8 | 31.6 | 40.5 | 2970 | 659 | 7.124 | 5.6 | 13.8 | 19.3 | 552 | 132 | 10.315 |
| EVA-CLIP | 18.7 | 35.7 | 44.7 | 1986 | 453 | 8.979 | 7.6 | 17.2 | 22.9 | 459 | 94 | 12.884 |
| EVA02 | **20.6** | **38.1** | **48.2** | **1671** | **376** | **9.928** | 7.9 | **17.9** | **24.0** | **443** | **86** | 13.455 |
| ImageBind | 20.2 | 37.7 | 47.3 | 1813 | 402 | 9.650 | **8.3** | 17.8 | 23.6 | 451 | 88 | **13.565** |
| CLIP-ViT-B/32 | 14.7 | 29.1 | 37.8 | 2084 | 566 | 7.341 | 6.0 | 14.1 | 19.3 | 515 | 121 | 10.611 |
| PVSE($k = 5$) | 14.9 | 29.7 | 38.0 | 2061 | 561 | 7.359 | 6.0 | 14.3 | 19.6 | 512 | 123 | 10.624 |
| PCME | 15.0 | 29.6 | 39.0 | 2041 | 556 | 7.385 | 5.8 | 14.3 | 19.7 | 511 | 121 | 10.628 |
| DivE($k = 5$) | 15.4 | 30.0 | 38.5 | 1992 | 542 | 7.500 | 5.9 | 14.8 | 20.5 | 458 | 131 | 10.746 |

DivE introduces slot attention modules, and PCME employs probabilistic embeddings. We implemented these approaches using an adapter (Upadhyay et al., 2023) with LoRA parameters (Hu et al., 2021) on a frozen CLIP ViT-B/32 backbone and trained them on a 5M subset of CC12M. Experiments across multiple dataset scales (100K, 500K, and 5M) allowed us to compare their performance against the standard CLIP model, with results for the 500K subset presented in Table 3.

The one-to-many models achieve slightly better overall results than the baseline CLIP ViT-B/32 but still struggle to handle the full complexity of IMP. Among them, DivE demonstrates a marginal advantage, with lower meanR and higher text-to-image medR, suggesting that slot attention mechanisms improve ranking diversity. However, this improvement comes at the cost of increased text-to-image median rank, reflecting difficulties in ranking highly diverse captions for a given image. These findings suggest that while one-to-many approaches show promise for handling polysemy, further advancements are needed to balance diversity and retrieval accuracy.

To better understand retrieval performance across the easy, medium, and hard categories of IMP, we evaluated SOTA models by filtering retrieval results based on category. The results, summarized in Table 4, show Recall@5 (R@5) and median rank (medR).

The overall trends for the easy and medium categories are consistent with Table 3. Most models achieve low medR for the easy category, reflecting strong performance for simpler captions. However, medR increases for medium and hard categories, suggesting that ranking diverse or abstract captions poses greater challenges. Siglip achieves the highest R@5 in the medium category, indicating relatively better performance in handling captions of moderate complexity. For BLIP2, the ViT-g/14 variant fine-tuned on MSCOCO performs worse than the ViT-L/14 backbone on easy captions but achieves higher R@5 on the medium category, showing potential improvements for intermediate-level retrieval. On hard captions, while fine-tuned BLIP2 improves R@5, its medR worsens, which suggests that it struggles to rank all valid captions consistently. Additionally, FLAVA achieves the best medR for text-to-image retrieval in the hard category, though its R@5 remains low overall, indicating that it may capture certain nuances but not as effectively as other models.

Table 4: **Zero-shot cross-modal retrieval performance on IMP for SOTA models per category.** The best results within each column are highlighted with **bold**.

| | Image-to-Text | | | | | | Text-to-Image | | | | | |
| | Easy | | Medium | | Hard | | Easy | | Medium | | Hard | |
| Method | R@5 | medR | R@5 | medR | R@5 | medR | R@5 | medR | R@5 | medR | R@5 | medR |
|---|---|---|---|---|---|---|---|---|---|---|---|---|
| CLIP-ViT-L/14 | 36.5 | 506 | 4.1 | 538 | 0.0 | 555 | 11.9 | 6 | 3.3 | 149 | 0.1 | 1129 |
| AltCLIP | 38.1 | 415 | 6.6 | 433 | 0.0 | 481 | 12.1 | 6 | 4.1 | 133 | 0.1 | 1035 |
| ALIGN | 34.5 | 550 | 9.8 | 578 | 0.2 | 640 | 10.3 | 10 | 4.5 | 150 | 0.1 | 1122 |
| ConvNeXt | 32.4 | 696 | 6.5 | 683 | 0.0 | 772 | 10.1 | 10 | 3.6 | 177 | 0.1 | 1279 |
| GroupViT | 21.3 | 1285 | 4.5 | 1241 | 0.0 | 1356 | 8.1 | 17 | 2.8 | 235 | 0.1 | 1328 |
| Siglip | 38.7 | 486 | **11.2** | 509 | 0.1 | 482 | 11.7 | 7 | **5.5** | 124 | 0.1 | 1076 |
| ALBEF | 18.3 | 1395 | 4.0 | 1354 | 0.0 | 1445 | 0.3 | 1185 | 0.1 | 1776 | 0.0 | 2271 |
| BLIP | 27.6 | 867 | 6.8 | 831 | 0.1 | 944 | 9.7 | 12 | 3.8 | 198 | 0.1 | 1261 |
| FLAVA | 26.4 | 805 | 7.0 | 825 | 0.1 | 852 | 10.2 | 10 | 4.4 | 142 | 0.1 | **936** |
| COCA | 41.1 | 427 | 8.5 | 429 | 0.0 | 465 | 12.2 | 6 | 4.6 | 121 | 0.1 | 1018 |
| BLIP2-g | 27.3 | 618 | 8.4 | 591 | 0.2 | 658 | 9.5 | 11 | 4.2 | 168 | 0.1 | 1090 |
| BLIP2-ViT-L | 33.5 | 567 | 9.1 | 549 | 0.1 | 652 | 10.0 | 10 | 4.0 | 164 | 0.1 | 1044 |
| BLIP2-g-COCO | 28.9 | 666 | 9.6 | 633 | **0.3** | 717 | 9.2 | 12 | 4.5 | 159 | **0.2** | 1011 |
| EVA-CLIP | 39.9 | 439 | 7.7 | 453 | 0.0 | 482 | 12.5 | 6 | 4.6 | 119 | 0.1 | 1019 |
| EVA02 | **41.7** | **361** | 9.2 | **377** | 0.1 | **406** | **12.7** | **5** | 5.1 | **111** | 0.1 | 946 |
| Imagebind | 41.1 | 399 | 9.1 | 394 | 0.0 | 437 | **12.7** | **5** | 5.0 | 113 | 0.1 | 1020 |
| CLIP-ViT-B/32 | 33.5 | 562 | 4.4 | 555 | 0.0 | 602 | 10.8 | 8 | 3.3 | 150 | 0.0 | 1291 |
| PVSE($k=5$) | 34.5 | 551 | 5.2 | 540 | 0.0 | 591 | 10.9 | 8 | 3.3 | 148 | 0.0 | 1285 |
| PCME | 35.2 | 543 | 5.3 | 539 | 0.1 | 585 | 10.6 | 8 | 3.5 | 145 | 0.1 | 1277 |
| DivE($k=5$) | 35.7 | 538 | 5.3 | 535 | 0.1 | 576 | 10.7 | 8 | 3.5 | 141 | 0.1 | 1233 |

Another observation is the performance gap between image-to-text and text-to-image tasks across categories. Most models perform better on image-to-text retrieval, particularly in the medium and hard categories, where visual cues might more directly match captions. However, for text-to-image retrieval, the higher medR scores across categories suggest that models often rank incorrect images higher than the target image, indicating a potential gap in understanding the semantic connections from text to visuals.

### 5.3 Image-Text Binary Classification Evaluation

We evaluated the binary classification performance of SOTA models on IMP using the AUPRC metric, which measures how effectively models distinguish between matching and non-matching pairs across the easy, medium, and hard subsets. As shown in Table 5, models achieve near-perfect scores on the easy subset but show a significant drop in performance on the hard subset. For example, EVA-02 achieves an AUPRC of 1.00 on the easy subset but only 0.68 on the hard subset, demonstrating the challenge of handling semantically diverse captions.

An important observation is the variability in performance across models on the hard subset. While models often perform similarly on the easy and medium subsets, their scores diverge significantly on harder captions. For instance, while both EVA-02 and ImageBind achieve identical scores (1.00) on the easy subset and nearly identical scores (0.97) on the medium subset, EVA-02 outperforms ImageBind on the hard subset, scoring 0.68 compared to 0.65. This divergence highlights the limitations of current VLMs in generalizing to complex, polysemic image-caption pairs.

The drop in AUPRC across categories, combined with the increased divergence in scores on the hard subset, reflects the difficulty of handling semantically diverse captions. Figure 9 provides qualitative examples illustrating these challenges. For instance, the caption "Nature's painting with water on a canvas of rocks" is correctly matched by ConvNeXt but not by EVA02, while the caption "Sunset captured on a single branch"

Table 5: **Binary image-text classification performance on IMP for SOTA models per category.** The best result within the hard category is highlighted with **bold**.

| Method | Easy | Medium | Hard |
|---|---|---|---|
| CLIP-ViT-L/14 | 1.00 | 0.96 | 0.63 |
| AltCLIP | 1.00 | 0.97 | 0.66 |
| ALIGN | 0.99 | 0.96 | 0.66 |
| ConvNeXt | 0.99 | 0.96 | 0.62 |
| GroupViT | 0.97 | 0.93 | 0.60 |
| Siglip | 0.99 | 0.96 | 0.66 |
| ALBEF | 0.73 | 0.80 | 0.52 |
| BLIP | 0.94 | 0.94 | 0.62 |
| FLAVA | 0.98 | 0.95 | 0.67 |
| COCA | 1.00 | 0.97 | 0.65 |
| BLIP2-g | 0.99 | 0.96 | 0.64 |
| BLIP2-ViT-L | 0.99 | 0.96 | 0.64 |
| BLIP2-g-COCO | 0.98 | 0.95 | 0.63 |
| EVA-CLIP | 1.00 | 0.97 | 0.65 |
| EVA02 | 1.00 | 0.97 | **0.68** |
| Imagebind | 1.00 | 0.97 | 0.65 |

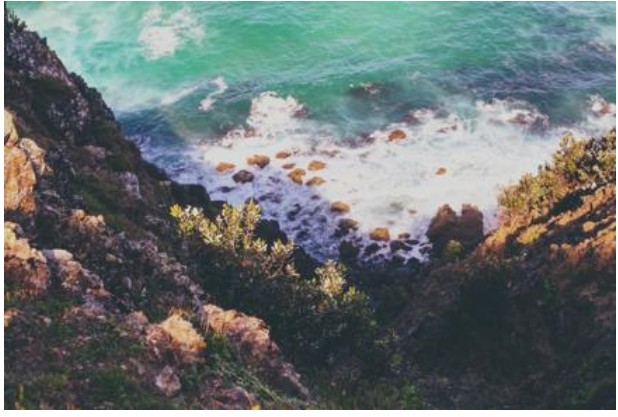 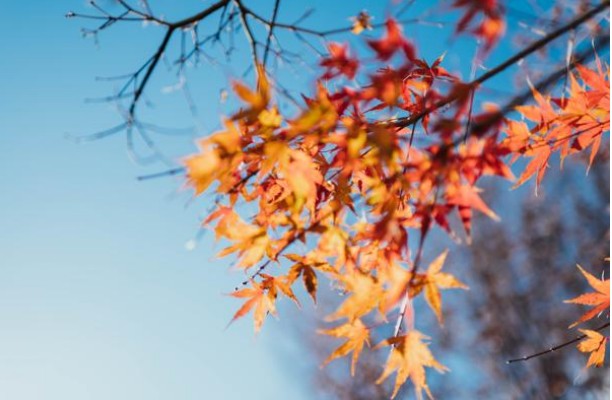

(a) Nature's painting with water on a canvas of rocks. ConvNeXt: correct, EVA02: incorrect.

(b) Sunset captured on a single branch. ConvNeXt: incorrect, EVA02: correct.

Figure 9: **Examples of image-text pairs in the IMP hard subset with model outputs.**

is correctly matched by EVA02 but not by ConvNeXt. These examples demonstrate the variation we see between models in which image-text pairs they match.

Additionally, to evaluate the impact of scale we benchmark CLIP across different model and pre-training dataset sizes. We use six datasets of different scales, CC12M (Changpinyo et al., 2021), YFCC15M (15M subset of YFCC100M (Thomee et al., 2016)), CLIP400M (Radford et al., 2021), Commonpool (Gadre et al., 2023), DataComp (Gadre et al., 2023), and LAION (Schuhmann et al., 2022). Commonpool and DataComp datasets each have multiple size subsets: for Commonpool, $s$, $m$, $l$, and $xl$ correspond to dataset sizes of 12.8M, 128M, 1.28B, and 12.8B pairs, respectively; for DataComp, $s$, $m$, $l$, and $xl$ correspond to 1.4M, 14M, 140M, and 1B pairs. Across these datasets, models of varying sizes (expressed in parameter counts) were evaluated. The results are presented in Table 6.

On each dataset, larger models consistently outperform smaller ones, demonstrating the impact of model size and capacity. For example, on CLIP400M, the AUPRC on the hard subset increases from 0.58 for RN50 to 0.60 for ViT-B/32 and further to 0.63 for ViT-L/14. A similar pattern is observed on DataComp-$xl$,

Table 6: **Binary image-text classification performance on IMP for SOTA models per category.**

| Method | # Params | Easy | Medium | Hard | Method | # Params | Easy | Medium | Hard |
|---|---|---|---|---|---|---|---|---|---|
| **YFCC15M** (Thomee et al., 2016) | | | | | **CC12M** (Changpinyo et al., 2021) | | | | |
| RN50 | 102M | 0.95 | 0.91 | 0.59 | RN50 | 102M | 0.96 | 0.92 | 0.58 |
| RN101 | 102M | 0.96 | 0.92 | 0.60 | **DataComp** Gadre et al. (2023) | | | | |
| **CLIP400M** (Radford et al., 2021) | | | | | ViT-B/32-*s* | 150M | 0.81 | 0.83 | 0.52 |
| RN50 | 102M | 1.00 | 0.96 | 0.58 | ViT-B/32-*m* | 150M | 0.95 | 0.91 | 0.59 |
| RN101 | 120M | 1.00 | 0.97 | 0.55 | ViT-B/32-*xl* | 150M | 1.00 | 0.96 | 0.63 |
| ViT-B/32 | 150M | 1.00 | 0.96 | 0.60 | ViT-B/16-*l* | 150M | 0.99 | 0.95 | 0.62 |
| ViT-B/16 | 150M | 1.00 | 0.97 | 0.60 | ViT-B/16-*xl* | 150M | 1.00 | 0.96 | 0.64 |
| ViT-L/14 | 428M | 1.00 | 0.96 | 0.63 | ViT-L/14-*xl* | 428M | 1.00 | 0.97 | 0.65 |
| ViT-L/14-336 | 428M | 1.00 | 0.97 | 0.64 | **LAION** Schuhmann et al. (2022) | | | | |
| **Commonpool** Gadre et al. (2023) | | | | | ViT-B/32-400*M* | 150M | 0.99 | 0.95 | 0.61 |
| | | | | | ViT-B/32-2*B* | 150M | 0.99 | 0.96 | 0.63 |
| ViT-B/32-*s* | 150M | 0.86 | 0.86 | 0.53 | ViT-B/16-400*M* | 150M | 0.99 | 0.96 | 0.61 |
| ViT-B/32-*m* | 150M | 0.96 | 0.91 | 0.58 | ViT-B/16-2*B* | 150M | 0.99 | 0.96 | 0.63 |
| ViT-B/16-*l* | 150M | 1.00 | 0.95 | 0.61 | ViT-L/14-400*M* | 428M | 0.99 | 0.96 | 0.63 |
| ViT-L/14-*xl* | 428M | 1.00 | 0.97 | 0.65 | ViT-L/14-2*B* | 428M | 0.99 | 0.97 | 0.64 |

where ViT-B/32 achieves 0.63, while ViT-L/14 improves to 0.65. These results indicate that larger models are generally better at capturing semantic nuances, though the improvements are modest, particularly for challenging subsets like IMP hard.

The performance of ViT-B/32 varies across the four datasets, highlighting the influence of dataset scale and curation. On the hard subset, ViT-B/32 achieves an AUPRC of 0.58 on on Commonpool-*m*, improving slightly to 0.60 on CLIP400M. In contrast, it scores 0.63 on DataComp-*xl*, which benefits from stricter filtering applied to a curated subset of Commonpool, emphasizing the importance of dataset quality. These results suggest that dataset curation plays a critical role in improving performance on semantically diverse captions, and increasing dataset size alone is insufficient for consistent improvements.

### 5.4 Summary

Overall, our evaluations highlight the significant challenges posed by polysemy in vision-language tasks, with performance consistently dropping on the hard subset compared to the easy and medium subsets. Dataset quality emerged as a more critical factor than size, as curated datasets with better filtering and annotations consistently outperformed larger but less curated ones. Similarly, larger models generally achieved better results than smaller ones, but the improvements diminished on the hard subset, indicating that scaling alone, whether in terms of dataset size or model parameters, is insufficient to address the challenges of semantic diversity. These findings suggest that addressing polysemy effectively requires not just scale, but also improved training strategies and architectures designed to better capture semantic nuances.

## 6 Discussion

### 6.1 Underlying Reasons for VLM struggles with Image Polysemy

Our analysis shows that VLMs struggle with image polysemy, we suspect this is due to the fundamental assumptions embedded in their training objective (Radford et al., 2021; Chen et al., 2020). Standard vision-language contrastive losses implicitly assume semantic similarity among all captions paired with a single image, encouraging embeddings for these captions to cluster closely. This assumption generally holds true for descriptive datasets such as MS-COCO, where captions predominantly focus on observable visual elements. However, image polysemy breaks this assumption, as a single image (or visually similar image set) may

be validly described by multiple captions with significantly different semantics. Under ideal conditions, embeddings for these captions would be distant to reflect semantic differences, yet simultaneously close due to their visual association.

The standard contrastive loss cannot resolve this contradiction optimally. Instead, VLMs appear to favor semantic coherence at the cost of embedding misalignment, effectively failing to align certain valid polysemic captions correctly to their corresponding images. The IMP benchmark explicitly illustrates this phenomenon and offers a rigorous basis for future exploration into polysemy.

## 6.2 Importance of Addressing Polysemy

Image polysemy is ubiquitous in natural human communication, yet it remains significantly understudied in vision-language research. Addressing polysemy explicitly is crucial not only because of potential performance gains in downstream tasks, but also—and perhaps primarily—for the development and evaluation of better training datasets. Current large-scale vision-language dataset creation methods (e.g., DataComp (Gadre et al., 2023), LAION (Schuhmann et al., 2022)) heavily rely on filtering web-scale data using CLIP-based similarity thresholds.

However, our analysis reveals critical limitations in this approach: as demonstrated by Gadre et al. (2023), adjusting the CLIP similarity threshold from 0.129 (discarding only 10% of pairs) to 0.384 (discarding 99% of pairs) occurs within an extremely narrow similarity range, causing many genuinely polysemic and semantically valid captions to be excluded unintentionally. Even at the most permissive threshold, annotator-validated polysemic cases risk exclusion due to their inherently lower cross-modal similarity scores (e.g., Figure 4). Thus, current filtering mechanisms inadvertently limit the dataset's diversity and richness, significantly constraining the potential for larger models to naturally acquire the ability to handle semantic diversity effectively, despite scaling to larger dataset sizes.

## 6.3 Potential Directions for Improving Model Handling of Polysemy

Given the limitations exposed by IMP, we propose several promising directions for future technical contributions that explicitly address polysemy: (1) Polysemy-aware Training Objectives: Develop loss functions specifically designed to handle multiple semantically valid positives, potentially using probabilistic embeddings or multi-instance embedding approaches. (2) Polysemy-aware Model Architectures: Extend traditional contrastive models to explicitly represent and distinguish multiple plausible semantic alignments simultaneously, effectively modeling rich semantic diversity. (3) Novel Retrieval and Classification Metrics: Design evaluation frameworks capable of explicitly addressing multiple valid image-caption alignments, going beyond traditional single-positive retrieval metrics. (4) Data filtering and Augmentation Strategies: Develop more sophisticated filtering mechanisms that can effectively capture polysemic cases, potentially leveraging human-in-the-loop approaches to ensure dataset validity. These directions all represent potential avenues for improvements in handling polysemy, and where IMP may contribute as a benchmark.

## 7 Conclusion

This paper highlights the challenges posed by image polysemy in vision-language learning, in particular, in how to align a single image to multiple possible interpretations. We proposed a clustering-based approach to identify polysemy in web-scale datasets and explored the limitations of traditional metrics for measuring polysemy. We introduced IMP, a novel benchmark dataset designed to evaluate the performance of VLMs on semantically diverse image-caption pairs. Our comprehensive benchmarking reveals that while state-of-the-art VLMs excel at handling straightforward image-text pairs, their performance deteriorates when increasing semantic complexity. Our findings emphasize the need for new approaches that better account for polysemy in vision-language tasks. By presenting a challenging yet systematic evaluation framework, this work aims to guide future research toward building robust VLMs that well handle polysemy.

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

## A   Appendix

## B   Ethical Impact Statement

An underlying ambition of exploring, and hopefully incorporating, polysemy in vision-language representation learning is to increase the potential for capturing underrepresented views and perspectives. This may have the positive effect of increasing cultural and geographic diversity, however, it may also unearth potentially undesirable views. Yet, we would argue that these views are already in the data, and may now imply be affecting learned representations in unknown ways - and potentially leading to bias in downstream tasks.

Additionally, there are a number of potential (long term) ethical impacts related to polysemy.

1. **Global cultural diversity**: We aim to capture a diverse set of captions, however, the dataset may still be biased towards certain cultures. For example, sunset images might be consider more as romantic in some cultures, but tragic in others. Full image polysemy requires greater geographic spread of the data.

2. **Misinformation**: Better understanding of the polysemous nature of images may be exploited to create or spread misinformation. For example, visual memes, can carry subtextual meanings which are not immediately apparent, but can lead to harmful political or social implications.

With regards to the dataset, we use images sampled from the Unsplash Lite dataset, which is granted by Unsplash a non-exclusive, non-transferable, non-sublicensable license to download and store any photos, images, or other data contained in the Lite Dataset, and internally use the commercial licenced data for research purposes. We do not change the image in the dataset, and will not publish any portion of the licensed data. We follow all terms and conditions of the Unsplash License.

The captions gathered from web-curation are from websites that are publicly accessible, and we use only the website titles as the curated captions. We do not store any of the website content.

All participants for the human evaluation were recruited on a voluntary basis and compensated in the form of chocolate and candy on completion of the task.

## C   IMP

### C.1   Image Source

We select the Unsplash Lite Dataset as the image source, which consists of around 25k images. The dataset is licensed under the Unsplash License, which allows us to use the images for free, including both commercial and non-commercial purposes. The dataset is available at `https://unsplash.com/data`.

We sample a subset of 5k images from the Unsplash Lite Dataset, by first extract the visual embeddings of the images using DINOv2 (Oquab et al., 2024) and project them into a 2D space using UMAP (McInnes et al., 2020). We then select top 5k images by farthest point sampling in the UMAP space. The result can be visualized in Figure 10. The sampling prevents the dataset from containing very similar images which might potentially lead to high correlation between non-paired images and captions.

### C.2   Caption Source

There are two sources where captions are gathered, (1) through clustering in the Conceptual Datasets, and (2) web-curation. We utilize the full Conceptual 3M (Sharma et al., 2018) and a 7M subset of Conceptual 12M (Changpinyo et al., 2021) to perform the clustering-based caption gathering. By manually inspecting the clustering results, we use a radius size of around 50 (MSCOCO), which equals to a clutser size between 800 to 3000 in the 3+7 M Conceptual dataset, depending on the image similarity threshold. Since the way captions being used might vary from time, we also use the Google Vision API to search web-entities, which will return a list of websites containing identical or visually similar images. We then use website titles as the curated captions.

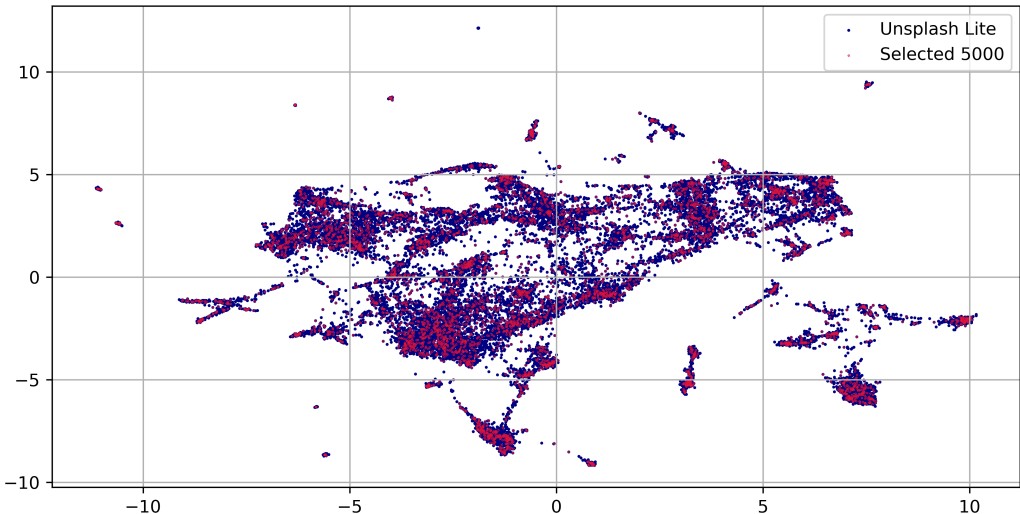

Figure 10: **A sample of 5k images from the Unsplash Lite Dataset**.

### C.3 Caption Cleaning

To ensure the quality of the captions, we perform a series of cleaning steps. We clean captions gathered from source (2), which are often noisy, containing HTML tags, website names as suffixes or prefixes, hashtags in website titles, a high rate of word repetition, and other irrelevant information. We use packages such as ftfy to first clean those noise, and use token analysis to remove captions with high word repetition rate, which are likely to be a list of keywords curated from internet. We also remove captions with no noun, verb or adjective (Sharma et al., 2018). The captions are cleaned by removing most of these noises while keeping the sentences fluent and readable. Only caption which has at least 5 words are kept. The cleaned captions along with captions from source (1) are merged into the candidate caption pool. To reduce the number of captions to be annotated, we remove (nearly) duplicate captions by n-grams of each caption pool.

### C.4 Annotation

Annotators are tasked with classifying each candidate caption as "good" , "bad", or "unsure" based on detailed guidelines to maximally capture the diversity of captions while ensuring minimal subjectivity. The quality assurance process is conducted in two stages, interannotator agreement of which the same set of images is annotated by multiple annotators, and review-feedback loop of which an initial set of annotations is reviewed by head annotator and provide feedback to other annotators. The inter-annotator agreement stage is used to ensure that the annotators are consistent in their judgments. The review-feedback loop is used to ensure that the annotators were following the guidelines and to provide feedback on their annotations.

Parallel to the binary classification, a list of changes are made to improve the quality of captions. The changes include but are not limited to:

1. **Color**: Change the color of the object in the caption to the actual color of the object in the image, for example, "a blue flower" to "a red flower".

2. **Number**: Change the wrong number of objects in the caption to the actual number of objects in the image, for example, "a group of birds" to "a single bird".

3. **Object Name**: Image objects such as animals and plants are often misidentified in the captions due to the nature of image clustering. We correct the object name to the actual object in the image

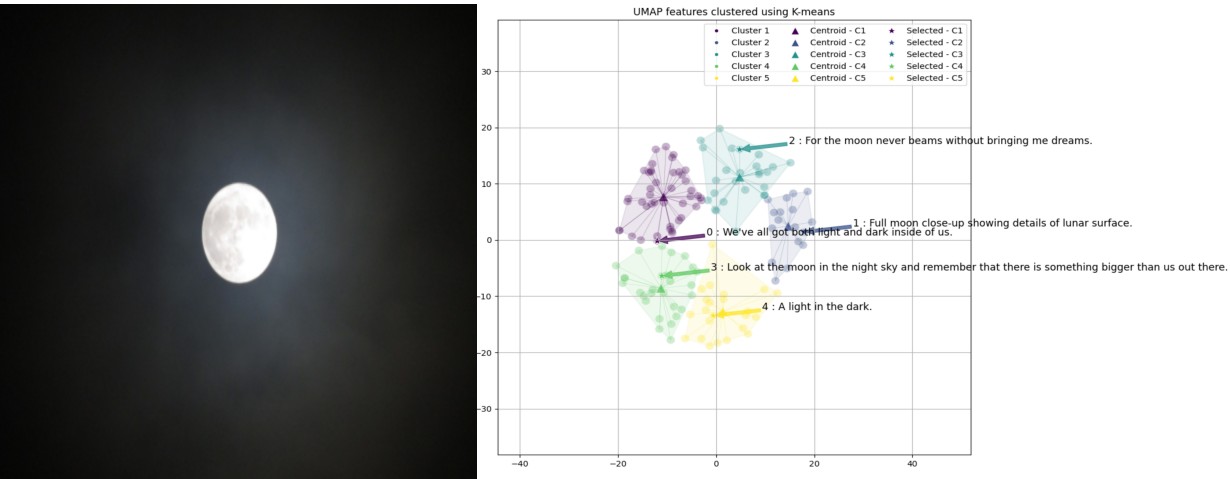

Figure 11: **An exampke of using SentenceBERT, UMAP, and K-means to select diverse captions from the candidate pool**.

and allows certain level of generalization, for example, we keep "elk" which is the correct species of the animal in the image, but change "Moose" to "deer".

4. **Person**: We use <person> placeholder to replace unclear or irrelevant person names in the caption.

5. **Location**: We replace the detailed location in the caption to a more general location, for example, "New York City" will be kept, but a specific building name will or replace by "a building".

To further improve the quality of the dataset, we remove captions which are too generic, such as "beautiful scene makes me happy". Such captions are often regarded as "good" by the annotators, but can also be good captions for many other images in the dataset.

## C.5 Diverse Caption Selection

After the manual selection process, we have a pool of good captions for each image. Any of these captions can be included in the final dataset. To maximize diversity, we employ SentenceBERT (Reimers & Gurevych, 2019) to extract textual features from the captions and project the textual features into 2D space using UMAP (McInnes et al., 2020). We then perform K-means clustering with $K = 5$ to group caption into five clusters. One caption is randomly selected from each cluster to create a diverse set of five captions for each image. An example is shown in Figure 11.

## C.6 Data Leakage

We ensure that there is no overlap (identical image) between samples from Unsplash and the 3+7 M Conceptual dataset. We cannot verify this when consider larger datasets, such as CLIP400M (Radford et al., 2021) or DataComp1B (Gadre et al., 2023), but the impact of data leakage is reduced by the way we gather and annotate captions.

## D Experimental Details

All models evaluated in this work are implemented with PyTorch (Paszke et al., 2019). We perform the zero-shot evaluation on most state-of-the-art models by using the pre-trained weights provided by Huggingface (Wolf et al., 2020) and OpenClip (Ilharco et al., 2021). The evaluation on ImageBind is done by using the original codebase (Girdhar et al., 2023).

Evaluation on one-to-many approaches, namely, PVSE (Song & Soleymani, 2019), PCME (Chun et al., 2021), and DivE (Kim et al., 2023), is done by using the official codebase provided by the authors. We use the modules that are relevant to each model as the adapter, with the CLIP ViT-B/32 pre-trained on CLIP400M (Radford et al., 2021) as the backbone (Wolf et al., 2020). We also use the training loss and evaluation metrics from the original codebase for each model.

We use the same hyperparameters (except for learning rate) and setup following previous work (Radford et al., 2021; Dong et al., 2022). We use 1e-4 as the base-learning rate, which is selected by grid sear ch ranging from 1e-3 to 1e-6. We adopt the default hyperparameters for LoRA (Hu et al., 2021; Sourab Mangrulkar et al., 2022). All models are trained for 6 epochs with batch size 1024 and cosineannealing learning rate scheduler. We follow the DivE setting (Kim et al., 2023), and the learning rates for both image encoder and text encoder are scaled with 0.05 and 0.5, respectively.

All experiments are done with two NVIDIA A6000 PCIe GPUs with 48GB memory.

# E   Additional Experimental Results

## E.1   Additional Results on One-to-Many Models

Table 7: **One-to-many models as adapter added to CLIP ViT-B/32 with trainable LoRA parameters, evaluated on IMP**. The best results within each recall are highlighted with **bold**. 100K, 500K, and 5M refer to the number of image-text pairs in the pre-training dataset.

| Method | | Image-to-Text | | | | | | Text-to-Image | | | | | |
|---|---|---|---|---|---|---|---|---|---|---|---|---|---|
| | | R@1 | R@5 | R@10 | meanR | medR | mAP@R | R@1 | R@5 | R@10 | meanR | medR | mAP@R |
| CLIP | base | 14.7 | 29.1 | 37.8 | 2084 | 566 | 7.341 | 6.0 | 14.1 | 19.3 | 515 | 121 | 10.611 |
| 100K | | | | | | | | | | | | | |
| PVSE | base | 13.8 | 29.5 | 37.8 | 2082 | 579 | 6.920 | 5.9 | 14.3 | 20.0 | 510 | 127 | 10.618 |
| PVSE | k=1 | 13.9 | 29.5 | 38.2 | 2073 | 583 | 6.900 | 5.8 | 14.3 | 19.8 | 509 | 125 | 10.556 |
| PVSE | k=2 | 13.9 | 29.3 | 38.0 | 2051 | 581 | 6.880 | 5.8 | 14.2 | 19.8 | 502 | 123 | 10.524 |
| PVSE | k=5 | 13.9 | 29.4 | 37.8 | 2037 | 545 | 6.880 | 5.8 | 14.2 | 19.8 | 502 | 121 | 10.532 |
| PCME | base | 13.9 | 29.3 | 37.9 | 2072 | 577 | 6.875 | 5.8 | 14.2 | 19.7 | 505 | 129 | 10.538 |
| DivE | k=2 | 14.4 | 29.6 | 37.8 | 2043 | 548 | 7.000 | 5.9 | 14.4 | 19.9 | 533 | 134 | 10.630 |
| DivE | k=5 | 14.4 | 29.6 | 37.9 | 2022 | 552 | 6.919 | 6.0 | 14.3 | 20.0 | 541 | 136 | 10.607 |
| 500K | | | | | | | | | | | | | |
| PVSE | base | 14.9 | 29.7 | 38.0 | 2061 | 561 | 7.359 | 6.0 | 14.3 | 19.6 | 512 | 123 | 10.624 |
| PVSE | $k=1$ | 15.0 | 29.5 | 38.5 | 2053 | 562 | 7.363 | **6.1** | 14.3 | 19.8 | 514 | 123 | 10.612 |
| PVSE | $k=2$ | 15.0 | 29.5 | 38.3 | 2044 | 560 | 7.369 | 6.0 | 14.3 | 19.7 | 512 | 125 | 10.627 |
| PVSE | $k=5$ | 15.0 | 29.7 | 38.3 | 2037 | 558 | 7.380 | **6.1** | 14.3 | 19.9 | 509 | **119** | 10.712 |
| PCME | base | 15.0 | 29.6 | 39.0 | 2041 | 556 | 7.385 | 5.8 | 14.3 | 19.7 | 511 | 121 | 10.628 |
| DivE | $k=2$ | **15.4** | 30.1 | 38.5 | 2002 | **540** | **7.504** | 5.9 | **14.8** | **20.5** | 459 | 130 | **10.748** |
| DivE | $k=5$ | **15.4** | 30.0 | 38.5 | 1992 | 542 | 7.500 | 5.9 | **14.8** | **20.5** | **458** | 131 | 10.746 |
| 5M | | | | | | | | | | | | | |
| PVSE | base | 14.1 | 28.1 | 36.8 | 2072 | 555 | 6.737 | 5.8 | 14.0 | 19.3 | 538 | 165 | 10.365 |
| PVSE | k=1 | 14.2 | 28.4 | 37.5 | 2018 | 564 | 6.850 | 5.9 | 14.1 | 19.6 | 533 | 150 | 10.457 |
| PVSE | k=2 | 14.1 | 28.9 | 37.9 | 1989 | 568 | 6.877 | 5.9 | 14.2 | 19.6 | 528 | 131 | 10.516 |
| PVSE | k=5 | 14.1 | 29.1 | 38.0 | 1970 | 558 | 6.895 | 6.0 | 14.2 | 19.7 | 528 | 131 | 10.549 |
| PCME | base | 14.1 | 29.0 | 38.0 | 1968 | 578 | 6.914 | 5.7 | 14.0 | 19.5 | 526 | 127 | 10.550 |
| DivE | k=2 | 14.7 | 29.6 | 38.6 | 1967 | 575 | 6.916 | 5.6 | 14.5 | 20.0 | 526 | 125 | 10.553 |
| DivE | k=5 | 14.7 | **30.2** | **39.1** | **1952** | 570 | 6.950 | 5.6 | 14.5 | 20.3 | 522 | 123 | 10.658 |

We report the additional results on fine-tuning the one-to-many models with different pre-training data size in Table 7. We observe that with 500k pre-training data, the performance of all three one-to-many

Table 8: **One-to-many models as adapter added to CLIP ViT-B/32 with trainable LoRA parameters, evaluated on IMP**. The best results within each recall are highlighted with **bold**. Scores lower than CLIP are highlighted in red.

| Method | | Image-to-Text | | | | | | Text-to-Image | | | | | |
|---|---|---|---|---|---|---|---|---|---|---|---|---|---|
| | | R@1 | R@5 | R@10 | meanR | medR | mAP | R@1 | R@5 | R@10 | meanR | medR | mAP |
| CLIP | base | 14.7 | 29.1 | 37.8 | 2084 | 566 | 7.341 | 6.0 | 14.1 | 19.3 | 515 | 121 | 10.611 |
| PVSE | base | 14.9 | 29.7 | 38.0 | 2061 | 561 | 7.359 | 6.0 | 14.3 | 19.6 | 512 | 123 | 10.624 |
| PVSE | $k$=1 | 15.0 | 29.5 | 38.5 | 2053 | 562 | 7.363 | **6.1** | 14.3 | 19.8 | 514 | 123 | 10.612 |
| PVSE | $k$=2 | 15.0 | 29.5 | 38.3 | 2044 | 560 | 7.369 | 6.0 | 14.3 | 19.7 | 512 | 125 | 10.627 |
| PVSE | $k$=5 | 15.0 | 29.7 | 38.3 | 2037 | 558 | 7.380 | **6.1** | 14.3 | 19.9 | 509 | **119** | 10.712 |
| PCME | base | 15.0 | 29.6 | **39.0** | 2041 | 556 | 7.385 | 5.8 | 14.3 | 19.7 | 511 | 121 | 10.628 |
| DivE | $k$=2 | **15.4** | **30.1** | 38.5 | 2002 | **540** | **7.504** | 5.9 | **14.8** | **20.5** | 459 | 130 | **10.748** |
| DivE | $k$=5 | **15.4** | 30.0 | 38.5 | **1992** | 542 | 7.500 | 5.9 | **14.8** | **20.5** | **458** | 131 | 10.746 |

models improves over zero-shot CLIP. When the data size drops to 100k, there is a clear gap between the performance of these models and CLIP on the R@1 and mAP@R scores. We notice that with the same data size, the meanR and medR scores decrease consistantly with the increase of $k$ for PVSE and DivE. Same trend can be seen when the data size increases to 5M, where DivE with $k = 5$ achieves the best R@5 and R@10 and lowest meanR. However, we also observe that DivE has a performance drop on text-to-image meanR and medR after trained on either 100K or 5M data.

## E.2 Additional Results on Model Scales and Dataset Sizes

We report the additional results on the performance of VLM with model variants and dataset sizes in Table 9.

Table 9: **Performance for VLM with model variants and dataset size**. Where $s, m, l$ after model names are scale factors for the corresponding dataset size.

| Method | variant | Image-to-Text | | | | | | Text-to-Image | | | | | |
| | | R@1 | R@5 | R@10 | meanR | medR | mAP@R | R@1 | R@5 | R@10 | meanR | medR | mAP@R |
|---|---|---|---|---|---|---|---|---|---|---|---|---|---|
| **CC12M** Changpinyo et al. (2021) | | | | | | | | | | | | | |
| RN50 | quickgelu | 7.4 | 17.4 | 24.5 | 4150 | 1609 | 3.609 | 3.3 | 8.5 | 12.1 | 849 | 300 | 6.454 |
| **YFCC15M** Thomee et al. (2016) | | | | | | | | | | | | | |
| RN50 | quickgelu | 5.1 | 14.2 | 19.9 | 4475 | 1842 | 2.823 | 2.8 | 7.6 | 11.4 | 841 | 319 | 5.747 |
| RN101 | quickgelu | 5.6 | 15.3 | 21.5 | 4333 | 1732 | 3.086 | 2.9 | 7.9 | 11.6 | 825 | 302 | 6.023 |
| **CLIP400M** Radford et al. (2021) | | | | | | | | | | | | | |
| RN50 | quickgelu | 14.8 | 29.5 | 37.2 | 2198 | 595 | 7.197 | 5.4 | 13.2 | 18.4 | 537 | 135 | 9.929 |
| RN50 | x4 | 14.8 | 30.1 | 37.9 | 2095 | 546 | 7.435 | 6.0 | 14.2 | 19.3 | 529 | 127 | 10.707 |
| RN50 | x16 | 15.5 | 31.1 | 39.8 | 2073 | 533 | 7.746 | 6.4 | 14.9 | 20.4 | 508 | 118 | 11.240 |
| RN50 | x64 | 16.4 | 32.0 | 40.6 | 2056 | 517 | 8.087 | 7.0 | 15.6 | 20.9 | 487 | 106 | 11.826 |
| RN101 | quickgelu | 14.2 | 29.0 | 37.4 | 2160 | 574 | 7.150 | 5.4 | 13.3 | 18.6 | 546 | 134 | 10.013 |
| ViT-B-32 | quickgelu | 14.7 | 29.1 | 37.8 | 2085 | 566 | 7.342 | 6.0 | 14.1 | 19.3 | 514 | 121 | 10.612 |
| **Commonpool** Gadre et al. (2023) | | | | | | | | | | | | | |
| ViT-B-32-$s$ | clip | 0.7 | 3.6 | 5.6 | 7145 | 5098 | 0.672 | 0.4 | 1.5 | 2.5 | 1410 | 972 | 1.329 |
| ViT-B-32-$s$ | image | 0.5 | 1.9 | 3.3 | 8570 | 6795 | 0.417 | 0.3 | 1.0 | 1.7 | 1676 | 1301 | 0.952 |
| ViT-B-32-$s$ | text | 0.8 | 3.1 | 5.4 | 7172 | 5125 | 0.677 | 0.3 | 1.3 | 2.3 | 1435 | 989 | 1.206 |
| ViT-B-32-$s$ | basic | 0.6 | 2.3 | 4.3 | 7185 | 5165 | 0.544 | 0.2 | 1.2 | 2.2 | 1453 | 1034 | 1.117 |
| ViT-B-32-$m$ | clip | 6.2 | 15.7 | 21.7 | 4253 | 1851 | 3.119 | 2.0 | 6.0 | 8.9 | 894 | 377 | 4.562 |
| ViT-B-32-$m$ | image | 6.6 | 17.1 | 23.9 | 3972 | 1647 | 3.28 | 2.1 | 6.5 | 9.5 | 861 | 332 | 4.831 |
| ViT-B-32-$m$ | text | 6.9 | 16.6 | 23.2 | 4218 | 1880 | 3.189 | 1.9 | 5.8 | 8.8 | 923 | 386 | 4.457 |
| ViT-B-32-$m$ | basic | 6.0 | 15.9 | 22.2 | 4113 | 1748 | 2.992 | 1.8 | 5.5 | 8.7 | 878 | 350 | 4.360 |
| ViT-B-16-$l$ | clip | 14.3 | 28.8 | 37.8 | 2790 | 835 | 6.500 | 4.7 | 11.9 | 16.5 | 636 | 179 | 8.840 |
| ViT-B-16-$l$ | image | 15.2 | 30.9 | 39.6 | 2495 | 678 | 6.958 | 5.2 | 13.1 | 18.1 | 597 | 151 | 9.670 |
| ViT-B-16-$l$ | text | 14.4 | 29.5 | 37.9 | 2735 | 786 | 6.431 | 4.7 | 11.8 | 16.2 | 666 | 182 | 8.796 |
| ViT-B-16-$l$ | basic | 13.1 | 27.5 | 35.8 | 2642 | 758 | 6.252 | 4.8 | 12.2 | 17.0 | 631 | 166 | 9.085 |
| **DataComp** Gadre et al. (2023) | | | | | | | | | | | | | |
| ViT-L-14 | CLIPA | 19.8 | 37.9 | 46.3 | 2138 | 498 | 8.991 | 7.3 | 16.6 | 22.3 | 505 | 106 | 12.478 |
| ViT-L-14-336 | CLIPA | 19.7 | 37.4 | 46.5 | 2120 | 486 | 9.039 | 7.5 | 17.1 | 22.6 | 501 | 105 | 12.681 |
| ViT-H-14 | CLIPA | 20.2 | 38.1 | 47.7 | 1935 | 431 | 9.509 | 7.9 | 17.9 | 23.6 | 467 | 93 | 13.285 |
| ViT-H-14-336 | CLIPA | 21.1 | 38.9 | 48.1 | 1897 | 428 | 9.669 | 8.2 | 18.1 | 23.9 | 457 | 89 | 13.629 |
| ViT-H-14-336 | CLIPA | 20.2 | 37.9 | 47.6 | 1922 | 421 | 9.556 | 8.0 | 18.0 | 23.7 | 464 | 91 | 13.398 |
| ViT-bigG-14 | CLIPA | 21.5 | 38.3 | 47.8 | 1845 | 396 | 9.978 | 8.3 | 18.3 | 24.0 | 467 | 91 | 13.671 |
| ViT-bigG-14-336 | CLIPA | 20.9 | 38.8 | 48.8 | 1830 | 389 | 10.018 | 8.5 | 18.5 | 24.4 | 467 | 90 | 13.931 |

Table 10: **Performance for VLM with increasing parameter size and across varying dataset size**. The best results within each column are highlighted with **bold**. Where $s, m, l, xl$ after model names are scale factors for the corresponding dataset size.

| Method | Para | Image-to-Text | | | | | | Text-to-Image | | | | | |
| | | R@1 | R@5 | R@10 | meanR | medR | mAP@R | R@1 | R@5 | R@10 | meanR | medR | mAP@R |
|---|---|---|---|---|---|---|---|---|---|---|---|---|---|
| **CC12M** Changpinyo et al. (2021) | | | | | | | | | | | | | |
| RN50 | 102M | 7.6 | 18.3 | 24.9 | 4407 | 1746 | 3.670 | 3.1 | 8.1 | 11.6 | 879.3 | 320 | 6.152 |
| **YFCC15M** Thomee et al. (2016) | | | | | | | | | | | | | |
| RN50 | 102M | 6.1 | 15.1 | 21.1 | 4482 | 1814 | 3.125 | 2.5 | 7.3 | 10.9 | 853 | 331 | 5.519 |
| RN101 | 120M | 6.1 | 16.0 | 22.5 | 4285 | 1683 | 3.303 | 2.8 | 7.5 | 11.1 | 830 | 313 | 5.796 |
| **CLIP400M** Radford et al. (2021) | | | | | | | | | | | | | |
| RN50 | 102M | 14.8 | 29.4 | 37.2 | 2198 | 595 | 7.196 | 5.4 | 13.2 | 18.4 | 538 | 135 | 9.928 |
| RN101 | 120M | 14.2 | 29.0 | 37.4 | 2160 | 574 | 7.149 | 5.4 | 13.3 | 18.6 | 547 | 134 | 10.013 |
| ViT-B/32 | 150M | 14.7 | 29.1 | 37.8 | 2084 | 566 | 7.341 | 6.0 | 14.1 | 19.3 | 515 | 121 | 10.611 |
| ViT-B/16 | 150M | 15.3 | 30.1 | 38.6 | 2039 | 539 | 7.724 | 6.2 | 14.7 | 20.0 | 502 | 115 | 10.989 |
| ViT-L/14 | 428M | 15.8 | 30.9 | 38.8 | 2057 | 531 | 7.865 | 6.4 | 15.3 | 20.5 | 503 | 114 | 11.363 |
| ViT-L/14-336 | 428M | 17.0 | 32.8 | 41.8 | **1908** | 472 | 8.512 | 7.0 | 16.0 | 21.5 | 478 | 105 | 12.029 |
| **Commonpool** Gadre et al. (2023) | | | | | | | | | | | | | |
| ViT-B/32-$s$ | 150M | 0.8 | 3.6 | 5.6 | 7145 | 5099 | 0.673 | 0.4 | 1.5 | 2.5 | 1410 | 972 | 1.329 |
| ViT-B/32-$m$ | 150M | 6.3 | 15.8 | 21.7 | 4252 | 1851 | 3.125 | 2.0 | 6.0 | 8.9 | 895 | 377 | 4.562 |
| ViT-B/16-$l$ | 150M | 14.3 | 28.8 | 37.7 | 2789 | 836 | 6.498 | 4.7 | 11.9 | 16.5 | 636 | 179 | 8.837 |
| ViT-L/14-$xl$ | 428M | 18.8 | 36.0 | 44.8 | 2057 | 489 | 8.962 | 7.3 | 16.3 | 21.8 | 484 | 104 | 12.312 |
| **DataComp** Gadre et al. (2023) | | | | | | | | | | | | | |
| ViT-B/32-$s$ | 150M | 0.5 | 1.9 | 3.3 | 8571 | 6796 | 0.419 | 0.3 | 1.0 | 1.7 | 1677 | 1302 | 0.952 |
| ViT-B/32-$m$ | 150M | 6.1 | 14.3 | 20.0 | 4659 | 2227 | 2.840 | 1.9 | 5.6 | 8.3 | 970 | 438 | 4.308 |
| ViT-B/32-$xl$ | 150M | 17.8 | 34.2 | 42.9 | 2340 | 599 | 8.136 | 6.2 | 14.7 | 19.7 | 547 | 127 | 10.972 |
| ViT-B/16-$l$ | 150M | 15.2 | 30.0 | 38.6 | 2693 | 768 | 6.838 | 5.3 | 12.8 | 17.6 | 612 | 163 | 9.599 |
| ViT-B/16-$xl$ | 150M | 18.7 | 35.0 | 43.4 | 2249 | 562 | 8.499 | 6.7 | 15.6 | 21.2 | 525 | 117 | 11.698 |
| ViT-L/14-$xl$ | 428M | **19.7** | **37.0** | **45.6** | 1951 | **454** | **9.289** | **7.7** | **17.0** | **22.8** | **467** | **96** | 12.851 |

## F   Additional Qualitative Results

We report more qualitative examples of image-text pairs with low CLIP similarity but high human agreement in Figure 12 and Figure 13.

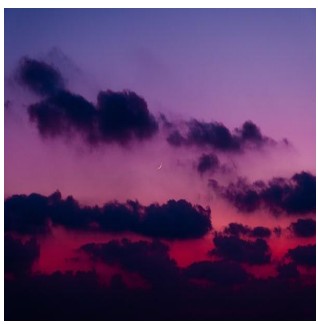

(a) The kind of day when coffee and a good book are mandatory.
**CLIP**:0.068, **Human**: 92.9%

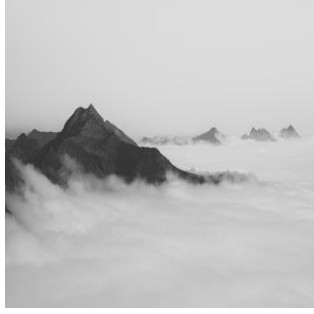

(b) Not all treasures glitter. some are cast in stone.
**CLIP**:0.080, **Human**: 89.5%

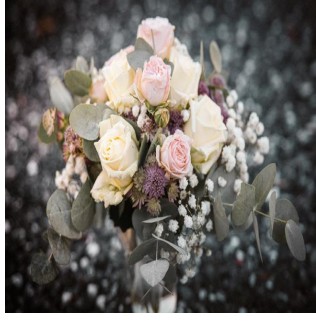

(c) In a room full of art, I would still stare at it.
**CLIP**:0.081, **Human**: 90.9%

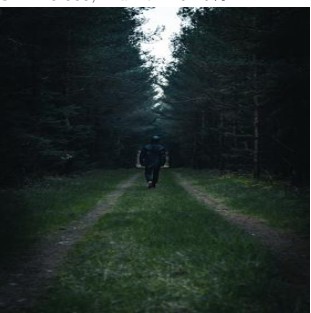

(d) Each step a brushstroke in the painting of their day.
**CLIP**:0.100, **Human**: 92.9%

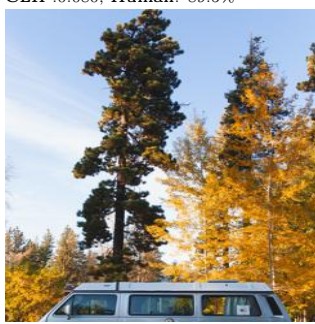

(e) The white knight rests before the next adventure.
**CLIP**:0.107, **Human**: 100%

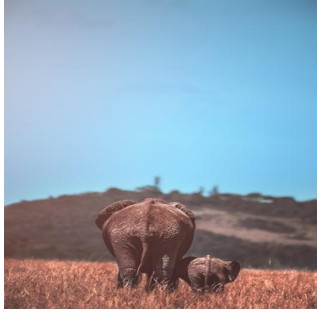

(f) A furry tide ebbs and flows across the green shore.
**CLIP**:0.102, **Human**: 100%

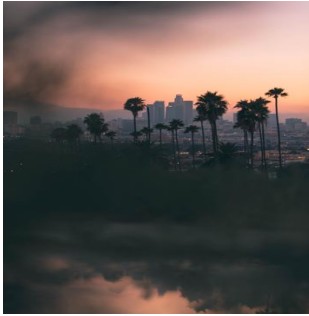

(g) The calm before the coconut's playful tumble.
**CLIP**:0.104, **Human**: 100%

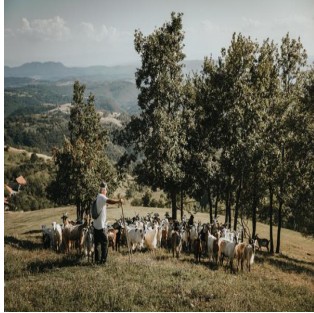

(h) A majestic stroll through nature's autumn tapestry.
**CLIP**:0.105, **Human**: 94.1%

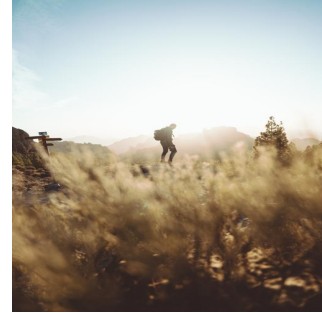

(i) The quiet before the storm of thoughts inside <person>'s head.
**CLIP**:0.106, **Human**: 85.7%

Figure 12: **Examples of false negative image-text pair**.

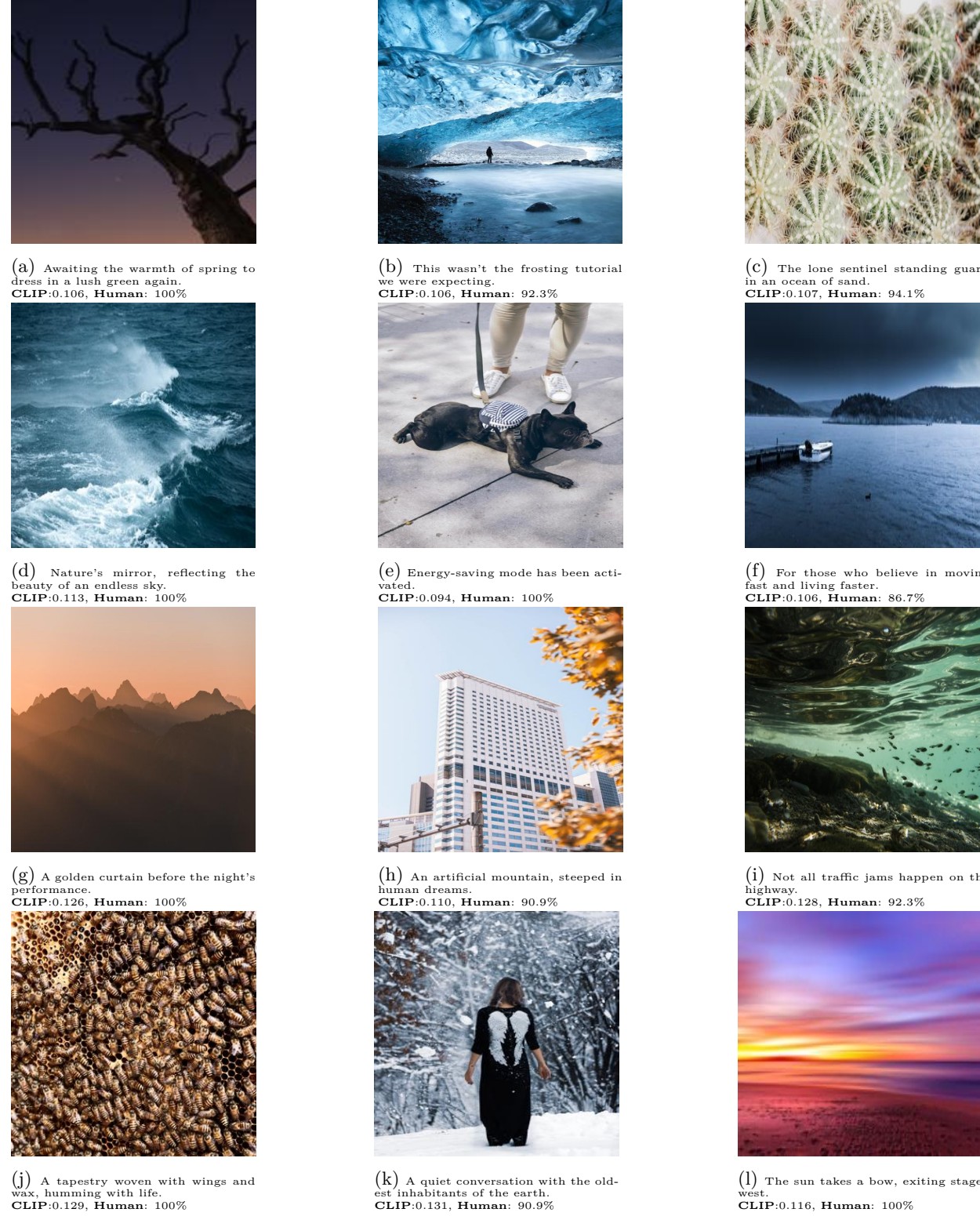

(a) Awaiting the warmth of spring to dress in a lush green again.
**CLIP**:0.106, **Human**: 100%

(b) This wasn't the frosting tutorial we were expecting.
**CLIP**:0.106, **Human**: 92.3%

(c) The lone sentinel standing guard in an ocean of sand.
**CLIP**:0.107, **Human**: 94.1%

(d) Nature's mirror, reflecting the beauty of an endless sky.
**CLIP**:0.113, **Human**: 100%

(e) Energy-saving mode has been activated.
**CLIP**:0.094, **Human**: 100%

(f) For those who believe in moving fast and living faster.
**CLIP**:0.106, **Human**: 86.7%

(g) A golden curtain before the night's performance.
**CLIP**:0.126, **Human**: 100%

(h) An artificial mountain, steeped in human dreams.
**CLIP**:0.110, **Human**: 90.9%

(i) Not all traffic jams happen on the highway.
**CLIP**:0.128, **Human**: 92.3%

(j) A tapestry woven with wings and wax, humming with life.
**CLIP**:0.129, **Human**: 100%

(k) A quiet conversation with the oldest inhabitants of the earth.
**CLIP**:0.131, **Human**: 90.9%

(l) The sun takes a bow, exiting stage west.
**CLIP**:0.116, **Human**: 100%

Figure 13: **Examples of false negative image-text pair (2)**.

# G Top-k Retrieval Results

We report additional top-k retrieval of random samples from IMP, the original captions and the image corresponding to the retrieved text.

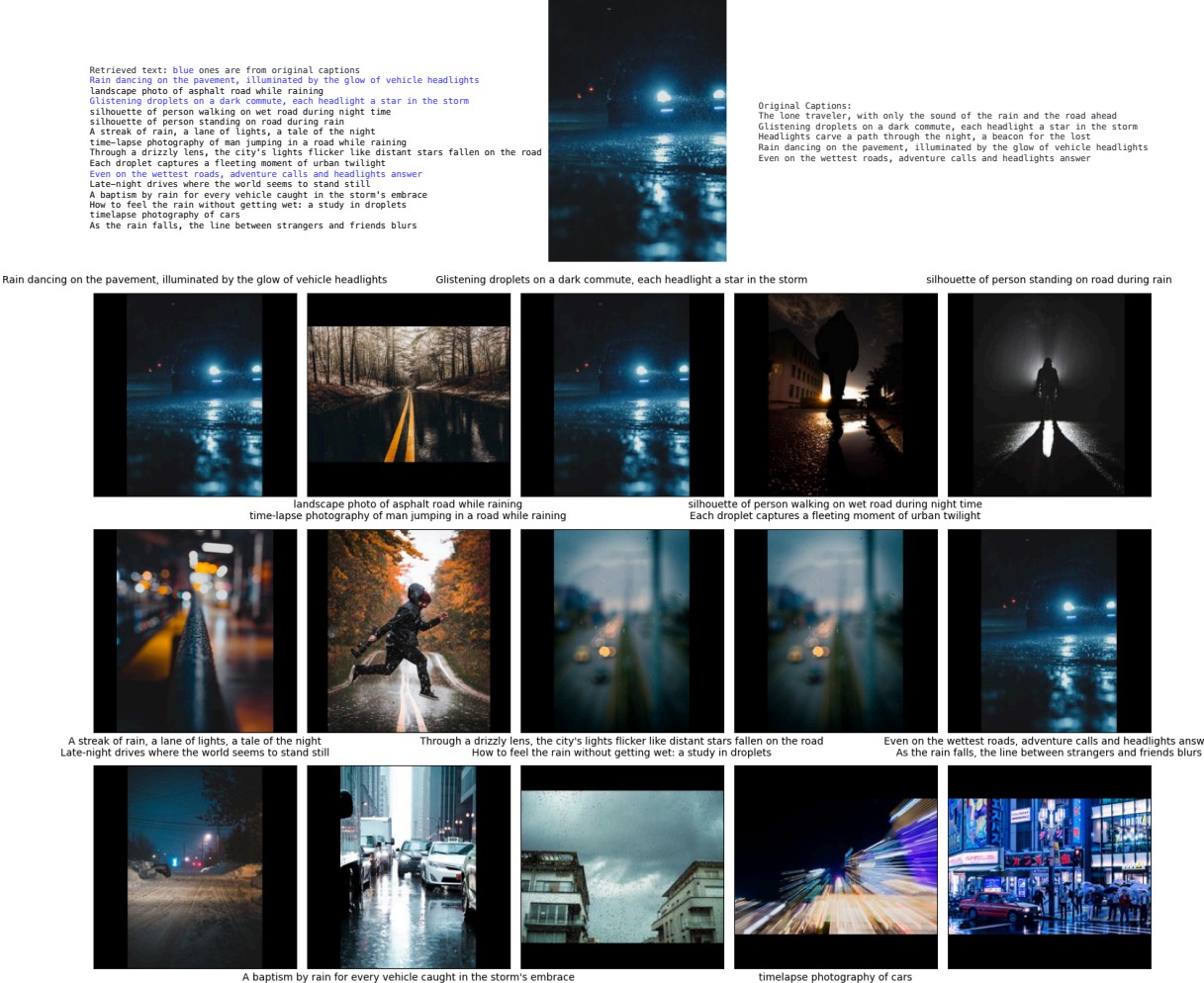

Figure 14: Example of top-k retrieval of random samples (1) from IMP.

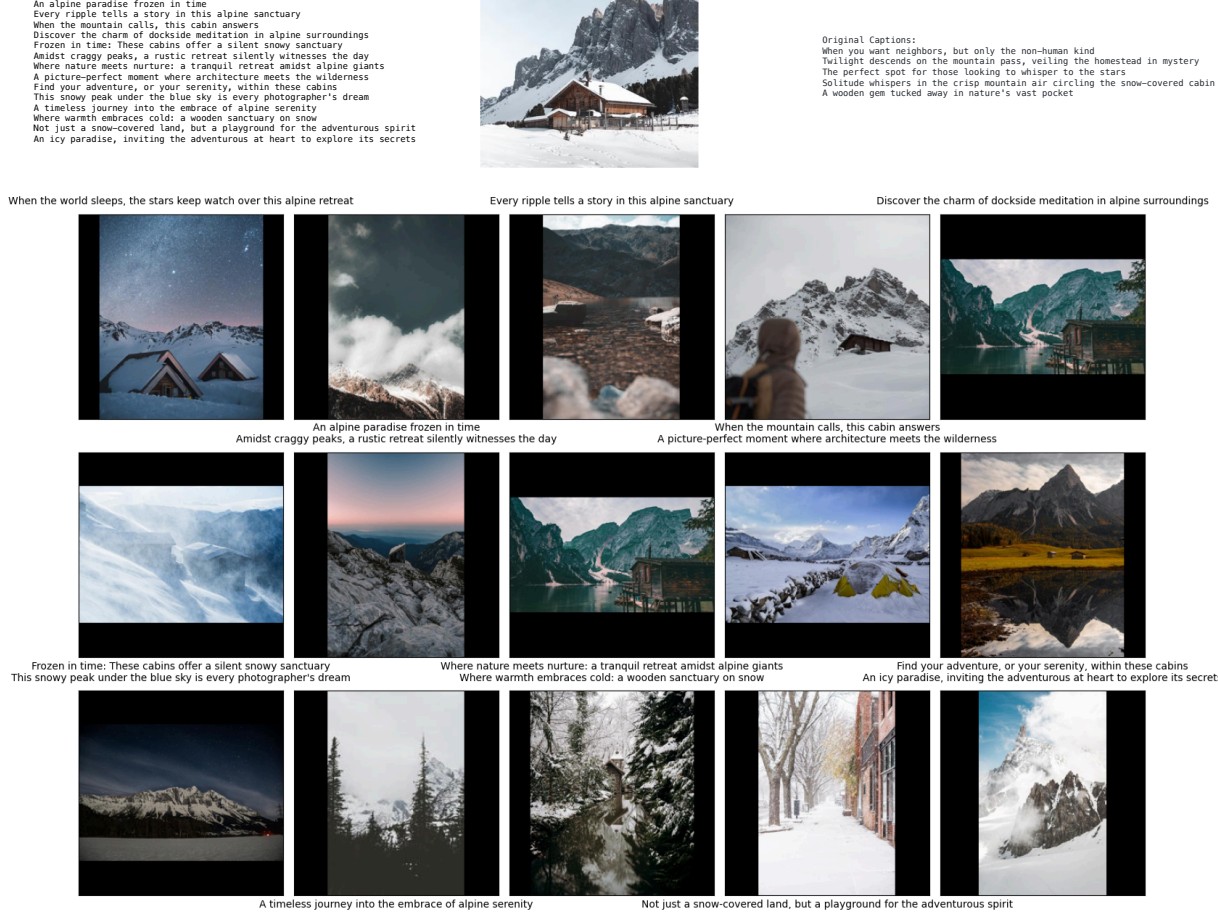

Figure 15: Example of top-k retrieval of random samples (2) from IMP.

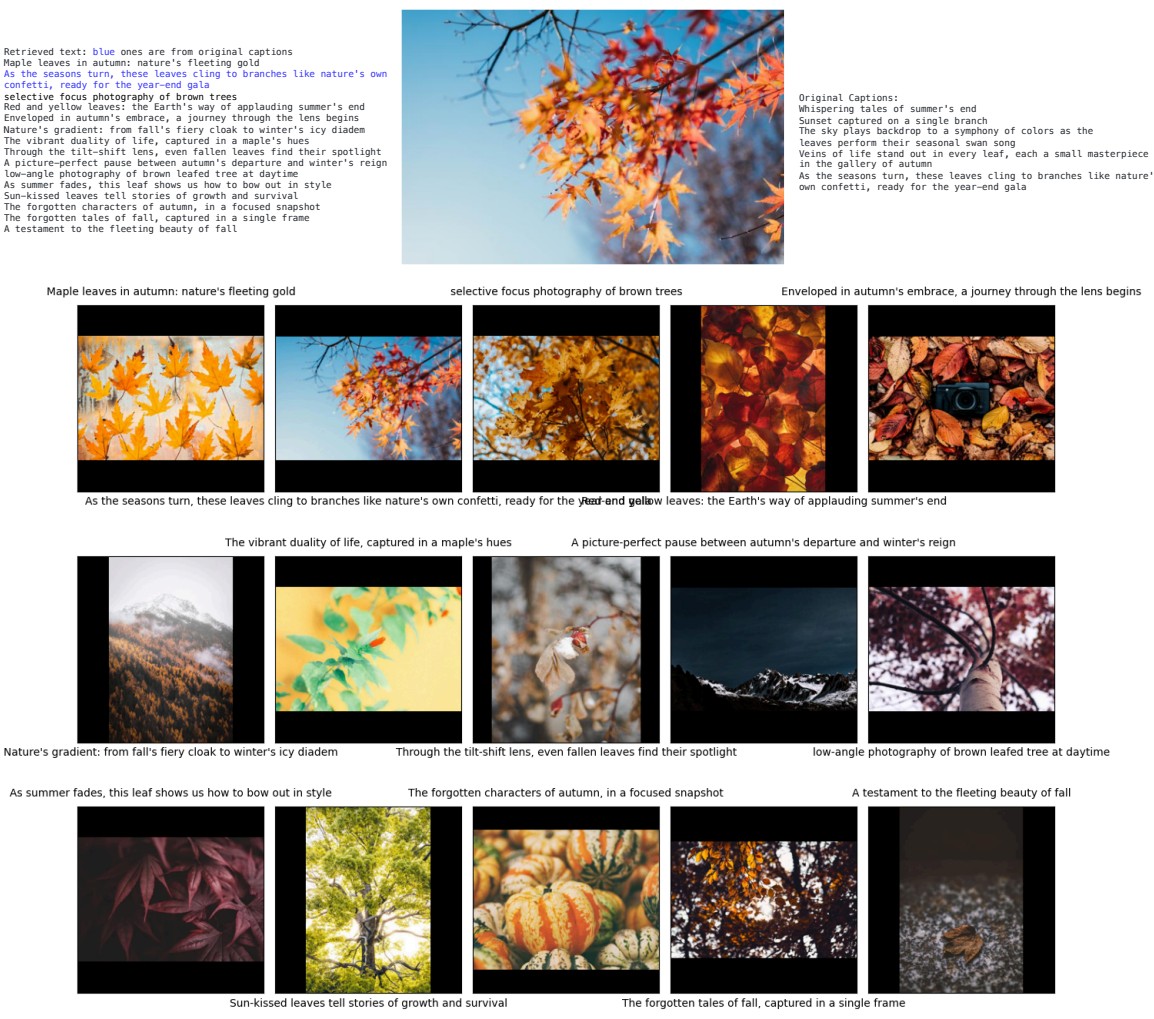

Figure 16: Example of top-k retrieval of random samples (3) from IMP.

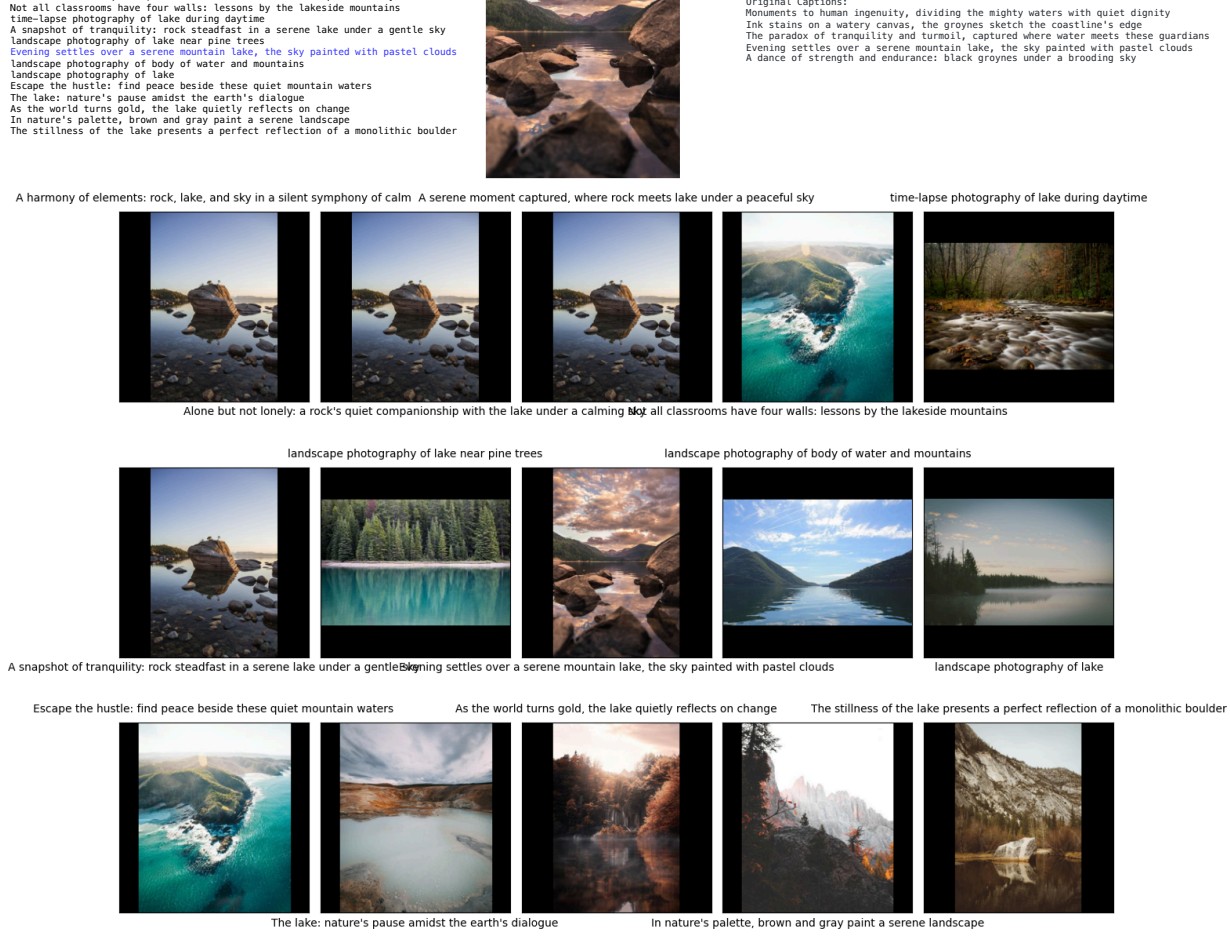

Figure 17: Example of top-k retrieval of random samples (4) from IMP.

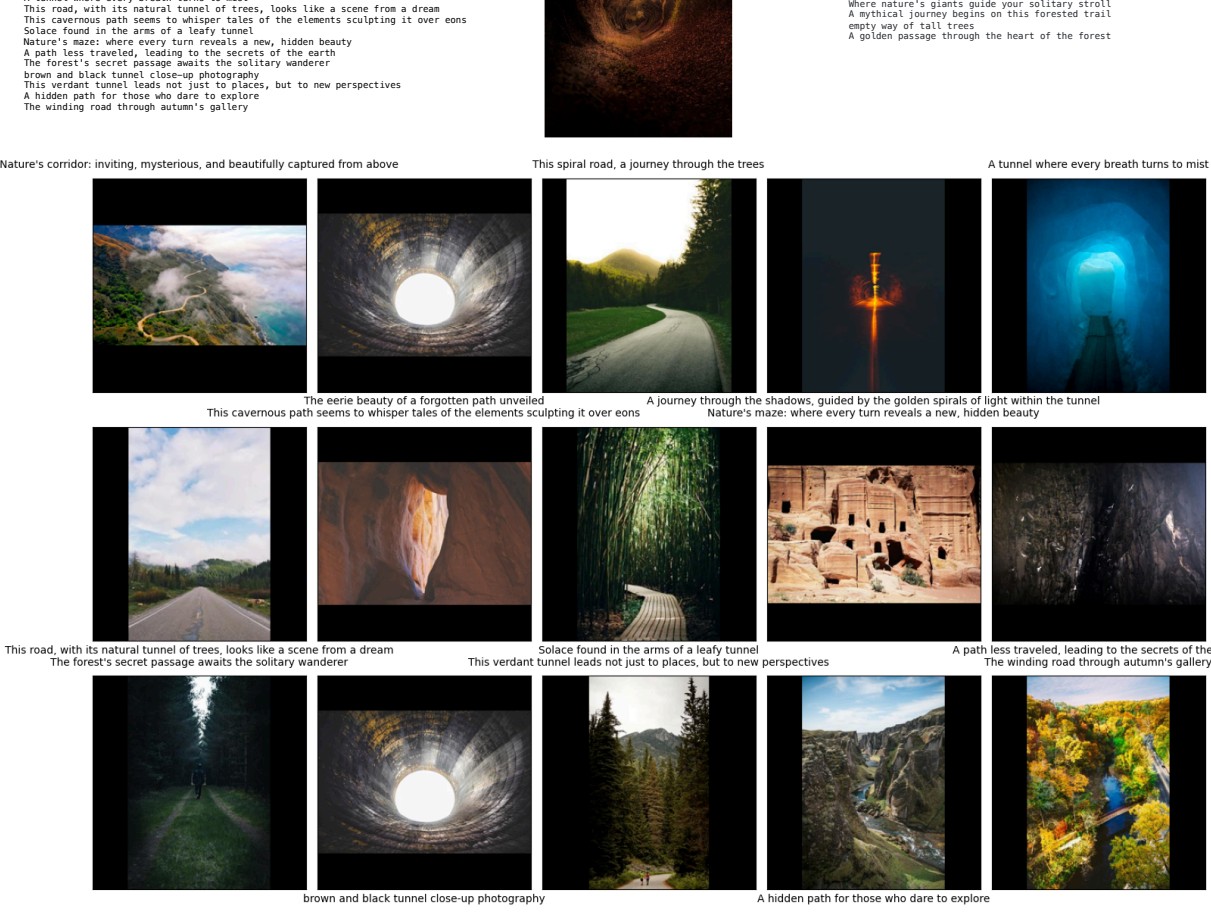

Figure 18: Example of top-k retrieval of random samples (5) from IMP.

## H    Qualitative Examples Comparison across Datasets

We report additional qualitative examples comparing examples from IMP with MS-COCO, Flickr30k, ConceptualCaptions, and Redcaps.

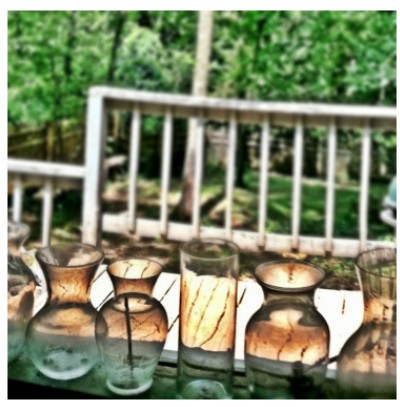

Many different vases lined up on a shelf.
a close up of a number of vases on a table
A table topped with lots of vases sitting next to a bench.
some vases sitting on a window ledge on a sunny day
Several glass vases sitting on a shelf near a window.

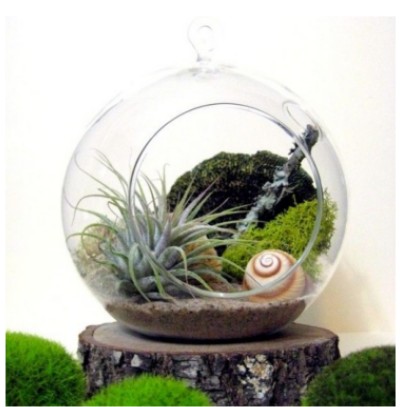

i am totally going to make a terrarium .

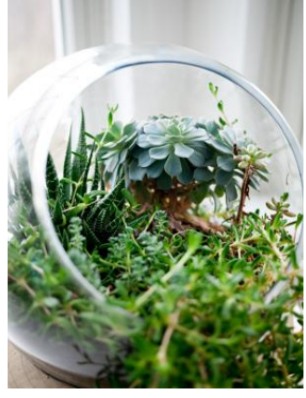

Delicate ecosystems, perfectly encased in geometric simplicity
The delicate dance of light and life within a glass sanctuary
From the desert to the desktop: A journey in miniature
Gaze upon this glass-enclosed garden, where succulents bask in a self-contained paradise
Nature's artistry, meticulously maintained in a transparent dome

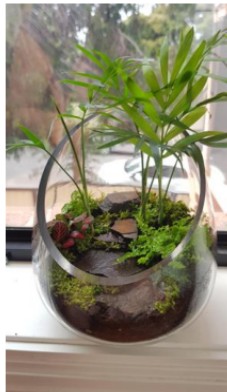

first attempt at a terrarium. moss and rocks are all locally sourced!

Figure 19: Examples from IMP (top-left), CC3M (top-right), MS-COCO (bottom-left), and Redcaps (bottom-right).

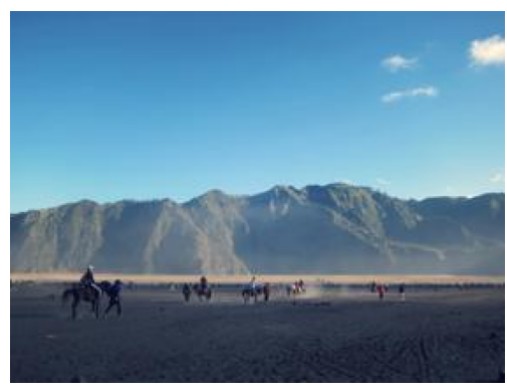

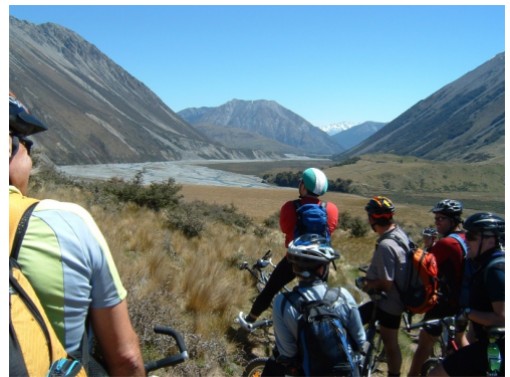

```
In the distance, a group of riders adds life to the quiet domain at the mountain's base       taking a break - mountain biking
United with the wind, each stride a symphony of power and grace
A journey back in time, riding the trails of history
Galloping towards the horizon, where earth meets sky
A celebration of heritage and the simple joy of being together
```

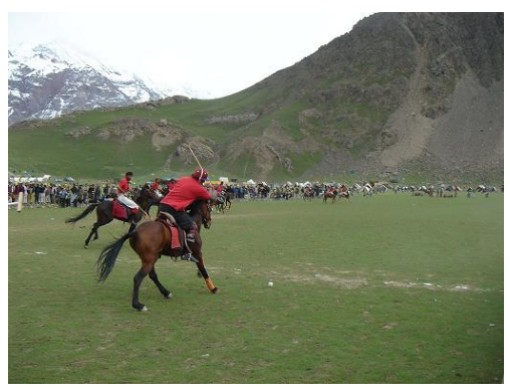

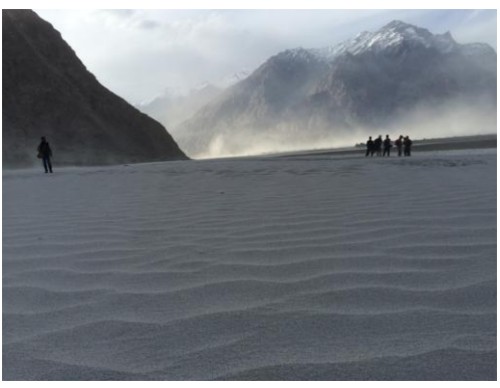

```
A man riding on the back of a brown horse through a lush green field.       world first cold desert...skardu
Two men are horse racing in front of a huge crowd.
a number of people on horses playing polo
There are some people riding horses on a field
A group of polo players on a field with horses next to a hill
```

Figure 20: Examples from IMP (top-left), CC3M (top-right), MS-COCO (bottom-left), and Redcaps (bottom-right).

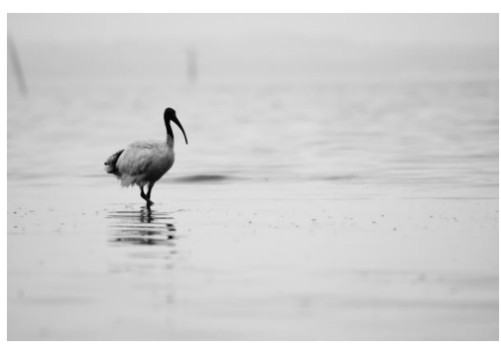

Nature's monochrome ballet: a black and white performer pirouettes on water
A single ibis in an expanse of water, a minimalist's dream in black and white
Amidst the whispers of water, a black and white spectacle unfolds in silent majesty
The art of stillness, as a black and white avian grace adorns the tranquil waters
Navigating the calm waters, a striking visitor contrasts the serene landscape

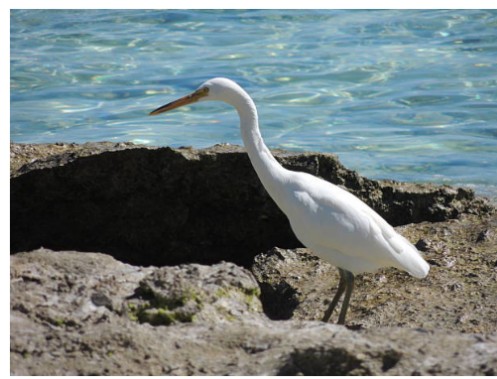

short story by the sea

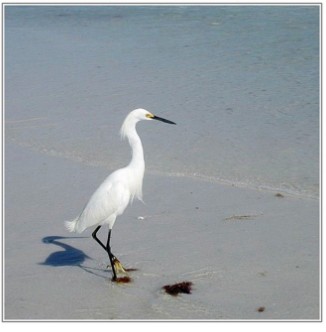

a single bird standing on the beach by the water
A white bird standing on beach with water coming in.
A large white bird on a beach in the water.
A white bird is standing at the water's edge.
A bird stands at the edge of the water at the beach.

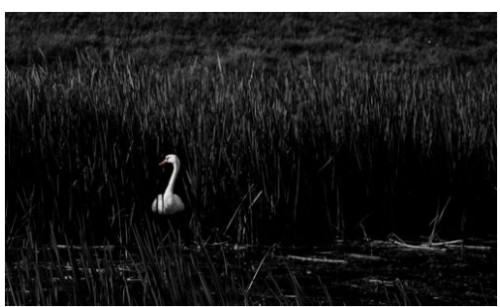

good crop for minimalism?

Figure 21: Examples from IMP (top-left), CC3M (top-right), MS-COCO (bottom-left), and Redcaps (bottom-right).

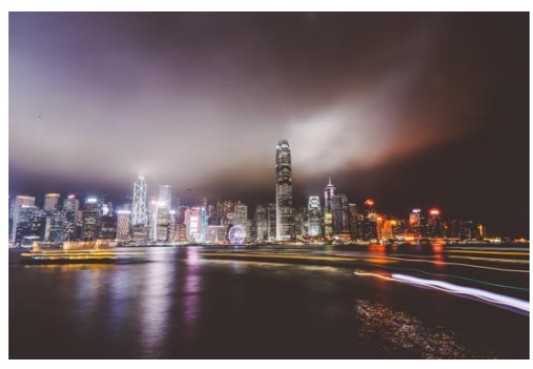

Nightfall brings a sparkle, city lights mimicking the stars
Echoes of footsteps, whispers of the high-rise
The geometric beauty of urban sprawl
Night falls, and the city's heartbeat is captured in lights and reflections
Nighttime cityscape by the bay, where the urban glow meets the sea

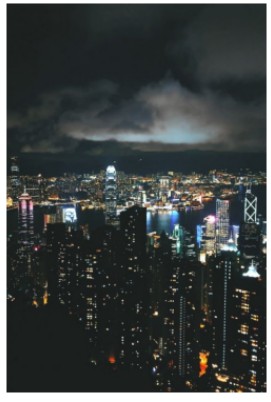

turn on the lights ~

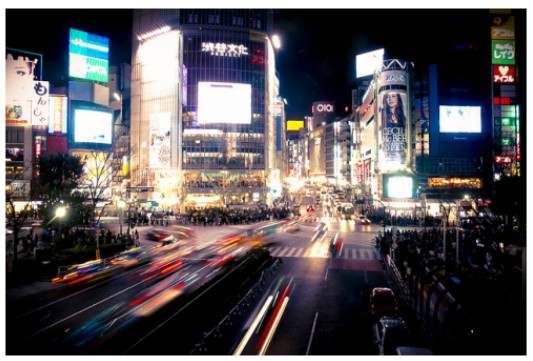

Busy city street with traffic and pedestrians at night.
Downtown at night with a lot of lights and people.
A busy city street bright with lights at night
Traffic is going down a city street at night.
Large city with a lot of lights and vehicles in Asian.

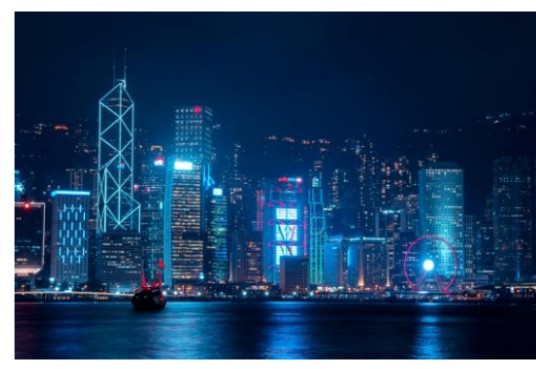

what i imagine when i see hong kong's skyline

Figure 22: Examples from IMP (top-left), CC3M (top-right), MS-COCO (bottom-left), and Redcaps (bottom-right).

# I   SentenceBERT and BERTscore Density Plots

We report the density of the sentenceBERT similarity and BERTscore of co-captions in IMP, MS-COCO, and Flickr30k.

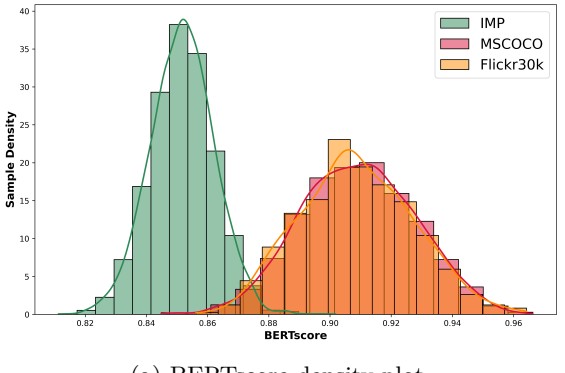
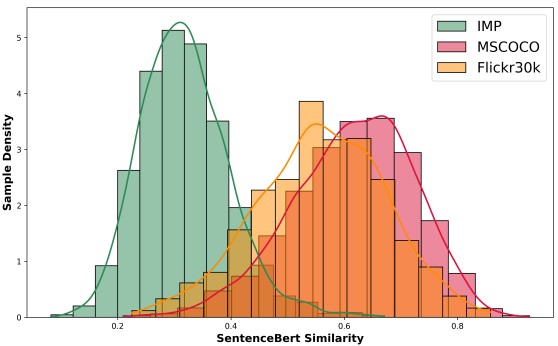

(a) BERTscore density plot.                    (b) SentenceBERT density plot.

# J   VQAscore and CLIPscore Evaluation

Although decoder-based VLMs (such as LLaVA with VQAscore) are not within the primary scope of this paper, we acknowledge their relevance. To this end, we evaluated IMP using both CLIPscore and VQAscore metrics across all difficulty splits and report the result in Table 11. We chose CLIP-flant5-xl as the VQAscore model, and CLIP-ViT/B-32 (trained on CLIP400M) as the CLIPscore model.

Table 11: CLIPscore and VQAscore evaluated on IMP dataset across difficulty categories.

| Category | CLIPscore | VQAscore |
|----------|-----------|----------|
| Full IMP | $0.25 \pm 0.03$ | $0.66 \pm 0.17$ |
| Easy | $0.28 \pm 0.02$ | $0.73 \pm 0.16$ |
| Medium | $0.25 \pm 0.02$ | $0.66 \pm 0.16$ |
| Hard | $0.22 \pm 0.02$ | $0.58 \pm 0.17$ |

We report the visualization of VQAscores and CLIPscores, and show case an example.

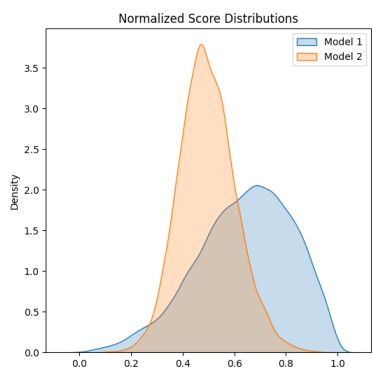
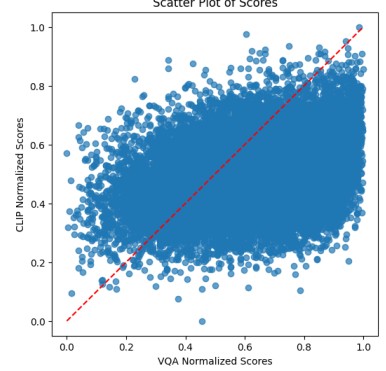
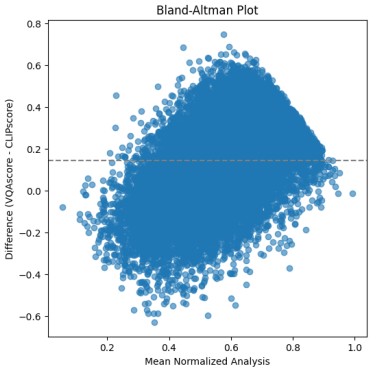

(a) Distribution of normalized VQAscores and CLIPscores

(b) Scatter Plot of normalized VQAscores and CLIPscores

(c) Bland-Altman Analysis of VQAscores and CLIPscores

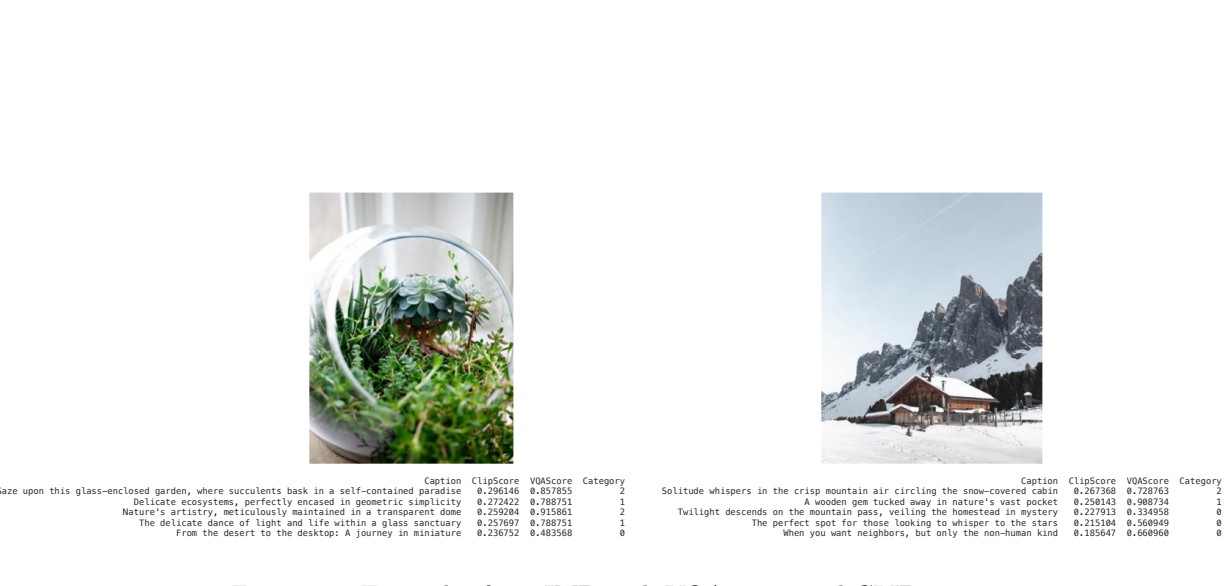

| Caption | ClipScore | VQAScore | Category |
|---|---|---|---|
| Gaze upon this glass-enclosed garden, where succulents bask in a self-contained paradise | 0.296146 | 0.857855 | 2 |
| Delicate ecosystems, perfectly encased in geometric simplicity | 0.272422 | 0.788751 | 1 |
| Nature's artistry, meticulously maintained in a transparent dome | 0.259204 | 0.915861 | 2 |
| The delicate dance of light and life within a glass sanctuary | 0.257697 | 0.788751 | 1 |
| From the desert to the desktop: A journey in miniature | 0.236752 | 0.483568 | 0 |

| Caption | ClipScore | VQAScore | Category |
|---|---|---|---|
| Solitude whispers in the crisp mountain air circling the snow-covered cabin | 0.267368 | 0.728763 | 2 |
| A wooden gem tucked away in nature's vast pocket | 0.250143 | 0.908734 | 1 |
| Twilight descends on the mountain pass, veiling the homestead in mystery | 0.227913 | 0.334958 | 0 |
| The perfect spot for those looking to whisper to the stars | 0.215104 | 0.560949 | 0 |
| When you want neighbors, but only the non-human kind | 0.185647 | 0.660960 | 0 |

Figure 25: Examples from IMP with VQAscores and CLIPscores

