# OpenReview forum: "Image Polysemy in Contrastive Vision-Language Learning"
_TMLR — Rejected by TMLR_

### Review · Reviewer_tnxx · 2025-02-26

**Summary Of Contributions:**

Every image can be described by multiple captions with different semantic meanings. This work argues that the existing contrastive learning paradigm overlooks this issue (i.e., the difficulty of different captions). Thus, this work introduces a dataset with diverse image captions from different difficulty levels, dubbed Image Polysemy (IMP). They define the "caption difficulty" based on the degree of misalignment between human and machine understanding, and use human evaluation to confirm the quality of these image-text pairs. They also evaluate several existing VLMs on the new proposed IMP benchmark.

**Audience:**

Yes

**Broader Impact Concerns:**

NA.

**Claims And Evidence:**

No

**Requested Changes:**

It would be very important to show:
1) More objective definition about the "caption difficulty".
2) More intuitive comparisons with existing image-caption datasets to demonstrate the advance of the proposed dataset.

**Strengths And Weaknesses:**

Strengths:
1. The research problem (i.e., discussing the semantic level of image-pair data) itself is very important to real-world scenarios. This proposed dataset can help to bridge some gaps between existing VLM research with real-world applications.

Weaknesses:
1. Overclaim the contributions. As listed in the introduction section, the first key contribution is "uncover polysemy in web-scale dataset". To be honest, I think this issue is a common sense, even without using the proposed clustering-based method.
2. The whole definition about "caption difficulty" is very vague and not rigorous. There is no accurate evaluation of the "degree of misalignment between human and machine understanding". In other words, the "difficulty level" of an image-caption pair may change based on the choice of different VLMs.
3. From Figure 5, the similarity density distribution of co-caption and image-text pairs, it is hard to understand why IMP is better than MSCOCO.
4. There is no technical contribution. They only propose this benchmark, and don't show any technical ideas for using this dataset to improve models' cross-modal retrieval or binary image-text classification performance.

---

> ### Author Response · Authors · 2025-03-26
> **Rebuttal Response (1/2)**
>
> We thank reviewer tnxx for their insightful comments and their recognition of the importance of the research problem. In the following we address the concerns they raised.
>
> ### Q1: Claimed contribution “uncover polysemy” is trivial.
>
> Polysemy in web-scale datasets, while intuitively known, has not previously been quantified in a systematic manner. Our prime contribution is not merely identifying the existence of polysemy, but rather quantifying it through a novel clustering-based approach applied at scale, which provides a new framework for evaluating cross-modal semantic diversity. We have revised the manuscript accordingly to ***emphasize the systematic quantification of polysemy clearly***.
>
> ### Q2: Vague definition of “caption difficulty”
>
> Caption difficulty is intended to capture how challenging it is expected to be for a VLM to match a particular caption to an image (or an image to that caption). Unfortunately, no metric exists which perfectly captures this notion - instead we propose to approximate this by looking at the agreement over a set of VLMs. In our paper we considered 7 models, as the reviewer expressed concerns about whether this is sufficiently rigorous perform an analysis to measure how robust the difficulty assignment is.
>
> For this robustness analysis we considered a larger set of **30** models and randomly selected a subset of 7 models from this to repeat the caption difficulty assignment. We repeated this process **10** times. This process is analogous to annotating samples with multiple annotators, hence we measure the internal agreement between annotators with Cohen's Kappa.
>
> Across all repetitions, we observed Cohen’s Kappa values consistently ranging from 0.65 (substantial agreement) to 0.87 (almost perfect agreement). This high level of internal consistency strongly supports the robustness and objectivity of our caption difficulty assignment methodology, alleviating the reviewer’s concerns.
>
> Moreover, our analysis reveals that difficulty categorization is primarily influenced by the performance extremes among the selected models: specifically, shifts between “easy” and “medium” or “medium” and “hard” occur depending on whether the weakest model improves or the strongest model deteriorates. An explicit example illustrating such shifts is provided in Figure 9 of the manuscript.
>
> We have added the results from this robustness analysis to the manuscript in ***Section 4.2***.
>
> ### Q3: Intuitive Dataset Comparisons
>
>
> To further highight the contribution of IMP and how it differs from prior datasets we perform two comparisons, one qualitative and one quantitative. These comparisons demonstrate that IMP is unique in that it has multiple captions per image which are diverse semantically, thereby having explicit polysemy and thereby providing a benchmark for how VLMs handle polysemy.
>
> #### Qualitative Comparison
>
> We sampled representative images from **IMP** and retrieved visually similar images from four popular datasets: **MS-COCO**, **CC3M** and **Redcaps**. For each dataset, we included corresponding captions, as shown in ***Appendix H*** of the revised manuscript.
>
> Our qualitative comparison highlights key differences:
>
> - **IMP** uniquely provides multiple captions per image explicitly annotated to be semantically diverse, effectively capturing explicit polysemy.
> - **MS-COCO** also includes multiple captions per image, but these captions are typically semantically similar, yielding easy to medium difficulty.
> - **Redcaps** and **CC3M** offer more challenging and diverse captions but typically provide only a single caption per image.
>
> #### Quantitative Comparison
>
> We quantitatively compared IMP against MS-COCO and Flickr30k, both containing human-annotated multiple captions per image. For each dataset, we computed the inter-caption similarity using Sentence-BERT (purely textual semantic similarity) and BERTScore (semantic textual similarity).
>
> Specifically, we randomly permuted co-captions to create derangements and computed similarity scores. Density plots of these metrics are available in ***Appendix I***.
>
> Below we summarize these comparisons numerically:
>
> | Dataset | Sentence-Bert | BERTscore |
> |--- | --- | --- |
> |Flickr30k| 0.56 +- 0.11 | 0.91 +- 0.02 |
> |MS-COCO| 0.62 +- 0.11 | 0.91 +- 0.02|
> |IMP |**0.32 +- 0.08** | **0.85 +- 0.01**|
>
> we can observe that with both Sentence-Bert and BERTscore the co-caption similarity for MS-COCO and Flickr30k are much higher than for IMP, underscoring the diversity of the captions in IMP. As all three datasets were human-verified, we can determine that this diversity is the result of image polysemy.
>
> Both metrics demonstrate significantly lower caption similarity scores for IMP compared to MS-COCO and Flickr30k, clearly underscoring IMP’s semantic diversity. Given that all three datasets were human-verified, this substantial diversity uniquely reflects true image polysemy rather than annotation variability.

---

> ### Author Response · Authors · 2025-03-26
> **Rebuttal Response (2/2)**
>
> ### Q4: Technical Contribution
>
> We acknowledge the reviewer’s perspective regarding explicit technical contributions. However, our primary goal and intended contribution is the introduction of a rigorous benchmark (IMP) specifically designed to evaluate image-caption polysemy—a significant but previously underexplored aspect of vision-language understanding. Historically influential datasets (e.g., ImageNet, MS-COCO, Flickr30k) similarly provided foundational contributions by highlighting clear model limitations, thereby inspiring subsequent technical innovation.
>
> Furthermore, IMP uniquely quantifies semantic diversity systematically and at scale, providing a strong foundation to motivate future technical advancements. Indeed, our initial analyses already suggest that one-to-many approaches (such as probabilistic embeddings or multiple-instance embeddings) offer promising directions to surpass the original CLIP backbone, potentially due to their architectural or loss-function enhancements.
>
> To further clarify the significance and impact of our benchmark, we’ve explicitly outlined in the revised conclusion several promising future research directions enabled by IMP:
> 1. Polysemy-aware training objectives: designing loss functions explicitly capable of modeling multiple plausible positives.
> 2. Polysemy-aware model architectures: extending current approaches to better represent rich semantic diversity and uncertainty.
> 3. Novel retrieval and classification metrics: explicitly addressing evaluations with multiple plausible image-caption alignments.
>
> These points have been explicitly incorporated into the ***Discussion section*** of our revised manuscript.

---

> ### Comment · Reviewer_tnxx · 2025-04-06
>
> Thanks for the response.
>
> **For Q1**: As the authors claim that they propose a quantification method, how can we evaluate the quality of the quantification results?
>
> **For Q2**: For the "caption difficulty", I think it should be an objective metric to evaluate the "difficulty". Or it should have a clear definition of the difficulty levels. Although it shows that current VLMs can share some common behaviors on the benchmark, it is still unclear whether the caption difficulty is a well-defined concept.
>
> **For Q4**: The technical contribution parts are still limited.

---

> > ### Author Response · Authors · 2025-04-07
> >
> > We thank reviewer tnxx for their further questions and clarification.
> >
> > > Q1: how can we evaluate the quality of the quantification results?
> >
> > As our work is the first to propose a method for attempting to quantify the polysemy in web-scale datasets there is no metric to compare it against unfortuntately. However, we believe the approach we suggest in section 3.1 is an intuitive manner to measure polysemy - as it closely aligns with the definition of polysemy.
> >
> > Moreover, we demonstrate that our approach is able to verify that CC12M has more polysemy than MS-COCO, which is a result that we would intuitively expect - as our method is able to confirm this it speaks to the veracity of our proposed approach.
> >
> > > Q2: For the "caption difficulty", I think it should be an objective metric to evaluate the "difficulty".
> >
> > We would kindly ask reviewer tnxx if they could further clarify their definition of "objective" in this context, and why they believe this is a necessity. To facilitate this we will explain our reasoning for the current metric.
> >
> > The ground truth for caption difficulty is based on human verification of image-caption pairs, and we determine the difficulty of captions **for VLM** by the extent at which VLM are able to match these pairs. *This inherently requires observation of VLM performance*. If we were to use a single VLM then the conclusions about difficulty would only be valid for that VLM, hence, we sample a larger amount of VLM and aggregate their findings to be able to arrive at a generalisable conclusion.
> >
> > Our robustness analysis confirms that the difficulty categorisation is robust to the choice of VLM, which supports our approach as leading to a good estimate of caption difficulty. Hence, if we consider the results of a single VLM to be "subjective" then a robust estimate based on sampling across VLM would be "objective" - however, we would appreciate clarification from the reviewer if this aligns with their view on "subjective" and "objective" in this context.
> >
> > > Q2: Or it should have a clear definition of the difficulty levels.
> >
> > The definitions we used for the difficulty levels are as follows:
> > - Easy: All VLM consider this caption a valid match for its paired image
> > - Medium: Some VLM consider caption a valid match for its paired image
> > - Hard: No VLM considers this caption a valid match for its paired image
> >
> > We believe this naturally reflects the difficulty of captions for VLM.
> >
> > > For Q4: The technical contribution parts are still limited.
> >
> > The focus of our manuscript is on proposing a benchmark to evaluate how VLM handle polysemy. To do so we propose a method for quantifying polysemy in existing datasets, and a method for constructing a dataset with multiple human-verified and semantically diverse captions. Neither of these methods have been proposed before.
> >
> > Moreover, our findings provide clear insights into the limitations and capabilities of VLM - a category of methods widely used by the community. As such we believe the contributions of our work are in line with the scope and acceptance criteria of TMLR.

---

> > > ### Comment · Reviewer_tnxx · 2025-04-08
> > >
> > > For an objective metric, I mean that based on this metric, the definition of the difficulty level of each sample should be consistent and fixed. If there are no objective metrics, the difficulty level for each sample will change based on different choices of VLMs. Although the authors claimed they randomly sampled different VLMs, it is hard to say the selected VLMs are representative enough. That is why I think it would be better if we had an objective metric to evaluate the caption difficulty.
> > >
> > > On the other hand, since there is no clear definition of caption difficulty, we also cannot evaluate the quality of these quantification results (this is also the motivation why I asked Q1). And this is one of the main contributions of this paper, it's hard to convince me that the proposed benchmark is high-quality enough.
> > >
> > > Overall, as I said in my original review, image polysemy is a commonsense (i.e, an image often corresponds to multiple captions, each varying in semantic meaning). I knew that "polysemy" is a comprehensive concept that has very broad meanings. And the reasons for polysemy come from multiple reasons: 1) These captions describe different content appearances, 2) Some captions contain some extra "unjudgable" hallucinations, or even 3) Some captions may convey some implicit meanings, and so on.

---

> > > > ### Author Response · Authors · 2025-04-09
> > > >
> > > > We thank reviewer tnxx for the additional clarification.
> > > >
> > > > > the definition of the difficulty level of each sample should be consistent and fixed. [..] That is why I think it would be better if we had an objective metric to evaluate the caption difficulty.
> > > >
> > > > Given this definition we do not believe such a metric *can* exist. Difficulty is inherently a relative and observed concept. Benchmarks are, in part, proposed because they present a new frontier of difficult cases. Before AlexNet and deep learning, ImageNet was considered a difficult benchmark, similarly, MS-COCO was difficult at one point. As technology has improved so has the definition of difficulty for these tasks.
> > > >
> > > > The difficulty levels in IMP are explicitly a snapshot of the difficulty that the captions present for current-era VLM. Hopefully, technology will improve that in approximately five years, many of the IMP captions will shift towards easy or medium. However, at present, the image-caption pairs in IMP are explicitly validated as matches by humans, yet VLMs cannot consistently reproduce this. Our aim with IMP is to demonstrate this phenomenon and to present a benchmark that makes it possible to measure progress on handling polysemy.
> > > >
> > > > > Although the authors claimed they randomly sampled different VLMs, it is hard to say the selected VLMs are representative enough.
> > > >
> > > > The list of VLMs sampled is based on the OpenCLIP library, from which we used the following architectures \[RN50, RN101, ViT-B-32, ViT-B-16, ViT-L-14, ViT-L-14-366, ViT-H-14, ViT-G-14, convnext_base, convnext_large, EVA01, EVA02, COCA \] trained on (if checkpoints were available) these datasets \[yfcc5m, cc12m, openai400M, commonpool(four sizes), LAION(two sizes), datacomp(four sizes)\]. We believe this represents a diverse overview of the various VLM available.
> > > >
> > > > > On the other hand, since there is no clear definition of caption difficulty
> > > >
> > > > Our definition for caption difficulty is based on a repeatable procedure that gives robust results across different samples of VLM used. Moreover, we believe this definition is easy and intuitive. Given that we know that the image and caption form a valid pair (as confirmed by human annotators) we determine the difficulty based on whether VLM are able to reproduce this pairing. The caption for which fewer VLM are able to reproduce the pairing are then logically considered more difficult. Moreover, this procedure is validated by the fact that we consistently see that other VLM (which were not included in the difficulty categorisation) perform much worse on the hard captions than they do on the medium or easy captions. This confirms that our definition of difficulty generalises to 'unseen' VLM.
> > > >
> > > > > Overall, as I said in my original review, image polysemy is a commonsense (i.e, an image often corresponds to multiple captions, each varying in semantic meaning).
> > > >
> > > > We thank the reviewer for pointing this out, as this is the core motivation for our work - we find it *incredibly surprising* that despite image polysemy being so commonplace in human communication, it is ignored in the fundamental formulations that underlie VLM. The standard contrastive loss that is used to train many, if not most, VLM does not accommodate polysemy - nor are there approaches that improve on this formulation to be able to handle polysemy.
> > > >
> > > > Moreover, existing benchmarks are typically built through annotation procedures that are designed to reduce polysemy. Yet, as we show, polysemy does (perhaps unsurprisingly) exist in the web-scraped datasets that are being used to train VLM. Hence, we believe it is both valid and crucial to investigate how VLM trained on such data are able to handle polysemy.
> > > >
> > > > Our findings show that VLM are unable to handle polysemy well, and that scaling data our model parameters is not sufficient to overcome this issue. To our knowledge there exists no other benchmark which could support such a conclusion, which underscores the empirical value of our work and the value of IMP as a benchmark.
> > > >
> > > > > And this is one of the main contributions of this paper, it's hard to convince me that the proposed benchmark is high-quality enough.
> > > >
> > > > IMP is designed as a practical and robust framework to measure current VLM performance on image polysemy, providing a clear baseline from which future progress can be measured, and our paper explicitly provides empirical evidence supporting our claims. We understand and appreciate the reviewer’s preference for a more universally “objective” notion of caption difficulty, however, we currently are not aware of prior literature or established metrics that provide such a universally accepted definition. If the reviewer has suggestions or is aware of relevant literature or alternative methods we could explore, we would greatly appreciate the recommendation.

---

### Review · Reviewer_Jbq1 · 2025-03-06

**Summary Of Contributions:**

This paper introduces a new benchmark IMP (Image Polysem) to measure VLM’s understanding of image captions of different levels (easy/medium/hard). To build this dataset, 5K images are sampled from Unsplash Lite Dataset; then captions are retrieved from CC3M/CC12M by image similarity and filtered by human; finally 5 captions per image are selected by the LOWEST pair-wise SentenceBert similarity.

The authors evaluated SOTA VLMs on IMP, demonstrating significant gaps on semantically complex captions. The results show that currently VLMs are good at understanding captions descriptive to the objects in the image, but hard to address the semantic diversity in captions. This paper also shows the importance of data quality and scaling, but they are still insufficient to solve the semantic diversity issue.

**Audience:**

Yes

**Claims And Evidence:**

Yes

**Requested Changes:**

[Major] Can we show the top-k retrieved captions together with the ground-truth captions for some example images? So that we can better understand the quality of IMP captions. Sometimes the low eval numbers may just be caused by the inaccuracy of the captions in IMP.

[Minor] In section 3.2 “highlighting the density of the CC12m dataset” – is it possible just because CC12M is much larger than COCO (40x larger)?

[Minor] Could you explain in the “CC12M Geodesic Distance” curve of Figure 3 why similarity also grows when radius increases?

[Minor] In introduction, the paper claims “curated through a combination of web-scale data and human annotation”. The dataset has only 5m images, so “web-scale” is clearly not true here. Same for other places, such as related work.

**Strengths And Weaknesses:**

Strengths
- The problem this paper addresses is important, but neglected by the community. When evaluating VLM’s semantic understanding capabilities, captions which describe the concrete objects in the image are often used (though in benchmarks there have been multiple captions per image, they all similarly focus on the same concrete objects in the image with different rewording).
- The open sourced IMP benchmark enables the community to assess VLM’s understanding of captions of diverse difficulty. The evals in this paper have shown the headroom for the SOTA VLMs, which encourages training stronger VLMs which can address more difficult semantics in captions.
- The evaluation is comprehensive, which has considered both the top-k accuracy and also the ranking capability (meanR/medR/AUPRC).

Weaknesses
- The captions are retrieved from existing datasets by image similarity. Even though results are filtered by two annotators, there’s still a very high chance to have inaccurate description in the retrained captions (because captions are not curated by annotators, which is non-trivial). - This sometimes makes VLM hard to rank captions by factuality. For example, in Figure 4: in caption 1, it’s unclear whether the tree is brown and the time is night (sky should be dark if at night); in caption 4, it’s unclear whether the season is winter or not.
Because of the issue above, the evaluation result is not very convincing. E.g., in Figure 9 (b), “Sunset” and “single” are not accurate (or can’t be inferred from the image).

---

> ### Author Response · Authors · 2025-03-26
> **Rebuttal Response**
>
> We thank reviewer Jbq1 for their insightful comments and their recognition of the value of benchmarks, the importance of the research problem, and of how comprehensive our evaluation is. In the following we address the concerns they raised.
>
> ### Q1: Captions retrieved by image similarity may contain inaccuracies
>
> We recognise the risk of inaccurate captions and how this may impact ranking captions by factuality. However, our annotation process was designed to discard inaccurate of inplausible captions for images, and in the case of minor inaccuracies the annotators corrected the information - details of this process are given in appendix C.4.
>
> Moreover, we would argue that the cases pointed out by the reviewer are better described as factual ambiguity. While it may be hard to say with absolute certainty that the tree in Figure 4 is brown, it is sufficiently likely that a human observer will take it for granted by using the caption to resolve the ambiguity. Similarly, given the relatively dark setting in this image it is sufficiently plausible for a human observer to accept that it is night in this image. Yet, there is also no clear visual information that excludes these interpretations (e.g., there is no clock visible or sun in the sky). Hence, complementary information from both modalities is used to resolve any ambiguities. Given the inherent ambiguity in language we would also argue such captions are impossible to exclude from any realistic dataset.
>
> As suggested by the reviewer, we have included additional experimental results of the top-k retrieved captions for a set of example images in ***Appendix G***. We discuss these results under the next heading.
>
> ### Q2: Top-k retrieval captions alongside GT captions
>
> Included in the manuscript in ***Appendix G*** we show the top-k retrieved captions for a query image, as well as the annotated image to which these retrieved captions belong. From these cases we can observe that the top-K captions are relevant to the image, and that some of the annotated captions are found among the top-K, but that not all captions (despite being relevant to the image) will be included in the top-K.
>
> ### Q3: Higher density in CC12M might result simply from larger dataset size
>
> Indeed, the density of CC12M is a consequence of the dataset size. Our aim here was to emphasise this, but we realise this remark may be ambigious in this context. As it does not contribute to our main point we have removed it.
>
> ### Q4: Figure 3 - similarity grows with increased radius
>
> We acknowledge that Figure 3 may be challenging to interpret due to the inclusion of two Y-axes and the ambigious explanation of how the geodesic distance is calculated. We have updated the caption and textual descriptions of this figure to explain the following:
>
> For the left Y-axis (the blue lines) the initial image-image similarity at radius 1 is 0 because there is only one image considered at this point, hence the similarity is undefined. Only from radius 5 we can measure the image-image similarity, after which it gradually drops as the radius increases. This naturally captures that as we consider a larger area in the embedding space the images within it become less similar. This occurs for both datasets, however, for all radius the similarity in CC12M is greater - which is not a consequence of the dataset size.
>
> The right Y-axis (red lines) shows the geodesic distance between an image query and the captions belonging to similar images (i.e., images close to the image query in CLIP space). Hence, what we observe is that as the radius grows we see a much greater increase in the geodesic distance for CC12M than for MS-COCO.
>
> As intepreting the geodesic distance values depends on the image-image similarity, we have chosen to include these in a single figure. One way to read this figure is to note that at a particular radius, the area in embedding space considered for each dataset is the same across both metrics. As such we can observe that from radius 10 to 25 there is only a small drop in image-image similarity for CC12M (dashed blue line), yet the geodesic distance (dashed red line) increases a lot, whereas the lines for MS-COCO remain effectively flat in this range. This shows that for a set of images from CC12M which is visually more similar than a set of images from MS-COCO, there is much more diverse text in CC12M. This shows that the polysemy in CC12M is greater than MS-COCO.
>
> ### Q5: Curated through web-scale data and human annotation
>
> Our usage of the web-scale dataset termininology was based on the CC12M paper (Changpinyo et al. 2021), which uses the web-scale terminology. However, we acknowledge that since 2021 datasets have grown tremendously, and that millions of images may not be considered web-scale anymore. We have explained the terminology used in the manuscript.

---

> > ### Comment · Reviewer_Jbq1 · 2025-03-27
> >
> > Thanks for the detailed explanation.
> >
> > > [Major] Can we show the top-k retrieved captions together with the ground-truth captions for some example images? So that we can better understand the quality of IMP captions. Sometimes the low eval numbers may just be caused by the inaccuracy of the captions in IMP.
> >
> > Thanks for the top-k examples Appendix G. It exactly shows the inaccuracy of the evals. For example (for each image I checked its top-5 retrieved captions),
> > - image 1 also matches "landscape photo of asphalt road while raining"
> > - image 2 also matches "An alpine paradise frozen in time", "Every ripple tells a story in this alpine sanctuary", "When the mountain calls, this cabin answers"
> > - image 3 also matches "Maple leaves in autumn: nature's fleeting gold", "selective focus photography of brown trees", "Red and yellow leaves: the Earth's way of applauding summer's end", "Enveloped in autumn's embrace, a journey through the lens begins"
> >
> > Some of them are even better than the captions from IMP. That being said, the low retrieval accuracy can't actually indicate the model's lower capability in polysemy understanding.

---

> > > ### Author Response · Authors · 2025-03-27
> > >
> > > We thank reviewer Jbq1 for the follow-up response, and taking the time to consider our updated manuscript.
> > >
> > > As pointed out, it is indeed possible that other captions (which were not included in the 5 ground truth captions) may be suitable for an image. This is the problem discussed in CrissCrossed captions (Parekh et al, 2021) and ECCV captions (Chun et al., 2022), which we reference in section 2.1. However, this issue is distinct from the image polysemy that we investigate, as demonstrated in our paper in two ways:
> > >
> > > 1) If the lower retrieval performance was caused by false negative annotations (e.g., possible positives that were not annotated) then this should equally affect all three difficulty categories (easy, medium, hard). However, as shown in Table 4 the retrieval performance for both medium and hard captions is significantly lower than easy. This difference is not explained by such cases.
> > >
> > > 2) Moreover, to avoid this issue altogether we performed a second type of evaluation, through the binary evaluation in section 5.3. This evaluation is completely unaffected by the possibility other positives, as it directly tests whether a caption which human annotators consider suitable for an image is considered suitable by a VLM. Here we similarly find that there is a difference between difficulty categories, with stronger models still performing well on medium difficulty, but all models performing notably worse on the hard captions.
> > >
> > > These findings demonstrate how VLM are affected by polysemy, which was made possible through our proposed IMP dataset.

---

> > > > ### Comment · Reviewer_Jbq1 · 2025-03-31
> > > >
> > > > Thanks for the further explanation.
> > > >
> > > > >If the lower retrieval performance was caused by false negative annotations
> > > >
> > > > I didn't claim "lower retrieval performance WAS caused by false negative annotations", but instead "lower retrieval performance COULD be caused by false negative annotations".
> > > >
> > > > > Here we similarly find that there is a difference between difficulty categories, with stronger models still performing well on medium difficulty, but all models performing notably worse on the hard captions.
> > > >
> > > > This actually doesn't answer my question directly. Some stronger models have better numbers on IMP, which doesn't mean better IMP numbers can reflect better model capability on polysemy understanding.
> > > >
> > > >
> > > > TLDR: from the added qualitative examples above (top-k image-to-text; it would be nice if we can have top-k text-to-image as well), my concern still holds. There lacks of enough evidence to show IMP is a fair benchmark to measure polysemy.

---

> > > > > ### Author Response · Authors · 2025-03-31
> > > > > **Rebuttal Response**
> > > > >
> > > > > We sincerely appreciate Reviewer Jbq1’s continued engagement and insightful comments.
> > > > >
> > > > > We acknowledge their concern regarding the possibility that lower retrieval performance could be influenced by false negative annotations (valid but unannotated captions). However, we emphasize that this does not invalidate our findings, nor the utility of IMP as a benchmark.
> > > > >
> > > > > To clarify our position:
> > > > > - Our primary objective with IMP is to benchmark **relative** model performance across different polysemy levels (easy, medium, hard). Even if false negatives occur, **we would expect them to similarly affect all three difficulty categories**. However, as shown in Table 4 and the binary evaluation (Sec. 5.3), there is a consistent performance degradation from easy to hard captions, clearly demonstrating that model capability is significantly influenced by polysemy rather than false negatives.
> > > > > - The binary evaluation conducted in Sec. 5.3 is specifically designed to mitigate potential false-negative effects by exclusively evaluating captions positively verified by human annotators. This evaluation again confirms significant and consistent performance differences across difficulty categories, strongly supporting the validity and fairness of IMP as a polysemy benchmark.
> > > > >
> > > > > We emphasize that IMP’s primary goal is to highlight relative performance differences among VLMs when faced with polysemic image-caption pairs, rather than claiming perfect absolute retrieval accuracy. However, we can still observe asymptotic performance improvement when the model size/ training size increase, as well as across different difficulty categories. Therefore, the consistent relative performance gaps we observe provide compelling evidence for the fairness and validity of IMP in assessing polysemy understanding.
> > > > >
> > > > > We agree with the reviewer that false negatives would undermine it, *if* IMP were intended purely as a retrieval test set. However, IMP is explicitly designed as a benchmark (with different difficulty levels) for assessing polysemy across various vision-language tasks, not retrieval tasks alone. Additionally, as mentioned explicitly in our Discussion and Future Work sections, we recognize the need for developing specialized metrics that better capture polysemic relationships, whether for retrieval, classification, or other downstream tasks.
> > > > >
> > > > > We also emphasise that completely resolving the potential false-negative annotation issue would require exhaustive human cross-validation, evaluating and annotating every possible positive caption across all images in the dataset. This approach, however, is prohibitively expensive and practically unrealistic at scale, nor do we believe that doing this would benefit our benchmark. Moreover, during the IMP annotation process, annotators are presented only with candidate captions specifically retrieved for each image. Thus, if a caption suitable for image A is also relevant to image B, this does not necessarily imply that the caption is flawed or should be altered to be less applicable to image B. Rather, it highlights the inherent complexity and ambiguity in caption-image relationships, further underscoring the need for benchmarks like IMP to assess and understand model performance under realistic semantic diversity constraints.
> > > > >
> > > > > Following the reviewer's recommendation, we will include qualitative examples of top-k text-to-image retrieval. We are actively preparing this supplementary analysis and will include explicit examples in the updated manuscript.
> > > > >
> > > > > We thank the reviewer for prompting us to clarify these points explicitly and for helping us reinforce IMP’s robustness and utility.

---

> > > > > > ### Comment · Reviewer_Jbq1 · 2025-04-03
> > > > > >
> > > > > > Thanks for the further explanation.
> > > > > >
> > > > > > > Our primary objective with IMP is to benchmark relative model performance
> > > > > > > We agree with the reviewer that false negatives would undermine it, if IMP were intended purely as a retrieval test set.
> > > > > > > we recognize the need for developing specialized metrics that better capture polysemic relationships
> > > > > >
> > > > > > All (or most) benchmarks are for measuring "relative" model performance. My concern is that if the ground-truth data in the benchmark is not convincing, the value of such a benchmark will be largely affected, no matter for retrieval or any eval tasks or for whatever metrics.
> > > > > >
> > > > > > > Even if false negatives occur, we would expect them to similarly affect all three difficulty categories.
> > > > > >
> > > > > > I am not convinced by "*expect* them to similarly affect". VLMs are affected much by their training data, so unless all models' training data has the same distribution, VLMs likely have different favors of aligned text given the same image.
> > > > > >
> > > > > > > We also emphasise that completely resolving the potential false-negative annotation issue would require exhaustive human cross-validation, evaluating and annotating every possible positive caption across all images in the dataset. This approach, however, is prohibitively expensive and practically unrealistic at scale, nor do we believe that doing this would benefit our benchmark.
> > > > > >
> > > > > > Yes, that is why building a high-quality benchmark is not easy :) I am not claiming each single example in the benchmark should have high-quality, but from the (randomly?) sampled top-k data above, I am not so convinced.

---

> > > > > > > ### Author Response · Authors · 2025-04-03
> > > > > > > **Rebuttal Response**
> > > > > > >
> > > > > > > Thank you for your continued engagement. We agree that building a high-quality benchmark is not easy, hence we spent a lot of effort ensuring that IMP is a high-quality dataset *for benchmarking image polysemy*.
> > > > > > >
> > > > > > > > My concern is that if the ground-truth data in the benchmark is not convincing, the value of such a benchmark will be largely affected, no matter for retrieval or any eval tasks or for whatever metrics.
> > > > > > >
> > > > > > > All ground-truth data in IMP are rigorously human-verified, with explicit annotator guidelines and strong inter-annotator agreement to ensure high-quality data. However, we indeed only have ground-truth data for the five positive captions for each image, yet, we do not believe this diminishes the quality of IMP. For the retrieval task, we assume, *as is the norm in the field*, that any caption not annotated is a negative. It is indeed valid to question this assumption made for retrieval, because as our top-k results show (and as CrissCrossed captions (Parekh et al, 2021) and ECCV captions (Chun et al., 2022) showed previously for MS-COCO) there may be valid captions returned which are counted as negatives.
> > > > > > >
> > > > > > > > I am not convinced by "expect them to similarly affect".
> > > > > > >
> > > > > > > Apologies if we were unclear. Here we intended to express that, for a single VLM, the effect of making this assumption for retrieval should equally influence all five captions within IMP. However, what we are observing is that those captions which we categorise as hard are much less likely to be retrieved by a VLM compared to those that are categorised as easy. Hence concerning relative model performance we were not referring to comparing multiple VLMs, which as you correctly note, indeed applies to all/most benchmarks.
> > > > > > >
> > > > > > > > All (or most) benchmarks are for measuring "relative" model performance.
> > > > > > >
> > > > > > > We fully agree. Specifically for IMP, our focus on relative performance explicitly compares a single VLM’s capabilities across easy, medium, and hard caption sets. For instance, Table 4 clearly illustrates SigLIP’s performance degradation from easy (R@5=38.7) to medium (R@5=11.2) to hard (R@5=0.1). Even assuming that these recall values might be affected by false negatives, the relative difference between categories remains large and explicitly underscores the polysemy challenge faced by VLMs. Importantly, this clear pattern consistently holds for all evaluated VLMs, not just SigLIP.
> > > > > > >
> > > > > > > > no matter for retrieval or any eval tasks or for whatever metrics.
> > > > > > >
> > > > > > > We would like to emphasise that this type of false negative annotations **cannot** affect the Image-Text Binary Classification Evaluation in section 5.3, as those captions are never considered for this evaluation. To illustrate this we refer to appendix F, which clearly shows that for a wide variety of images where there is overwhelming human agreement that these image-text pairs are valid, yet CLIP does not agree.
> > > > > > >
> > > > > > > > VLMs likely have different favors of aligned text given the same image.
> > > > > > >
> > > > > > > Thank you for highlighting this crucial observation—indeed, this point explicitly supports our central finding. Despite differences in their training data distributions and architectures, all evaluated VLMs consistently favor aligning images with captions in the easy set, significantly underperforming on the hard captions. Humans, however, easily identify both easy and hard sets as valid image-caption pairs. This explicitly highlights VLMs’ significant challenge with image polysemy compared to human performance.
> > > > > > >
> > > > > > > Given our experimental setup and explicit polysemy focus, the critical evaluation of IMP’s annotation quality does not hinge upon the presence of other potentially valid captions. Rather, it rests explicitly on whether the five provided captions per image are valid and representative. According to clear annotator consensus, rigorous annotation protocols, and our own careful observations, we confidently conclude that IMP represents a high-quality dataset suited for measuring image polysemy.

---

> > > > > > > > ### Comment · Reviewer_Jbq1 · 2025-04-04
> > > > > > > >
> > > > > > > > It seems we are closing the gap a bit more now:
> > > > > > > > - IMP's quality looks good, if we only look at the positive (ground-truth) captions of one image, i.e., without looking at other captions in IMP.
> > > > > > > > - IMP is more suitable for the eval of a single VLM.
> > > > > > > > - IMP is not very suitable for the eval of multiple VLMs (as most benchmarks did), because it has many abstract captions, which thus can easily introduce false negatives. This also means, papers using IMP are not comparable with each other (e.g. to have a leaderboard).
> > > > > > > >
> > > > > > > > Please correct me if my understanding is wrong. If the statements above is true, maybe we should make its usage more explicit? As it's usage is so different from common VLM benchmarks?

---

> > > > > > > > > ### Author Response · Authors · 2025-04-05
> > > > > > > > >
> > > > > > > > > Thank you for taking the time to understand our work and contributions.
> > > > > > > > >
> > > > > > > > > > IMP's quality looks good, if we only look at the positive (ground-truth) captions of one image, i.e., without looking at other captions in IMP.
> > > > > > > > >
> > > > > > > > > **True**. However, we do want to highlight that this is the norm within the vision-language field. Prior benchmarks like MS-COCO and Flickr30K also only have annotations for positives. Web-scale datasets even only have a single positive, which are not even verified to be valid.
> > > > > > > > >
> > > > > > > > > We could further emphasise this in section 4.1 and the discussion if necessary.
> > > > > > > > >
> > > > > > > > > >IMP is more suitable for the eval of a single VLM.
> > > > > > > > >
> > > > > > > > > **Partially True**. IMP can be used to assess the polysemy performance of a single VLM by comparing its performance on the easy, medium, and hard captions. However, it is also suitable for comparing multiple VLM.
> > > > > > > > >
> > > > > > > > > > IMP is not very suitable for the eval of multiple VLMs (as most benchmarks did), because it has many abstract captions, which thus can easily introduce false negatives. This also means, papers using IMP are not comparable with each other (e.g. to have a leaderboard).
> > > > > > > > >
> > > > > > > > > **False**. Multiple VLM **can** validly be compared on IMP. **The binary classification task is completely unaffected by false negatives** hence comparing multiple VLM on this task is completely fair and suitable, Table 5 could directly be used as a leaderboard.
> > > > > > > > >
> > > > > > > > > For the retrieval task the performance of a VLM may indeed be affected by false negatives, however, this does not make IMP unsuitable for comparing multiple VLM. Allow us to sketch a scenario to illustrate this. Given two hypothetical VLM:
> > > > > > > > > - VLM A, which has R@5 of 40.0 on easy captions, 32.0 on medium captions, and 20.0 on hard captions
> > > > > > > > > - VLM B, which has R@5 of 60.0 on easy captions, 5.0 on medium captions, and 0.0 on hard captions
> > > > > > > > >
> > > > > > > > > We can conclude that VLM A deals much better with polysemy - as the relative difference in performance across the three difficulty categories is much smaller. This comparison can be extended to include additional VLM. As such, **IMP is highly suitable for comparing multiple VLM**.
> > > > > > > > >
> > > > > > > > > In Table 3 we report overall retrieval results, and even if these are affected by false negatives we would still be able to determine which VLM are better at handling polysemy. In particular for the mean and median rank (meanR and medR) metrics we would expect significantly lower values, and the meanR and medR to be more similar for a VLM capable of handling polysemy. As such, the performance of a VLM on IMP reflects their ability in handling polysemy making it a suitable benchmark.
> > > > > > > > >
> > > > > > > > > > As it's usage is so different from common VLM benchmarks?
> > > > > > > > >
> > > > > > > > > IMP is specifically designed for benchmarking the ability of VLM to deal with image polysemy, which indeed makes it different from other VLM benchmarks. However, we believe this is what makes IMP uniquely valuable as a benchmark.
> > > > > > > > >
> > > > > > > > > We could add a subsection in section 5 that explains these points and how IMP can be used for benchmarking and comparing multiple VLM.

---

> > > > > > > > > > ### Comment · Reviewer_Jbq1 · 2025-04-05
> > > > > > > > > >
> > > > > > > > > > > Prior benchmarks like MS-COCO and Flickr30K also only have annotations for positives. Web-scale datasets even only have a single positive, which are not even verified to be valid.
> > > > > > > > > >
> > > > > > > > > > As I mentioned already, other benchmarks like COCO have much less false-negative problems as in IMP. This is because the captions in other benchmarks focus more on semantic description of the objects in the images, while IMP focuses more on *abstract" polysemy captions. Also as demonstrated in your qualitative top-k results, IMP has severe false-negative problems.
> > > > > > > > > >
> > > > > > > > > > > Multiple VLM can validly be compared on IMP. The binary classification task is completely unaffected by false negatives hence comparing multiple VLM on this task is completely fair and suitable, Table 5 could directly be used as a leaderboard.
> > > > > > > > > >
> > > > > > > > > > AUPRC is used in Table 5 as the metric, which is to measure the classification model's performance on ranking positive (matched pairs) and negative (unmatched pairs) examples. The false-negative problem still exists here.
> > > > > > > > > >
> > > > > > > > > >
> > > > > > > > > > TLDR: IMP has severe false-negative problems, compared to other benchmarks, partially due to the problem it aims to solve, partially due to the method to build it. This problem makes existing metrics including zero-shot retrieval and classification (actually they measure the same capability essentially) less convincing, thus I wouldn't recommend it as a high-quality benchmark for the eval *across VLMs*.
> > > > > > > > > >
> > > > > > > > > > I hope I have made my point clearer now.

---

> > > > > > > > > > > ### Author Response · Authors · 2025-04-05
> > > > > > > > > > >
> > > > > > > > > > > We are grateful to the reviewer for providing further clarification, the point is indeed much clearer now.
> > > > > > > > > > >
> > > > > > > > > > > > As I mentioned already, other benchmarks like COCO have much less false-negative problems as in IMP. This is because the captions in other benchmarks focus more on semantic description of the objects in the images, while IMP focuses more on *abstract" polysemy captions. Also as demonstrated in your qualitative top-k results, IMP has severe false-negative problems.
> > > > > > > > > > >
> > > > > > > > > > > We would like to emphasise that prior benchmarks, like COCO, have "massive false negatives" - to quote Chun et al (2022). However, it is reasonable to note that the retrieval performance on IMP is affected by this - yet, this is indeed inherent to the problem we are studying, and inherent to web-scale datasets as well. As these web-scale datasets are used to train present day VLM we believe it is crucial to understand how VLM handle polysemy.
> > > > > > > > > > >
> > > > > > > > > > > > AUPRC is used in Table 5 as the metric, which is to measure the classification model's performance on ranking positive (matched pairs) and negative (unmatched pairs) examples. The false-negative problem still exists here.
> > > > > > > > > > >
> > > > > > > > > > > Apologies if this is not emphasised sufficiently in the paper. In section 4.3 (4th paragraph, right above figure 7) it is explained that the negatives used for the calculation of the AUPRC are based on captions which the human annotators noted as not suitable for the image. Hence, the binary classifcation task uses **true positives** and **true negatives**.
> > > > > > > > > > >
> > > > > > > > > > > We hope this is sufficient to demonstrate that this task is indeed fair and suitable for comparing multiple VLM.
> > > > > > > > > > >
> > > > > > > > > > > > thus I wouldn't recommend it as a high-quality benchmark for the eval across VLMs.
> > > > > > > > > > >
> > > > > > > > > > > We hope the reviewer's concern can be addressed by extending the discussion to emphasise the issue of false negatives and how it influences benchmarking on IMP. We would thus note in this discussion that:
> > > > > > > > > > > - The binary classification task is suitable for comparing multiple VLM
> > > > > > > > > > > - The relative difference in performance across the easy, medium, hard captions is suitable for comparing multiple VLM
> > > > > > > > > > > -  Because of potential false negatives the raw retrieval metrics may not fully express the performance of a VLM, hence this should be used to draw conclusions about how these VLM handle polysemy and not to directly compare across VLM.
> > > > > > > > > > >
> > > > > > > > > > > We hope that with this addendum to the discussion all the reviewer's concerns are addressed, and we thank the reviewer for their continued effort in helping us refine the manuscript.

---

> > > > > > > > > > > > ### Comment · Reviewer_Jbq1 · 2025-04-05
> > > > > > > > > > > >
> > > > > > > > > > > > > Apologies if this is not emphasised sufficiently in the paper. In section 4.3 (4th paragraph, right above figure 7) it is explained that the negatives used for the calculation of the AUPRC are based on captions which the human annotators noted as not suitable for the image. Hence, the binary classifcation task uses true positives and true negatives.
> > > > > > > > > > > >
> > > > > > > > > > > > Good to know there are true negatives for eval. In section 4.3, it says there are 5k true negatives. IIUC, IMP has 5k*5=25k pairs. To clarify, how many images are used in the 5k true negatives?

---

> > > > > > > > > > > > > ### Author Response · Authors · 2025-04-07
> > > > > > > > > > > > >
> > > > > > > > > > > > > > To clarify, how many images are used in the 5k true negatives?
> > > > > > > > > > > > >
> > > > > > > > > > > > > As reported in section 4.3 there is "one true negative for each image", so all 5K IMP images are used.
> > > > > > > > > > > > >
> > > > > > > > > > > > > For Table 5, the AUPRC is calculated per difficulty category, so the distribution of samples is as follows:
> > > > > > > > > > > > > - Easy: 6344 positives, 5K negatives
> > > > > > > > > > > > > - Medium: 13808 positives, 5K negatives
> > > > > > > > > > > > > - Hard: 4848 positives, 5K negatives
> > > > > > > > > > > > >
> > > > > > > > > > > > > Here, the negatives are re-used across the difficulty levels. Since AUPRC is robust to imbalance this well reflects the performance of the VLM.

---

> > > > > > > > > > > > > > ### Comment · Reviewer_Jbq1 · 2025-04-07
> > > > > > > > > > > > > >
> > > > > > > > > > > > > > Thanks for the explanation. 5K negative examples sounds good. It would be nicer if we have the same 5K positive examples for each difficulty, as this will be more balanced and comparable across different difficulties.
> > > > > > > > > > > > > >
> > > > > > > > > > > > > > To explain why I care so much on the negative examples, even though there have been many false negatives in COCO and other existing benchmarks: the task IMP targets is more difficult than the benchmarks based on ground-truth objects, and easier to introduce false negative in evaluation, which makes it less useful for any ranking based metrics, including both retrieval, AUPRC etc. If we think this is not a problem for IMP, we'd better sample some data, and use 1) human evals to quantitatively compare the percentage of false negatives between IMP and other benchmarks like COCO, 2) qualitative top-k (both image to text and text to image) evals to show readers the quality difference between IMP and others.
> > > > > > > > > > > > > >
> > > > > > > > > > > > > > Or simpler, we can use the annotated 5k false negative examples for all evals, which IMO is more convincing than the larger raw IMP dataset. If we choose this approach, could you also present the new retrieval results?

---

> > > > > > > > > > > > > > > ### Author Response · Authors · 2025-04-08
> > > > > > > > > > > > > > >
> > > > > > > > > > > > > > > We thank reviewer Jbq1 for their their additional suggestions.
> > > > > > > > > > > > > > >
> > > > > > > > > > > > > > > > use the annotated 5k false negative examples for all evals
> > > > > > > > > > > > > > >
> > > > > > > > > > > > > > > Originally, we chose to only use the 5K *true* negative samples in the binary classification task for three reasons:
> > > > > > > > > > > > > > > 1. To lower the barrier of using IMP. We followed the retrieval setup of MS-COCO, such that existing evaluation code can be trivially adapted to IMP.
> > > > > > > > > > > > > > > 2. Retrieval with only true positives (and assuming that all other captions are negative) closely matches the training setup, as contrastively trained VLM are optimised with a loss that similarly assumes that all off-diagonal samples are negatives.
> > > > > > > > > > > > > > > 3. It is not obvious how to incorporate true negatives in retrieval - as the true negatives are only pairwise to a single image. As such they may be false negatives for other images.
> > > > > > > > > > > > > > >
> > > > > > > > > > > > > > > > If we choose this approach, could you also present the new retrieval results?
> > > > > > > > > > > > > > >
> > > > > > > > > > > > > > > We have amongst the authors discussed how to do this, but we are unfortunately not sure how to go about this in a manner that would lead to interpretable results. We discussed a number of variants:
> > > > > > > > > > > > > > >
> > > > > > > > > > > > > > > 1. Use the 5 positive captions and the 5K true negatives as the entire dataset for each image (retrieval on 5005 samples per image). However, this still introduces potential false negatives.
> > > > > > > > > > > > > > > 2. Ranking of 6 samples (5 true positives and 1 true negative) per image
> > > > > > > > > > > > > > > 3. Simply adding the 5K true negatives to the dataset, so the retrieval task is over ~30K captions instead of ~25K.
> > > > > > > > > > > > > > >
> > > > > > > > > > > > > > > We believe variants 1 and 2 would effectively be worse versions of the binary classification experiment, whereas variant 3 would most likely worsen the false negative problem as it would add 1 true negative and 4999 potential false negatives. As such we believe the original setup to be preferable to these variants. Especially, due to the combination of having both the retrieval and classification tasks.
> > > > > > > > > > > > > > >
> > > > > > > > > > > > > > > To explore this direction we would appreciate further guidance from the reviewer.
> > > > > > > > > > > > > > >
> > > > > > > > > > > > > > > > 2) qualitative top-k (both image to text and text to image) evals
> > > > > > > > > > > > > > >
> > > > > > > > > > > > > > > If we were to incorporate this suggestion by the reviewer - to show the additional top-k results - would that address their concerns? We believe this is the most transparent, whilst IMP remains easy to use, which has our preference.

---

> > > > > > > > > > > > > > > > ### Comment · Reviewer_Jbq1 · 2025-04-12
> > > > > > > > > > > > > > > >
> > > > > > > > > > > > > > > > Thanks for the further explanation.
> > > > > > > > > > > > > > > >
> > > > > > > > > > > > > > > > > It is not obvious how to incorporate true negatives in retrieval - as the true negatives are only pairwise to a single image. As such they may be false negatives for other images.
> > > > > > > > > > > > > > > >
> > > > > > > > > > > > > > > > Indeed, so the true negatives can only be used for the binary classification task.
> > > > > > > > > > > > > > > >
> > > > > > > > > > > > > > > > > To explore this direction we would appreciate further guidance from the reviewer.
> > > > > > > > > > > > > > > >
> > > > > > > > > > > > > > > > I agree with you on the proposed 3 approaches, which are all not ideal to solve the false negative problem.
> > > > > > > > > > > > > > > >
> > > > > > > > > > > > > > > > So if we still would like to use IMP for zero-shot *retrieval* evals, IMO, to get convincing eval results, the only left way is to conduct experiments and analysis to demonstrate false-negative is not a big issue. This includes both image-to-text and text-to-image retrievals. I will be convinced by such an analysis on sampled (instead of cherry picked) data.

---

> > > > > > > > > > > > > > > > > ### Author Response · Authors · 2025-04-14
> > > > > > > > > > > > > > > > >
> > > > > > > > > > > > > > > > > > So if we still would like to use IMP for zero-shot retrieval evals [..] demonstrate false-negative is not a big issue.
> > > > > > > > > > > > > > > > >
> > > > > > > > > > > > > > > > > We thank the reviewer for their suggestion, but we reiterate that because IMP is not intended as a generic vision-language retrieval evaluation dataset this is beyond the scope of the paper. IMP is intended **specifically for the evaluation of VLMs’ ability in handling polysemy**, and our findings about VLMs in relation to polysemy hold even without solving the false negative problem.
> > > > > > > > > > > > > > > > >
> > > > > > > > > > > > > > > > > To illustrate why the false negative problem can be ignored, we construct an example using one image, three paired positive captions (A, B, C), and five unannotated captions ($X_{0-4}$). If we perform retrieval using the image as query, a resulting ranking may look like:
> > > > > > > > > > > > > > > > > 1. B
> > > > > > > > > > > > > > > > > 2. $X_0$
> > > > > > > > > > > > > > > > > 3. $X_4$
> > > > > > > > > > > > > > > > > 4. C
> > > > > > > > > > > > > > > > > 5. $X_2$
> > > > > > > > > > > > > > > > > 6. $X_1$
> > > > > > > > > > > > > > > > > 7. A
> > > > > > > > > > > > > > > > > 8. $X_3$
> > > > > > > > > > > > > > > > >
> > > > > > > > > > > > > > > > > At this point we can clearly observe that the ranking of A is worse than the ranking of C, and B is ranked highest - **this can be concluded without considering whether $X_{0-4}$ are false negatives or true negatives**, as this does not change that the ranking of A, B, C is B < C < A.
> > > > > > > > > > > > > > > > >
> > > > > > > > > > > > > > > > > For IMP we scale this up by having three sets of captions (easy, medium, hard) which we are able to rank, and we find that easy < medium < hard - consistently across multiple VLM (different architectures and different training data) - this result allows us to draw conclusions about how VLM handle image polysemy.
> > > > > > > > > > > > > > > > >
> > > > > > > > > > > > > > > > > In this manner IMP can be used to evaluate VLMs **in how they handle image polysemy** without solving the false negative problem.
> > > > > > > > > > > > > > > > >
> > > > > > > > > > > > > > > > > We propose to expand the discussion section of the paper to clearly demarcate the purpose of IMP and which type of conclusions it does and does not facilitate. We hope that this addition to the paper would address the reviewer’s concerns.

---

> > > > > > > > > > > > > > > > > > ### Comment · Reviewer_Jbq1 · 2025-04-16
> > > > > > > > > > > > > > > > > >
> > > > > > > > > > > > > > > > > > > we reiterate that because IMP is not intended as a generic vision-language retrieval evaluation dataset this is beyond the scope of the paper.
> > > > > > > > > > > > > > > > > >
> > > > > > > > > > > > > > > > > > Please note that I never claimed that "IMP is intended as a generic vision-language retrieval evaluation dataset".
> > > > > > > > > > > > > > > > > >
> > > > > > > > > > > > > > > > > > > IMP is intended specifically for the evaluation of VLMs’ ability in handling polysemy
> > > > > > > > > > > > > > > > > >
> > > > > > > > > > > > > > > > > > Yes, that is why I keep asking relevant questions. What is the convincing metric "for the evaluation of VLMs’ ability in handling polysemy"? In 5.1 Evaluation Metrics section, cross-modal retrieval is claimed as the main metric for the eval of "VLMs’ ability in handling polysemy". Is there any misunderstanding here?
> > > > > > > > > > > > > > > > > >
> > > > > > > > > > > > > > > > > > > our findings about VLMs in relation to polysemy hold even without solving the false negative problem.
> > > > > > > > > > > > > > > > > > > ...
> > > > > > > > > > > > > > > > > > > this can be concluded without considering whether
> > > > > > > > > > > > > > > > > >  are false negatives or true negatives, as this does not change that the ranking of A, B, C is B < C < A.
> > > > > > > > > > > > > > > > > >
> > > > > > > > > > > > > > > > > > However, in this paper you proposed to use "Recall@K (R@K) with K = 1, 5, 10" as the metric. IMO, false negatives will affect the the top-k retrieval accuracy. To measure "the ranking of A, B, C", you might need to consider other metrics.
> > > > > > > > > > > > > > > > > >
> > > > > > > > > > > > > > > > > > > In this manner IMP can be used to evaluate VLMs in how they handle image polysemy without solving the false negative problem.
> > > > > > > > > > > > > > > > > >
> > > > > > > > > > > > > > > > > > For the reasons above, sorry that I am still not convinced about the retrieval metric for IMP. Please let me know if my understanding above has any mistakes. Thanks

---

> > > > > > > > > > > > > > > > > > > ### Author Response · Authors · 2025-04-16
> > > > > > > > > > > > > > > > > > >
> > > > > > > > > > > > > > > > > > > > In 5.1 Evaluation Metrics section, cross-modal retrieval is claimed as the main metric for the eval of "VLMs’ ability in handling polysemy". Is there any misunderstanding here?
> > > > > > > > > > > > > > > > > > >
> > > > > > > > > > > > > > > > > > > As far as we can determine this claim is not made in section 5.1. Instead, the first sentence of section 5.1 reads:
> > > > > > > > > > > > > > > > > > >
> > > > > > > > > > > > > > > > > > > `We employed a combination of standard cross-modal retrieval metrics and binary classification metrics to
> > > > > > > > > > > > > > > > > > > benchmark model performance, tailoring our approach to the challenges of image polysemy.`
> > > > > > > > > > > > > > > > > > >
> > > > > > > > > > > > > > > > > > > Which proposes the combination of cross-modal retrieval **and** binary classification.
> > > > > > > > > > > > > > > > > > >
> > > > > > > > > > > > > > > > > > > > However, in this paper you proposed to use "Recall@K (R@K) with K = 1, 5, 10" as the metric.  [...]  you might need to consider other metrics.
> > > > > > > > > > > > > > > > > > >
> > > > > > > > > > > > > > > > > > > Thank you for this suggestion, however, as stated in section 5.1, we already consider other metrics:
> > > > > > > > > > > > > > > > > > >
> > > > > > > > > > > > > > > > > > > `However, while R@K is a standard metric, it falls short in evaluating polysemy, as
> > > > > > > > > > > > > > > > > > > retrieving even a single easy caption can result in high scores without reflecting the ranking quality
> > > > > > > > > > > > > > > > > > > for other, more diverse captions. To address this, we include mean rank (meanR) and median rank
> > > > > > > > > > > > > > > > > > > (medR), which provide a more holistic view of the overall ranking of all relevant captions, capturing
> > > > > > > > > > > > > > > > > > > how well the model ranks diverse captions throughout the retrieval list (Parekh et al., 2021). `
> > > > > > > > > > > > > > > > > > >
> > > > > > > > > > > > > > > > > > > As such (and as can be observed from the Tables) we report R@K, meanR, and medR, as well as mAP@R. This latter metric was chosen based on findings in the ECCV caption paper by Chun et al. which shows that mAP@R is better aligned with humans than R@K. We would be happy to consider additional metrics if the reviewer has suggestions for metrics that would give additional insights.
> > > > > > > > > > > > > > > > > > >
> > > > > > > > > > > > > > > > > > > > To measure "the ranking of A, B, C"
> > > > > > > > > > > > > > > > > > >
> > > > > > > > > > > > > > > > > > > > Please let me know if my understanding above has any mistakes. Thanks
> > > > > > > > > > > > > > > > > > >
> > > > > > > > > > > > > > > > > > > Perhaps the misunderstanding stems from the difference between *absolute* and *relative* ranking. False negatives indeed influence the *absolute* ranking, however, conclusions about polysemy can be drawn based on a **relative** ranking - which as we highlighted in our previous response is not influenced by false negatives.

---

> > > > > > > > > > > > > > > > > > > > ### Comment · Reviewer_Jbq1 · 2025-04-19
> > > > > > > > > > > > > > > > > > > >
> > > > > > > > > > > > > > > > > > > > > As far as we can determine this claim is not made in section 5.1. Instead, the first sentence of section 5.1 reads:
> > > > > > > > > > > > > > > > > > > > > Thank you for this suggestion, however, as stated in section 5.1, we already consider other metrics:
> > > > > > > > > > > > > > > > > > > >
> > > > > > > > > > > > > > > > > > > > I know that more than retrieval metrics are proposed in this paper, but the key question on my side is **which are effective metrics and which are not**. To encourage others to use this new benchmark, this question must be answered clearly.
> > > > > > > > > > > > > > > > > > > >
> > > > > > > > > > > > > > > > > > > > However, in this paper, even though multiple metrics are reported, retrieval is recommended as the primary metric as demonstrated in Table 3 and 4, without highlighting the limitation I pointed out above.
> > > > > > > > > > > > > > > > > > > >
> > > > > > > > > > > > > > > > > > > > Before discussing other relatively less critical questions, maybe we should take one step back here and first reach a consensus on my original question: **whether retrieval is a convincing metric for IMP or not**? So far my answer is negative, because of the lack of analysis of false negatives, which affects IMP much more than other retrieval benchmarks due to the nature of this new benchmark (captions are more subjective instead of objective).

---

### Review · Reviewer_kbz2 · 2025-03-12

**Summary Of Contributions:**

This work explors the image polysemy problem in image-text pairs, where a single image can correspond to multiple, semantically diverse captions. The work identifies that current contrastive vision-language models struggle to handle polysemy in image-text datasets. The authors first identify image polysemy using a clustering-based approach. Then, as a data verification step involving both human evaluation and model predictions, they categorize samples using an agreement-based method across multiple CLIP models. Finally, they benchmark the proposed IMP dataset using multiple evaluation metrics.

**Audience:**

Yes

**Broader Impact Concerns:**

There is no potential issue.

**Claims And Evidence:**

Yes

**Requested Changes:**

I expect further discussion on the conflict between the use of the CLIP model and the paper's assumption, the underlying reasons for existing models' weaknesses in image polysemy, the significance of addressing the challenge, and potential approaches for improving image polysemy.

**Strengths And Weaknesses:**

Strengths

[+] It is quite surprising that from the qualitative demonstration, CLIP-based matching score could be extremely low although human evaluates the pair as positive matching. The problem formulation and verification is well constructed.

[+] The proposed IMP dataset and evaluation framework would provide a valuable tool for assessing the image polysemy capabilities of vision-language models, supported by thorough analyses in the paper.

---


Weaknesses

[-] In section 3.1, the use of CLIP-based image-text similarity for identifying polysemy is less convincing, as it seems to conflict with the basic assumption of this work that contrastive VLMs struggle with image polysemy.

[-] While this work well identifies and verifies the presence of image polysemy in VLMs, it lacks a deeper systematic analysis of the underlying causes of this phenomenon. Additionally, further exploration into the reasons behind model disagreements on semantically diverse captions would be particularly insightful.

[-] Although it is intuitive that humans are good at image polysemy, the fundamental reasons for addressing this challenge and its importance are not clearly presented. For instance, is there a correlation between a model's image polysemy and its performance on downstream tasks such as zero-shot classification and retrieval?

[-] Any discussion on improving image polysemy in models is missing.

---

Minor

- The scope of VLMs could be broader. For example, VQAScore [1] can be used to measure image-text similarity in decoder-based models like LLaVA. Including a family of such decoder-based VLMs for benchmarking the IMP dataset would be interesting.

- Some typographical errors: 'whish' (p. 4),

---

References

[1] Evaluating Text-to-Visual Generation with Image-to-Text Generation, in ECCV 2024.

---

> ### Author Response · Authors · 2025-03-26
> **Rebuttal Response (1/2)**
>
> We thank reviewer kbz2 for their insightful comments and their recognition of IMP as a valuable research tool for future research. In the following we address the concerns they raised.
>
> ### Q1: Conflict between using CLIP similarity for polysemy identification and the assumption that VLMs struggle with polysemy
>
> We believe this observation by the reviewer may be due to ambigious explanation in the paper about how we determine polysemy, which was also noted by reviewer Jbq1. We provided clarification of this procedure in our response to reviewer Jbq1 under question 4 (about Figure 3).
>
> The core idea of this procedure is to actually leverage the poor ability of VLM to handle polysemy. For the geodesic distance metric we look at a set of visually similar images (in the CLIP embedding space); if CLIP were able to well-handle polysemy then the captions for this set of images would all be close to their respective images, and hence close to each other. However, what we observe is that (especially for CC12M) the captions paired with these images are far away in CLIP embedding space, even when the images are highly similar. By leveraging this mismatch in CLIP embedding space (i.e., images being similar but their respective captions being dissimilar) we are thus able to identify polysemy.
>
> We have expanded the discussion in section 3.1 to clarify this.
>
> ### Q2: Lack of analysis on underlying reasons why VLMs struggle with polysemy
>
> We believe that the reason why VLM struggle with image polysemy follows from their training objective. The vision-language contrastive loss implicitly assumes, by forcing image and caption to be close in the embedding space, that co-captions (i.e., other captions belonging to the same image) are semantically similar. This assumption generally holds for datasets such as MS-COCO, which were designed to have straightforward descriptive captions.
>
> However, in the case of image polysemy we have a single image (or a set of visually similar images) described by multiple captions with different semantics. Based on the text semantics these captions should be far apart in embedding space, but based on image semantics they should be close. This is not something that the contrastive loss can resolve perfectly, but one suboptimal resolution may be to strive for coherent embeddings (i.e., embedding distance aligns with semantic distance) and simply not aligning all captions to their paired image. We believe this is what occurs in VLM training, and hence VLM struggle with polysemy. We hope that IMP can provide a foundation to demonstrate this phenomenon more rigorously.
>
> We have added a discussion (Section 6) in which we discuss this issue.
>
> ### Q3: Importance and motivation for addressing polysemy
>
> As image polysemy occurs naturally in human communication we believe it is crucial to address; while there may be benefits to downstream tasks, this is currently hard to demonstrate as most datasets do not account for it. Moreover, we believe that one of the main reasons to study image polysemy is actually for the construction of training datasets.
>
> A common procedure for creating vision-language datasets is to collect a large amount of webdata, for instance from Common Crawl, and subsequently to filter this data based on some metric that ideally excludes as few valid pairs while excluding as much noise as possible. DataComp (Gadre et al., 2023) (and LAION as well) use CLIP to filter Common Crawl - in the DataComp paper it is shown that a CLIP similarity threshold (for ViT-L/14) of 0.129, only discards 10% of the data, whereas a threshold of 0.384 discards 99% of the data. Not only is this a very small range of similarity thresholds to go from hardly discarding anything to discarding almost everything, even the least strict threshold would discard captions which are valid according to human annotators (see Figure 4).
>
> Hence, a better understanding of image polysemy is necessary to construct better vision-language datasets, as currently we do not have good mechanisms to construct datasets from webdata. Moreover, despite the limitations of contrastive loss, it may be that under the right circumstances we may observe the ability to handle polysemy as an emergement behaviour. In order to measure this, a benchmark like IMP is necessary.
>
> We have added a discussion (Section 6) in which we discuss this issue.
>
> ## Q4: No discussion on potential approaches to improve model handling of polysemy
>
> We have added a paragraph in the conclusion (Section 6) in which we discuss promising directions for improving handling of polysemy.

---

> ### Author Response · Authors · 2025-03-26
> **Rebuttal Response (2/2)**
>
> ## Q5: VQAscore
>
> We thank reviewer for the suggestion. Though decoder-based models are not within the scope of this paper, we calculate both the CLIPscore and the VQAscore for IMP, in the table below we report these scores for the entire dataset as well as for each split. We chose CLIP-flant5-xl as the VQAscore model, and CLIP-ViT/B-32 (trained on CLIP400M) as the CLIPscore model.
>
> | Category | CLIPscore | VQAscore |
> | -------- | -------- | -------- |
> | Full IMP     |  0.25 +- 0.03   | 0.66 +- 0.17    |
> | Easy     |  0.28 +- 0.02   | 0.73 +- 0.16   |
> | Medium     |  0.25 +- 0.02   | 0.66 +- 0.16   |
> | Hard     |  0.22 +- 0.02   | 0.58 +- 0.17   |
>
> We further compare the distribution of two scores using various of statistical analysis methods (correlation, mutual information, etc.). Some analysis and plots can be found in ***Appendix J***.
>
> From these statistical analysis we conclude that VQAscore measures different semantic aspects than CLIPscore, it has better alignment with human in some medium/hard categories, but also fall short in some others where CLIPscores agree. This indicates that VQAscore is still not a comprehensive solution to image polysemy. If necessary we can further expand the analysis with detailed numbers.

---

> > ### Comment · Reviewer_kbz2 · 2025-04-05
> >
> > Thank you for the detailed response and the revised manuscript. These are carefully checked and the problem formulation of image polysemy is well contextualized. There is no remaining concern to me.

---

### Author Response · Authors · 2025-03-26
**General Rebuttal Response**

We thank all reviewers for their thoughtful and constructive feedback, and their appreciation of the importance and novelty of our work. In response to their insightful comments, we have revised the manuscript to enhance clarity and rigor:
- We refined our explanations and improved clarity in Section 3 and 4, particularly regarding our methodology and definitions.
- We introduced a new Section 6 (Discussion), explicitly addressing motivations, underlying reasons for the observed limitations in VLMs, implications for dataset filtering practices, and potential future directions to handle image polysemy.
- We added extensive qualitative and quantitative analyses in the Appendices (G, H, I, and J) to comprehensively clarify dataset comparisons and robustness.

We have uploaded a revised version of the manuscript and all textual changes are marked in blue.

Additionally, we have responded in detail to each reviewer’s individual comments, carefully addressing specific suggestions and questions in the corresponding responses below.

---

### Decision · Action_Editor_3K8Y · 2025-04-20

**Recommendation:** Reject

**Comment:**

This paper explores the image polysemy problem in image-text pairs, and introduces a new benchmark IMP (Image Polysemy) to measure CLIP’s understanding of image captions of different levels (easy/medium/hard). The authors have had thorough discussions with the reviewers during the rebuttal. By the end, it received Leaning Accept, Leaning Reject, and Reject recommendations.

On one hand, one reviewer commented that identifying and quantifying the image polysemy capabilities of VLMs through the proposed benchmark and evaluation framework is both interesting and impactful, and this paper would be a valuable contribution as a benchmark paper. On the other hand, the other two reviewers think that several concerns still remain. (1) There lacks of enough evidence to show IMP proposed in this paper is a fair and convincing benchmark to measure polysemy. Through several rounds of detailed discussions, this concern still cannot be well resolved, which the AC also partially agreed with. (2) Although the problem regarding to caption difficulty is very useful and important, overall, it seems that the whole problem is not well-defined and too subjective. Meanwhile, it would be better if the authors could provide extra solutions to handle the current problem and issues.

Given the overall negative feedback and the remaining concerns after rebuttal, the AC would like to recommend rejection by the end.

**Audience:**

Yes.

**Claims And Evidence:**

No. The claims and evidence can be made more convincing via addressing the reviewers' major concerns.